# Bora phosphorylation substitutes in trans for T-loop phosphorylation in Aurora A to promote mitotic entry

N. Tavernier[1,10], Y. Thomas[2], S. Vigneron[3], P. Maisonneuve [1], S. Orlicky[1], P. Mader[1], S. G. Regmi[4], L. Van Hove[2], N. M. Levinson [5], G. Gasmi-Seabrook[6], N. Joly[2], M. Poteau[7], G. Velez-Aguilera[2], O. Gavet[7], A. Castro [3], M. Dasso [4], T. Lorca[3], F. Sicheri [1,8,9,11✉] & L. Pintard [2,11✉]

Polo-like kinase 1 (Plk1) is instrumental for mitotic entry and progression. Plk1 is activated by phosphorylation on a conserved residue Thr210 in its activation segment by the Aurora A kinase (AURKA), a reaction that critically requires the co-factor Bora phosphorylated by a CyclinA/B-Cdk1 kinase. Here we show that phospho-Bora is a direct activator of AURKA kinase activity. We localize the key determinants of phospho-Bora function to a 100 amino acid region encompassing two short Tpx2-like motifs and a phosphoSerine-Proline motif at Serine 112, through which Bora binds AURKA. The latter substitutes in *trans* for the Thr288 phospho-regulatory site of AURKA, which is essential for an active conformation of the kinase domain. We demonstrate the importance of these determinants for Bora function in mitotic entry both in Xenopus egg extracts and in human cells. Our findings unveil the activation mechanism of AURKA that is critical for mitotic entry.

[1] Centre for Systems Biology, Lunenfeld Tanenbaum Research Institute, Sinai Health System, Toronto, ON, Canada. [2] Programme équipe Labellisée Ligue Contre le Cancer, Institut Jacques Monod, UMR7592, Université de Paris, CNRS, Paris, France. [3] Centre de Recherche de Biologie cellulaire de Montpellier, UMR 5237, Université de Montpellier, CNRS, Montpellier, France. [4] Eunice Kennedy Shriver National Institute of Child Health and Human Development, Bethesda, MD, USA. [5] Department of Pharmacology, University of Minnesota, Minneapolis, MN, USA. [6] Princess Margaret Cancer Centre, University Health Network, Toronto, ON, Canada. [7] Institut Gustave Roussy CNRS UMR9019, Villejuif, France. [8] Department of Molecular Genetics, University of Toronto, Toronto, ON, Canada. [9] Department of Biochemistry, University of Toronto, Toronto, ON, Canada. [10] Present address: Programme équipe Labellisée Ligue Contre le Cancer, Institut Jacques Monod, UMR7592, Université de Paris, CNRS, Paris, France. [11] These authors contributed equally: F. Sicheri, L. Pintard. ✉email: sicheri@lunenfeld.ca; Lionel.pintard@ijm.fr

At onset of mitosis, eukaryotic cells undergo a drastic cellular reorganization[1]. The chromatin condenses into individual chromosomes and the nuclear envelope breaks down to permit the capture of chromosomes by the microtubules emanating from the centrosomes, leading to the assembly of the mitotic spindle. This choreography is orchestrated in space and time by evolutionarily conserved mitotic kinases including Aurora kinases (AURK), CyclinA/B-Cdk1, and the Polo-like kinase 1 (Plk1) (for review, see refs. [2–4]). In particular, Plk1 promotes chromosome condensation, nuclear envelope breakdown (NEBD), centrosome maturation, and kinetochore-microtubule attachments (for review, see refs. [5–7]). In addition to a role in this cellular reorganization, Plk1 triggers the G2/M transition[8,9] and is essential for the cell cycle restart after recovery from DNA damage[10]. To coordinate these different functions, Plk1 is dynamically recruited to very specific subcellular sites where it phosphorylates a plethora of substrates.

Plk1 is composed of two functional domains (Fig. 1a): an amino-terminal kinase domain (KD) and a C-terminal polo-box domain (PBD) critical for Plk1 subcellular localization and substrate targeting[11–13]. The PBD binds to phosphorylated motifs known as Polo-docking sites, generated by other kinases, notably by Cyclin-Cdk or Plk1 itself[14]. The PBD also regulates Plk1 kinase activity through repressive intramolecular interactions with the KD[15]. The binding of phosphopeptides to the PBD relieves these auto-inhibitory interactions thereby tightly coupling localization/substrate recognition with catalytic activation[16–18].

In its auto-inhibitory (closed) conformation, Plk1 is cytoplasmic. Upon activation at mitotic entry, Plk1 transitions to an open conformation that also exposes a nuclear localization sequence (NLS) within the KD to binding by importins[16,17]. Thus, Plk1 activation at mitotic entry is further coupled to its nuclear import[17–19].

Plk1 activation also depends on trans-phosphorylation of a conserved residue (Thr210) in its activation segment (T-loop)[20] by the Aurora A kinase (AURKA)[21,22], a member of the larger Aurora family that includes AURKB and AURKC (Fig. 1a and Supplementary Fig. S1). T-loop phosphorylation critically requires the evolutionarily conserved protein Bora[21–23] phosphorylated by Cyclin-Cdk[24–27]. Bora contains two evolutionarily conserved Cyclin-binding motifs (Cy)[25] (Fig. 1a) and its phosphorylation by Cyclin-Cdk is essential for Plk1 function in mitotic entry in *Caenorhabditis elegans* embryos[24], in Xenopus egg extracts[27], and for cell cycle restart after checkpoint recovery from DNA damage in human cells[25,26]. CyclinA-Cdk1, which coordinates DNA replication and mitotic entry, is a positive upstream regulator of Bora in Xenopus[27] and most likely in human cells where CyclinA appears to translocate into the cytoplasm in G2 phase to interact with and initiate Plk1 activation[9,28]. As Bora is exclusively cytoplasmic in human cells[28–30], it is likely responsible for activating the cytoplasmic pool of Plk1 at the G2–M transition thereby triggering Plk1 nuclear import.

While Bora and its phosphorylation by CyclinA-Cdk1 are essential for Plk1 activation by AURKA at mitotic entry, the precise mechanism by which Bora contributes to T-loop phosphorylation of Plk1 is not understood. Cyclin-Cdk phosphorylates Bora at multiple sites including notably a polo-docking site (S252), which primes Bora binding to the Plk1 PBD[22,31]. This binding event was proposed to increase the accessibility of T210, thereby supporting T-loop phosphorylation by AURKA[22,32]. However, a Bora[1–224] fragment, lacking the polo-docking site S252, can still promote AURKA-dependent phosphorylation of Plk1 on the T-loop[25] and support mitotic entry in Xenopus egg extracts[27], suggesting that Bora harbors additional functions beyond opening of the Plk1 structure.

In one possible model, Bora could act by interacting with and promoting the activation of AURKA. Indeed, precedents for the activation of Aurora family kinases by trans-acting factors are well documented[4,33–35]. For example, the microtubule and mitotic spindle regulator Tpx2 (Targeting Protein for Xenopus kinesin-like protein 2) allosterically binds the N-lobe of the AURKA KD to promote an active conformation competent for microtubule nucleation and mitotic spindle assembly[36–40]. In addition to triggering auto-phosphorylation of AURKA on activation segment residues Thr287 or Thr288 and by preventing T-loop dephosphorylation by phosphatases[40], Tpx2 may also allosterically activate AURKA independent of a need for T-loop phosphorylation[41,42].

To decipher the mechanism by which phospho-Bora (hereafter pBora) promotes T-loop phosphorylation on Plk1 by AURKA, we use a combination of biochemical and structural approaches along with functional studies in Xenopus and in human cells. We show that Bora contains three critical elements required to maximally activate AURKA kinase activity, namely two linear Tpx2-like motifs and a phosphorylated motif on Ser112 generated by Cyclin-Cdk activity. These elements act synergistically to stabilize AURKA in an active conformation, allowing AURKA to phosphorylate and activate Plk1. In particular, the Tpx2-like motifs position the Ser112 phospho-motif toward the kinase active site to substitute in *trans* for phosphorylation of the T-loop on AURKA.

## Results

### The KD of Plk1 is a sufficient substrate for pBora potentiated phosphorylation by AURKA.
Previous works established that an amino terminal fragment of Bora (residues 1–224) is sufficient to stimulate phosphorylation of Plk1 on the critical T-loop regulatory site T210 (henceforth Plk1 phosphorylation) by AURKA in vitro[25] and in Xenopus egg extracts[27]. To investigate the minimal determinants for this phosphorylation event, we reconstituted the reaction in vitro using purified components (Fig. 1a, b). As shown in Fig. 1c, in the absence of Bora, no phosphorylation of Plk1 was observed (lanes 3, 8, and 13). However, upon addition of Bora[1–224], weak phosphorylation of Plk1 was detected (lanes 2, 7, and 12) that could be potentiated greatly by pre-phosphorylation of Bora[1–224] by CyclinA2-Cdk2 (lanes 1, 6, and 11), consistent with a previous report[25]. Deletion of the PBD of Plk1 (ΔPBD) (lanes 6–10), or both the PBD and interdomain linker (i.e. the kinase domain-only construct (KD)) (lanes 11−15), had no effect on Plk1 phosphorylation by AURKA indicating that the critical determinants for efficient phosphorylation of Plk1 lie within the KD of Plk1. Likewise, a catalytically inactive mutant (K82R) of the Plk1 KD was also efficiently phosphorylated by AURKA, ruling out a role for auto priming by Plk1 in the process. Furthermore, by removing AURKA from the reaction, we excluded a direct role for Cyclin-Cdk in phosphorylating the T-loop of Plk1 (lanes 5, 10, 15). These results indicate that the KD of Plk1 is a minimal substrate for T-loop phosphorylation by AURKA in the presence of phosphorylated Bora.

### Bora is an intrinsically disordered protein that directly binds AURKA.
We next investigated the binding properties of Bora to AURKA and the Plk1 KD. To this end, we employed Heteronuclear Single Quantum Coherence (HSQC) Nuclear Magnetic Resonance (NMR) analysis with [15]N isotopically labeled Bora[1–224]. Initial HSQC analysis of non-phosphorylated [15]N-Bora[1–224] in isolation revealed a pattern indicative of an intrinsically disordered protein (Supplementary Fig. 2a, left panel). Specifically, all resonance peaks clustered between 8.5 and 7.5 ppm on the [1]H axis. Such a narrow dispersion reflects the similar

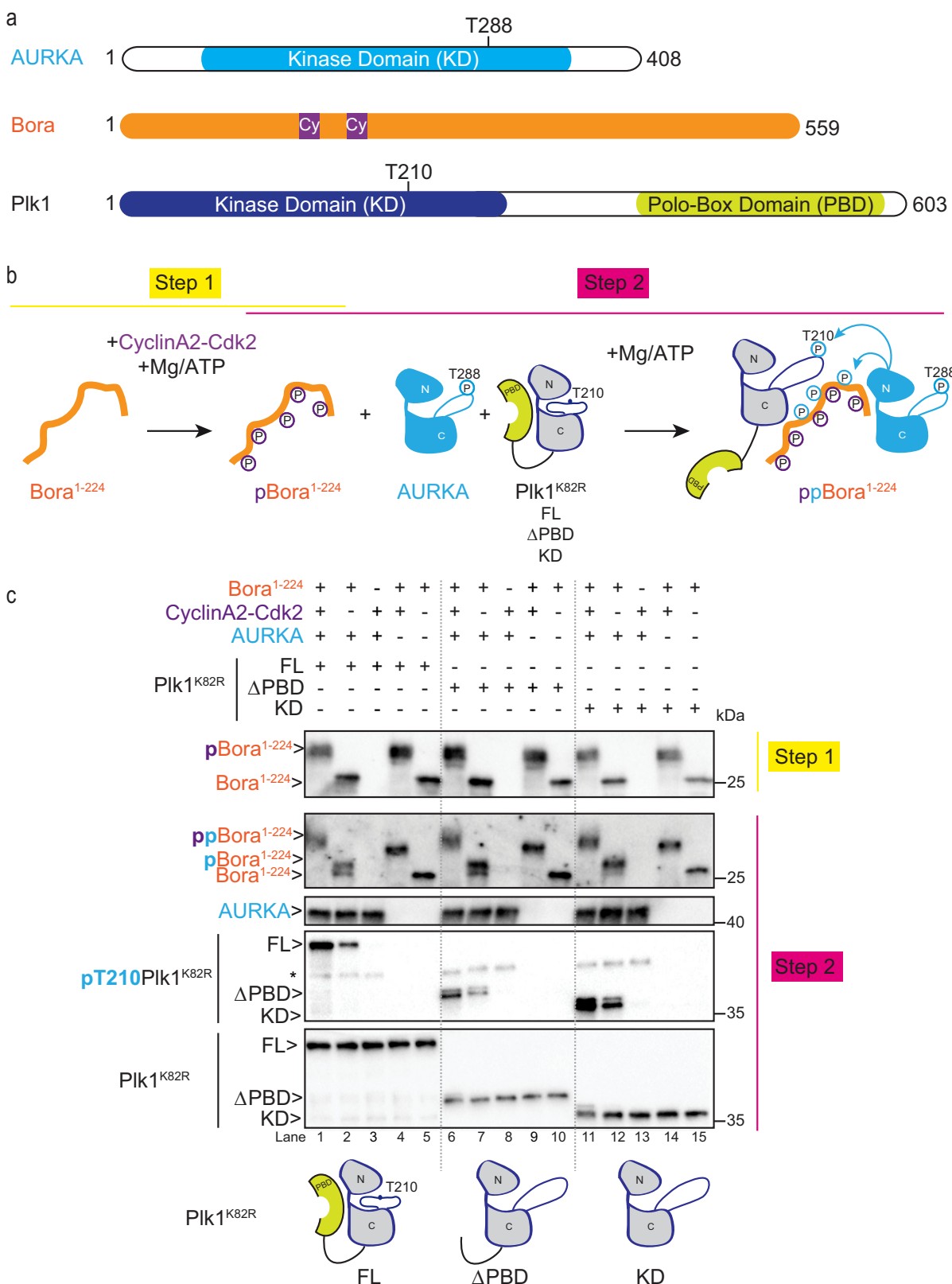

environment sampled by residues in a disordered state relative to the greater diversity of environments sampled by residues in a folded state. Phosphorylation of Bora[1–224] by Cyclin-Cdk on an average of 8 to 10 S/T-P sites (Supplementary Fig. 2a, right panel), led to the expected down-field shift of resonance peaks corresponding to phosphorylated amino acids. The majority of

resonance peaks remained clustered between 8.5 and 7.5 ppm on the [1]H axis suggesting that phosphorylation of Bora did not induce a transition to an ordered tertiary structure. Addition of unlabeled Plk1 KD to non-phosphorylated Bora[1–224] or to Cyclin-Cdk-phosphorylated Bora[1–224] revealed little peak shifting or line broadening (Fig. 2a, b, left panels respectively). In contrast,

**Fig. 1 In vitro reconstitution of a minimal system recapitulating T-loop phosphorylation of Plk1 by Bora and AURKA. a** Domain architecture of *Homo sapiens* AURKA, Bora, and Plk1. The positions of the T-loop phospho-regulatory sites of AURKA (T288) and Plk1 (T210) are indicated. Cy motifs in Bora correspond to Cyclin-binding motifs that mediate substrate targeting. **b** Two-step in vitro reconstitution of T-loop phosphorylation on T210 of Plk1 (pT210) by AURKA and Bora$^{1-224}$. In step 1, Bora is phosphorylated by CyclinA2-Cdk2 kinase (yielding pBora). In step 2, phosphorylated Bora from the reaction mix in step 1 is incubated with AURKA and the catalytically dead mutant Plk1$^{K82R}$ in full-length (FL), polo-box domain deleted (ΔPBD), or kinase domain-only (KD) forms. During step 2, AURKA phosphorylates Plk1 (noted pT210 Plk1$^{K82R}$) and Bora or pBora (noted ppBora). **c** Western blot analysis of kinase reactions carried out with Bora$^{1-224}$ phosphorylated (+) or not (−) by CyclinA2-Cdk2 (step 1) in the presence of Plk1$^{K82R}$ (FL or ΔPBD or KD) and AURKA (step 2). Schematics of the Plk1 constructs used as substrates are shown at bottom. Blots were probed with antibodies to Bora, AURKA, and phosphoT210 Plk1 or pan Plk1 as indicated (from top to bottom). Note that pBora phosphorylated by AURKA displays slower migration on SDS-PAGE compared to pBora phosphorylated only by CyclinA2-Cdk2. In the blot performed with the anti-pT210 Plk1 antibody, the asterisk denotes cross reactivity with the pT288 residue of AURKA.

addition of unlabeled AURKA caused dramatic perturbation to resonance peak heights including peak disappearances in both conditions, strongly indicating direct binding (Fig. 2a, b, right panels).

Consistent with our NMR observations, functional AVI-tagged Bora$^{1-224}$ (Supplementary Fig. 2b, c) in both non-phosphorylated and CyclinA2-Cdk2-phosphorylated states robustly pulled-down AURKA (Fig. 2c, left panel) but not Plk1 KD (Fig. 2c, center panel). This finding is consistent with previous observations showing that Bora does not co-immunoprecipitate with the Plk1 KD in human cells[31]. Furthermore, addition of AURKA to the Plk1 KD binding reaction did not assist binding of Plk1 KD to Bora$^{1-224}$ (Fig. 2c, right panel).

To further examine binding between Bora$^{1-224}$ and AURKA, we employed Surface Plasmon Resonance (SPR). AURKA bound to biotinylated Bora$^{1-224}$ and biotinylated pBora$^{1-224}$ immobilized to a sensor chip with $K_d$ values of ~52 μM and ~18 μM, respectively (Fig. 2d, e and Supplementary Fig. 2d, e; see also Supplementary Table 1 for quantified values described in this manuscript). Together, our NMR, pull-down and SPR binding studies indicated that Bora$^{1-224}$ can directly bind to the KD of AURKA, and not appreciably to the KD of Plk1. Furthermore, the phosphorylation of Bora$^{1-224}$ by Cyclin-Cdk appeared to enhance its affinity for AURKA to some degree (greater than 2.5-fold). These observations support a model in which pBora$^{1-224}$ promotes the phosphorylation of the Plk1 T-loop in part by directly binding to AURKA.

**pBora can activate AURKA lacking T-loop phosphorylation.** Auto-phosphorylation on T-loop residue Thr288 (corresponding to T295 in Xenopus AURKA, Supplementary Fig. 1) is essential for the catalytic activation of AURKA[43,44]. Indeed, similar to the catalytically dead mutant AURKA$^{D274N}$, dephospho-AURKA$^{WT}$ and the AURKA$^{T288V}$ mutant, which all lack T-loop phosphorylation (Supplementary Fig. 3a), were devoid of catalytic activity toward Plk1 KD$^{K82R}$ even in the presence of Bora$^{1-224}$ (Fig. 3a, b—compare lane 2 with 5, 8, and 11). Likewise, in contrast to AURKA$^{WT}$, the AURKA$^{T288V}$ mutant could not phosphorylate Bora$^{1-224}$ (Fig. 3b, compare the electrophoretic shift of Bora$^{1-224}$ in lanes 2 and 5 in step 2). Strikingly though, in the presence of pBora$^{1-224}$, both AURKA$^{T288V}$ and dephospho-AURKA$^{WT}$, but not the catalytically dead mutant AURKA$^{D274N}$, displayed robust activity against Plk1 KD$^{K82R}$ (Fig. 3b, compare lanes 1, 4, 7, and 10). This potentiating activity of Bora on AURKA$^{T288V}$ was completely dependent on CyclinA2-Cdk2 activity as Bora$^{1-224}$ phosphorylated solely by AURKA (lane 6) did not stimulate AURKA activity toward Plk1 (Supplementary Fig. 3b). Thus, the presence of phosphates on Cyclin-Cdk S/T-P consensus sites in Bora appears to compensate for the absence of essential T288 T-loop phosphorylation in AURKA.

To determine if the potentiating effect of Bora$^{1-224}$ on AURKA was substrate specific, we tested the unrelated Kemptide substrate

by monitoring ATPase activity using the ADP-Glo assay, which quantifies the amount of ADP produced during the kinase reaction. AURKA$^{WT}$ in the presence (Fig. 3c) and absence of Kemptide (Supplementary Fig. 3c) displayed very low basal ATPase activity. Titration of either Bora or pBora potentiated the ATPase activity of AURKA$^{WT}$ multifold. The EC$_{50}$ value for potentiation was superior for pBora relative to Bora ($0.19 \pm 0.1$ μM vs $1.2 \pm 0.7$ μM, respectively) (Fig. 3c), consistent with the tighter binding affinity of pBora for AURKA discerned by SPR (Fig. 2d, e).

AURKA$^{T288V}$ displayed negligible ATPase activity alone (Supplementary Fig. 3d) or in the presence of Kemptide substrate (Fig. 3d) and titration of Bora had no discernable activating effect on AURKA$^{T288V}$. Strikingly, however, pBora greatly potentiated the ATPase activity of AURKA$^{T288V}$ in both conditions (+/− Kemptide substrate) ($0.04 \pm 0.001$ μM vs $0.02 \pm 0.003$ μM, respectively). As reflected by a lower EC$_{50}$ value, this potentiation was even superior to that achieved by pBora on the AURKA$^{WT}$ (Fig. 3c), suggesting enhanced binding affinity of pBora for AURKA$^{T288V}$. This inference was supported by SPR as pBora bound AURKA$^{T288V}$ more tightly compared to non-phosphorylated Bora ($K_d = 6$ vs $11$ μM, respectively) (Fig. 3e, f and Supplementary Fig. 3e, f). Thus, phosphate(s) on Bora enhance binding affinity for AURKA$^{WT}$ and AURKA$^{T288V}$ and contribute to a more pronounced ability to stimulate AURKA kinase/ATPase activity.

To corroborate these findings against another known AURKA substrate, we examined AURKA phosphorylation of Histone 3 on Ser10. Consistent with our findings using Plk1 KD and Kemptide as substrates, pBora$^{1-224}$ but not its non-phosphorylated form greatly potentiated AURKA$^{T288V}$ phosphorylation of Histone 3 (Supplementary Fig. 4a, lanes 4 and 5). Interestingly, however, while pBora$^{1-224}$ was also able to promote AURKA$^{WT}$ phosphorylation of Histone 3 (to a small degree), non-phosphorylated Bora$^{1-224}$ was even more effective (Supplementary Fig. 4a, lanes 1 and 2). This result suggests that in some contexts, the presence of phosphate on both Bora and AURKA can lead to antagonistic interactions.

As the phosphomimetic AURKA$^{T288D}$ variant is often used as a proxy for the constitutively active enzyme, we decided to test whether this variant behaved more similar to wild-type or its AURKA$^{T288V}$ counterpart in response to Bora. Strikingly, the T-loop phospho-mimic mutant AURKA$^{T288D}$ behaved more similar to AURKA$^{T288V}$ with no ability to phosphorylate Plk1 on its T-loop alone or in the presence of Bora$^{1-224}$ but a strong ability to phosphorylate Plk1 in the presence of pBora$^{1-224}$ (Supplementary Fig. 4b). Thus, the AURKA$^{T288D}$ variant is not a reliable proxy for the activated form of AURKA achieved by pBora binding. We conclude that pBora is a potent activator of AURKA catalytic activity against multiple substrates, particularly when AURKA is not phosphorylated on its activation segment at T288 as exemplified by the T288V and T288D AURKA mutants.

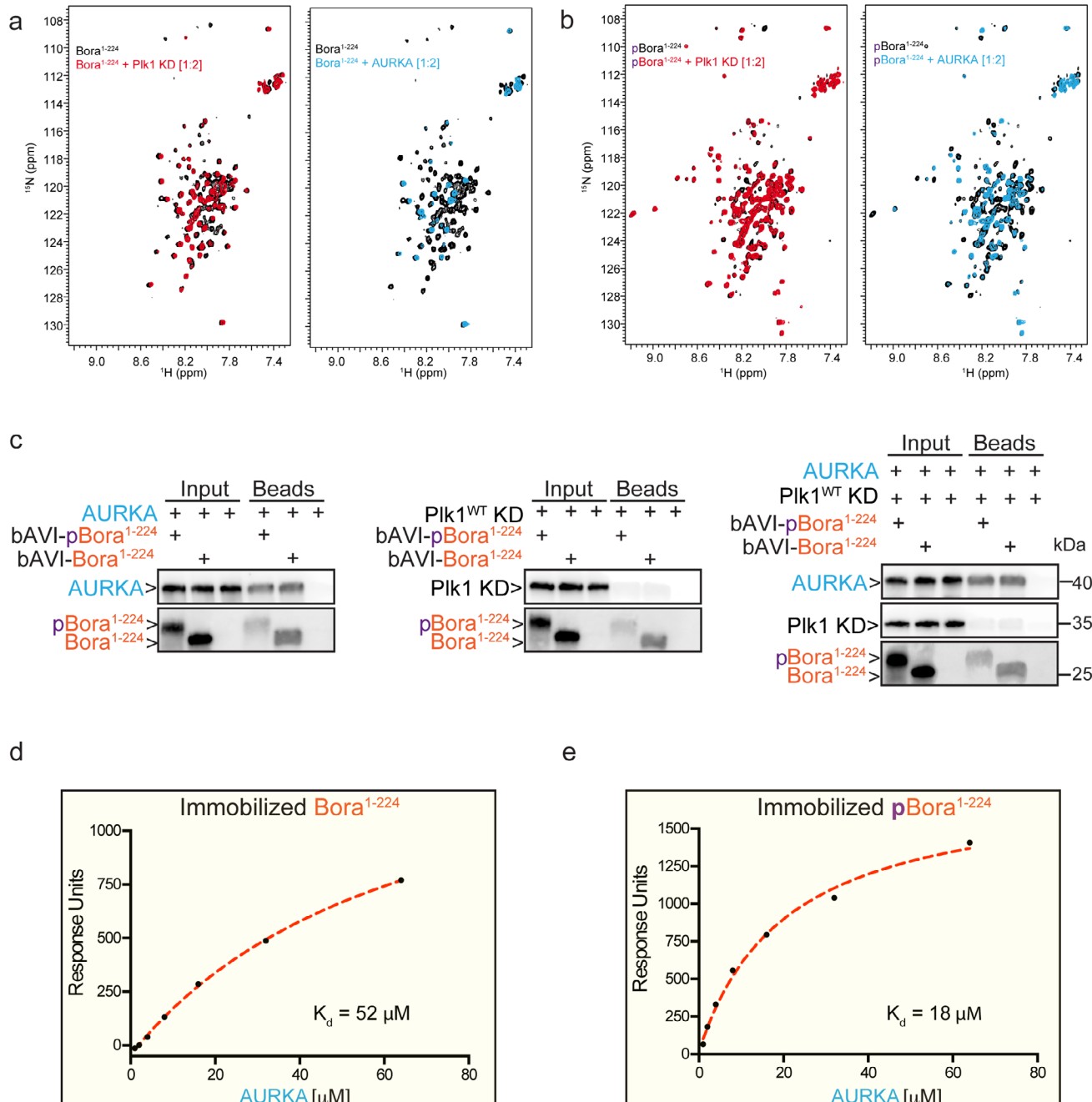

**Fig. 2 Bora is an intrinsically disordered protein that binds AURKA. a** NMR analysis of the binding interaction between non-phosphorylated $^{15}$N-labeled Bora ($^{15}$N-Bora) and unlabeled AURKA and Plk1. On the left, superimposed $^{1}$H-$^{15}$N HSQC spectra of $^{15}$N-Bora alone in black with $^{15}$N-Bora in red in the presence of two moles equivalents [1:2] of Plk1 KD. On the right, $^{15}$N-Bora alone in black with $^{15}$N-Bora in blue in the presence of two moles equivalents [1:2] of AURKA. **b** NMR analysis of the binding interaction between CyclinA2-Cdk2 phosphorylated $^{15}$N-Bora and unlabeled AURKA and Plk1. On the left, superimposed $^{1}$H-$^{15}$N HSQC spectra of phosphorylated $^{15}$N-Bora alone in black with phosphorylated $^{15}$N-Bora in red in the presence of two moles equivalents [1:2] of Plk1 KD. On the right, phosphorylated $^{15}$N-Bora alone in black with phosphorylated $^{15}$N-Bora in blue in the presence of two moles equivalents [1:2] of AURKA. **c** Pull-down experiments between immobilized biotinylated AVI-tagged Bora$^{1-224}$ (bAVI-Bora$^{1-224}$) or biotinylated AVI-tagged pBora$^{1-224}$ and AURKA (left panel), Plk1 KD (middle panel), or a mix of AURKA and Plk1 KD (right panel). **d, e** SPR binding analysis of immobilized Bora$^{1-224}$ (**d**) and pBora$^{1-224}$ (**e**) to AURKA$^{WT}$. $K_d$ values represent mean values, $n = 2$. Representative profiles shown are from one experiment. See Supplementary Fig. 2d, e, respectively, for sensogram traces.

**AURKA$^{T288V}$ is functional in the AURKA–Bora–Plk1 pathway in vivo**. The surprising observation that Cyclin-Cdk-phosphorylated Bora can activate AURKA even in the absence of T288 phosphorylation, led us to investigate whether pBora can similarly activate the AURKA$^{T288V}$ mutant in vivo. To this end, we used Xenopus egg extracts prepared from oocytes arrested in metaphase of meiosis II

(CSF: cytostatic factor arrested-extracts). CSF extracts are typically arrested in a mitotic state with high CyclinB-Cdk1 (MPF) activity. At this stage, the Greatwall kinase (Gwl) is active and maintains the mitotic state by inhibiting dephosphorylation of key cell cycle regulatory factors by the protein phosphatase 2A PP2A-B55 (complexed with its B55 regulatory subunit)[45] (Fig. 4a). However, Gwl

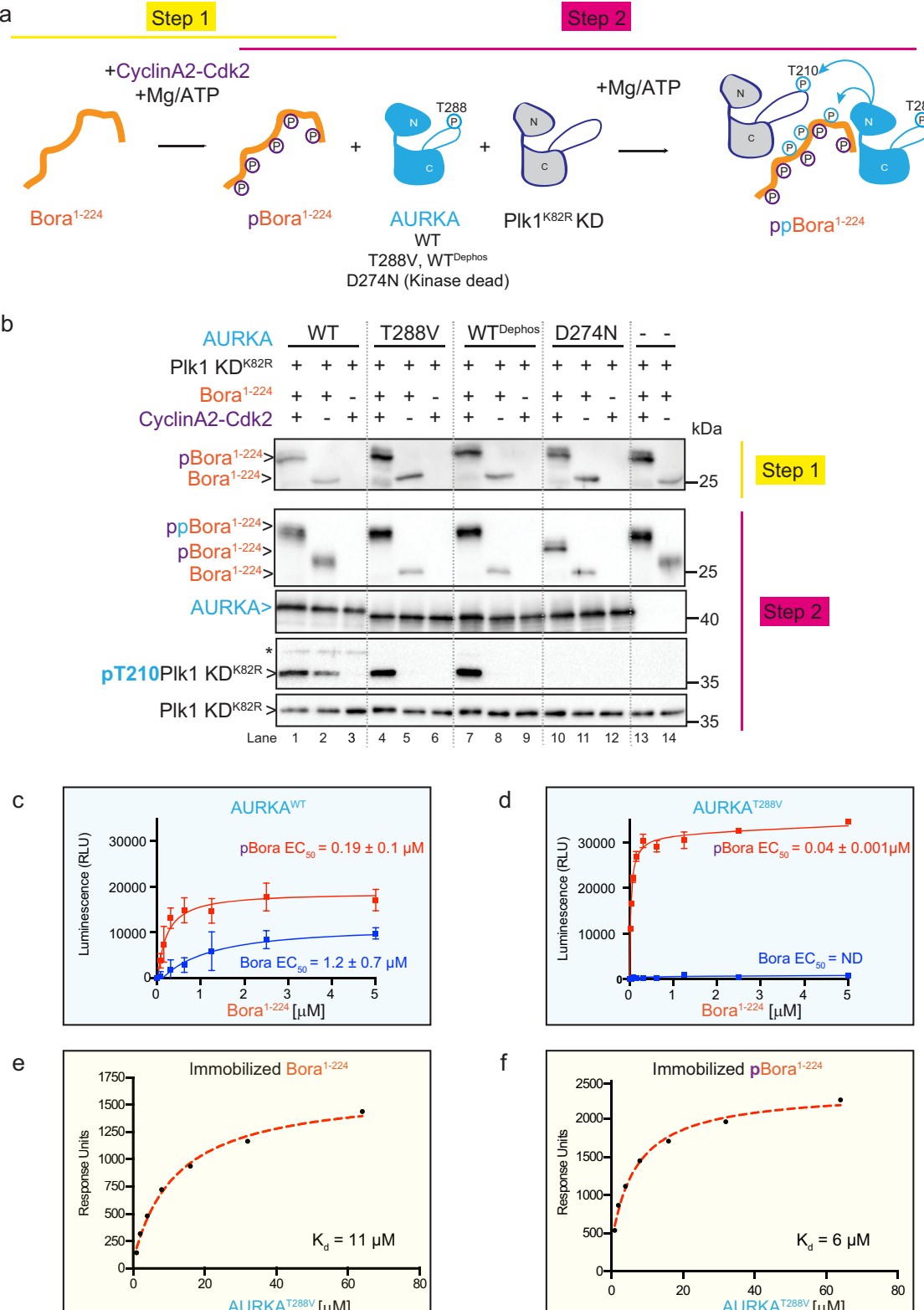

immunodepletion releases active PP2A-B55, which can then dephosphorylate CyclinB-Cdk1 substrates and consequently cause exit from the mitotic state[45] (Fig. 4a, b). Complementation of the extracts with hyperactive recombinant Gwl[K72M] can restore the mitotic state if the extracts contain an active AURKA, Bora, and Plk1 pathway (Fig. 4a, b).

Taking advantage of this system, we tested whether human (hs) AURKA[T288V] can replace Xenopus (Xe) AURKA function in CSF extracts. To this end, endogenous Gwl and AURKA were sequentially immunodepleted, and then the extract was supplemented with recombinant hyperactive Gwl[K72M] and either hs AURKA[WT], AURKA[T288V], or the catalytically dead mutant

**Fig. 3 phospho-Bora activates dephospho-AURKA and its mimic AURKA$^{T288V}$. a** Two-step in vitro reconstitution of T-loop phosphorylation on T210 of Plk1 KD (pT210) by AURKA WT, AURKA T-loop mutant (T288V), dephosphorylated AURKA (WT$^{dephos}$), or catalytically dead AURKA$^{D274N}$ mutant and the activator Bora$^{1-224}$. In step 1, Bora is phosphorylated (+) or not (−) by CyclinA2-Cdk2 kinase. In step 2, phosphorylated Bora$^{1-224}$ from the reaction mix in step 1 is incubated with AURKA (in the indicated forms) and the kinase domain KD of catalytically dead Plk1$^{K82R}$. **b** Western blot analysis of kinase reactions carried out with Bora$^{1-224}$ phosphorylated (+) or not (−) by CyclinA2-Cdk2 (step 1) in the presence of Plk1$^{K82R}$ KD and the indicated forms of AURKA, namely WT phosphorylated, WT dephosphorylated, T288V mutant, or D274N mutant (step 2). Blots were probed with antibodies to Bora, AURKA, and phosphoT210 Plk1 or pan Plk1 as indicated (from top to bottom). In the blot performed with the anti-pT210 Plk1 antibody, the asterisk denotes the cross reactivity with the pT288 residue of AURKA$^{WT}$. **c, d** Activation of AURKA$^{WT}$ (**c**) or AURKA$^{T288V}$ (**d**) kinase activity by Bora$^{1-224}$ and pBora$^{1-224}$ as assessed using the ADP-Glo assay and Kemptide substrate. Displayed data points and EC$_{50}$ values represent the average luminescence for each reaction condition with standard deviations of the mean as error bars ($n = 3$ independent experiment samples). RLU: relative light unit. ND: not determined. Reaction results carried out in the absence of Kemptide are shown in Supplementary Fig. 3c, d. **e, f** SPR binding analysis of immobilized Bora$^{1-224}$ (**e**) and pBora$^{1-224}$ (**f**) to AURKA$^{T288V}$. $K_d$ values represent mean values, $n = 2$. Representative profiles shown are from one experiment. See Supplementary Fig. 3e, f, respectively, for sensogram traces.

AURKA$^{D274N}$ (Fig. 4c). As shown in Fig. 4d (lane 3), CSF-arrested extracts exited the mitotic state upon Gwl immunodepletion, as reflected by the dephosphorylated state of the mitotic substrates Cdc25 and the protein phosphatase 1 (PP1) and the accumulation of inactive Cdk phosphorylated on the inhibitory site tyrosine 15 (pTyr). Similarly, Plx1 (Xenopus Plk1) was also rapidly dephosphorylated on its T-loop site T201 (denoted pPlx1) upon Gwl immunodepletion (ΔGwl) (Fig. 4d, lanes 3, 9, and 17). Complementation of the extract with recombinant Gwl$^{K72M}$ and AURKA$^{WT}$ (lanes 3–8), but not the catalytically dead AURKA$^{D274N}$ mutant (lanes 17–22), within minutes restored the mitotic state. Active Plx1 phosphorylated on its T-loop (pPlx1) accumulated 60–90 min after complementation of the extracts with recombinant Gwl$^{K72M}$ and AURKA$^{WT}$. This was accompanied by the accumulation of phosphorylated mitotic substrates Cdc25 and PP1 and the dephosphorylation of Cdk on tyrosine 15 (Fig. 4d, lanes 7 and 8). Remarkably, the down-regulated mutant AURKA$^{T288V}$, similar to AURKA$^{WT}$, promoted Plx1 T-loop phosphorylation and high Cyclin-Cdk activity in these extracts with comparable kinetics (Fig. 4d, lanes 9–14). Taken together, these results indicate that human AURKA$^{T288V}$ can support AURKA function in Xenopus oocyte extracts. We conclude that in this context, and similarly to the situation in vitro, pBora can activate AURKA independently of T-loop phosphorylation on T288 and can thus compensate for the lack of T-loop phosphorylation on AURKA.

**A minimal fragment of Bora encompassing residues 18–120 supports AURKA-dependent phosphorylation of Plk1 in vitro.** To identify the minimal region of Bora that can recapitulate Cyclin-Cdk-dependent activation of AURKA against Plk1, we performed a deletion analysis on the minimally active Bora$^{1-224}$ fragment (see Fig. 5a for schematic). Deletion of Cy motifs (Fig. 5a) in Bora adversely affected our ability to phosphorylate Bora with Cyclin-Cdk[25] (see Supplementary Fig. 5a for example). Therefore, we employed the proline-directed MAP kinase ERK in place of Cyclin-Cdk in our pre-phosphorylation reactions. Validating this approach, Bora phosphorylated by ERK and Bora phosphorylated by CyclinA2-Cdk2 both efficiently stimulated Plk1 T210 phosphorylation in our reconstitution assay (Supplementary Fig. 5b, compare lanes 1 and 6). Importantly, ERK itself was not able to directly phosphorylate Plk1 on the T-loop (Supplementary Fig. 5b, lane 5). Mass spectrometry analysis confirmed that ERK was able to phosphorylate all tested Bora fragments in vitro (Supplementary Fig. 6a–i).

We then tested the ability of the Bora deletion fragments to support Plk1 phosphorylation by AURKA$^{WT}$ or AURKA$^{T288V}$. pBora$^{1-224}$ supported the most efficient phosphorylation of Plk1 by both AURKA$^{WT}$ and AURKA$^{T288V}$ (Fig. 5b, lanes 1–4). N- and C-terminal deletions to Bora$^{1-224}$ in the form of the

Bora$^{35-157}$ fragment abolished all stimulatory function (lanes 5–8). Given the N-terminal part of Bora is highly conserved[25], we extended the N-terminal part of Bora$^{35-157}$ to recover activity. Whereas the Bora$^{35-157}$ fragment was totally inactive, Bora$^{18-157}$ was able to stimulate AURKA$^{WT}$ and AURKA$^{T288V}$ function similar to Bora$^{1-224}$ (lanes 9–12). Likewise, Bora$^{18-120}$ was able to stimulate AURKA$^{WT}$ and AURKA$^{T288V}$ (lanes 13–16). Collectively, these results indicate that the Bora$^{18-120}$ fragment is sufficient to stimulate the catalytic function of AURKA toward Plk1.

As phosphorylation of Bora$^{18-120}$ by a proline-directed kinase contributed greatly to its ability to stimulate the catalytic activity of AURKA (especially the AURKA$^{T288V}$ mutant), we used site-directed mutagenesis to probe which specific phospho-sites in Bora were responsible (Fig. 5b, right panel). Bora$^{18-120}$ contains four S/T-P sites, S27, S41, T52, and S112 that match the Cyclin-Cdk consensus sequence. Mutation of S27 and T52 together had no impact on the AURKA stimulating activity of Bora$^{18-120}$ (lanes 17–24) but the addition of a third mutation to either S41 (lanes 25–28) or S112 (lanes 29–32) impaired its function to some degree. Specifically, the triple mutant Bora$^{18-120}$ $^{[S27A\ S41A\ T52A]}$, phosphorylated or not by ERK, lost all ability to activate AURKA$^{WT}$ but retained the ability to activate AURKA$^{T288V}$ when pre-phosphorylated by ERK (i.e. phosphorylated on S112 only) (lane 27). In contrast, the triple mutant Bora$^{18-120}$ $^{[S27A\ T52A\ S112A]}$, phosphorylated or not by ERK (on S41 only), not only lost all ability to activate AURKA$^{WT}$ but also AURKA$^{T288V}$ (lanes 29–32). To corroborate the greater importance of the S112 phosphoregulatory site, we analyzed both Bora$^{18-120}$ $^{[S41A]}$ and Bora$^{18-120}$ $^{[S112A]}$ single site mutants. Consistent with expectations, while Bora$^{18-120}$ $^{[S41A]}$ lost the ability to activate AURKA$^{WT}$, it still retained the ability to activate AURKA$^{T288V}$ when pre-phosphorylated by ERK. In contrast Bora$^{18-120}$ $^{[S112A]}$, phosphorylated or not by ERK, lost the ability to activate both AURKA$^{WT}$ and AURKA$^{T288V}$ (Supplementary Fig. S5c). Together, these results indicate that Bora$^{18-120}$, phosphorylated on the Cyclin-Cdk site S112, is minimally sufficient to potentiate the catalytic activity of AURKA$^{T288V}$ toward Plk1. These results also reveal that Bora phosphorylation on S112 can compensate for the absence of T-loop phosphorylation on AURKA. Since mutation of Ser41 adversely affected the ability of both Bora and pBora to activate AURKA, we reason that this residue may play an important structural and non-phosphorylation-dependent role in supporting the Bora–AURKA interaction.

**Bora contains two short Tpx2-like motifs required for AURKA activation.** The microtubule-associated protein Tpx2 is a well-characterized allosteric activator of AURKA. The crystal structure of a Tpx2/AURKA complex (PDB: 1OL5) revealed how Tpx2

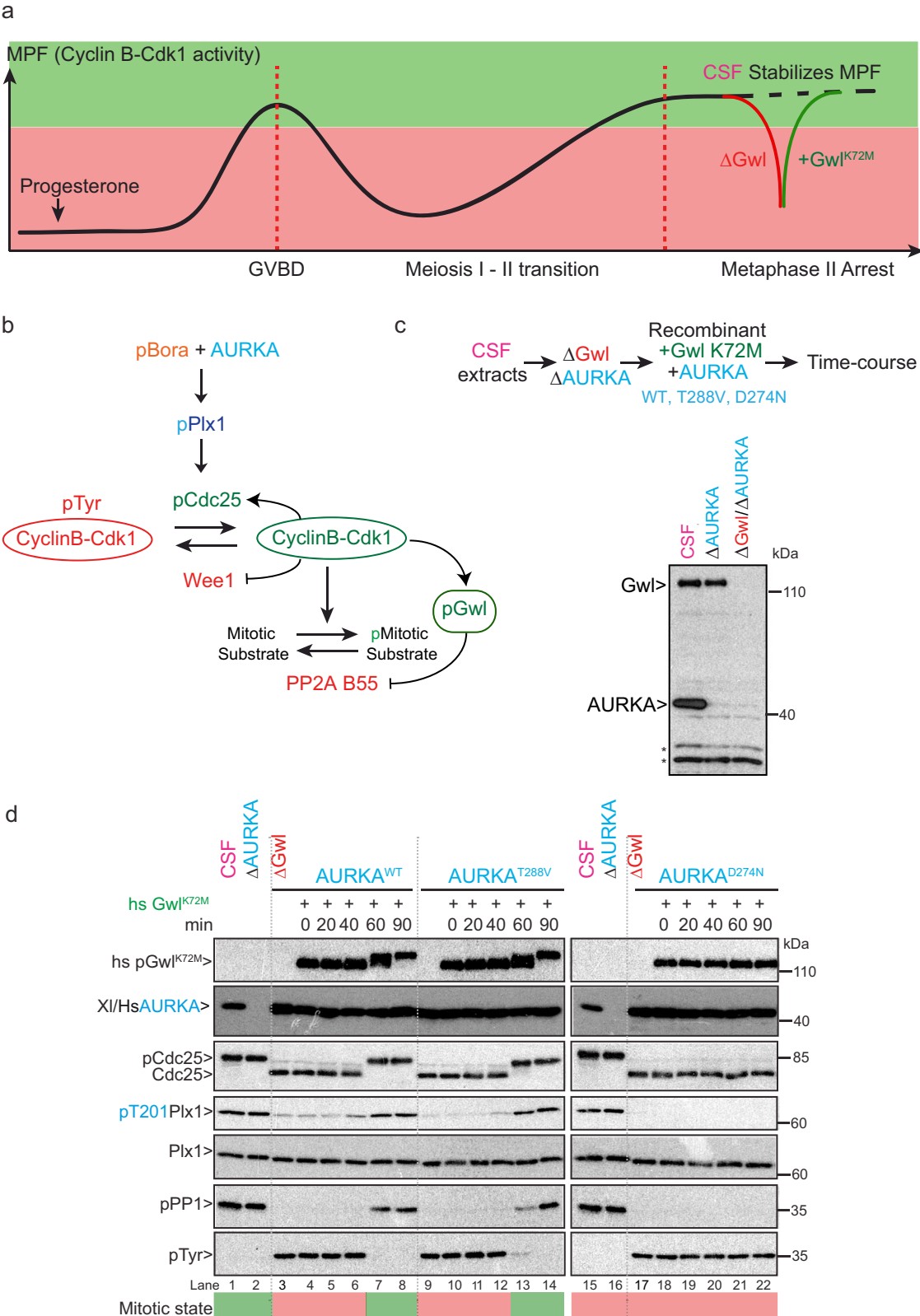

uses its first 43 amino acids to bind the catalytic domain of AURKA[40,41,46]. Sequence comparison of Bora[18–120] with the first 43 residues of Tpx2 revealed two regions of similarity, denoted motif 1 (M1; residues 25–34 in Bora) and motif 2 (M2; residues 101–110 in Bora) (Fig. 5c). Motif 1 of Tpx2 (residues 7–21) adopts an extended conformation that binds the N-terminal lobe

of AURKA while motif 2 (residues 30–43) adopts an alpha helical conformation that binds between the N- and C-terminal lobes, adjacent to helix αC and the activation segment[40].

Within the two Tpx2 motifs, Tyr8, Tyr10, Phe16, Phe19, Trp34, and Phe35 are crucial for binding to AURKA[47–49]. Remarkably, Tyr10, Phe16, Phe19, Trp34, and Phe35 in Tpx2 are

**Fig. 4 AURKA$^{T288V}$ supports AURKA function in maintaining the mitotic state in CSF-arrested Xenopus oocytes. a** In the ovary, oocytes are arrested in prophase of meiosis I. At the time of ovulation and upon hormone stimulation (progesterone), oocytes re-enter into meiosis and reach the second meiotic division where they arrest in metaphase II, awaiting fertilization (CSF arrest). Upon Gwl immunodepletion (ΔGwl in red), the extracts exit the "mitotic state". Complementation of the extract with recombinant hyperactive Gwl$^{K72M}$ kinase (+Gwl$^{K72M}$ in green) restores the mitotic state, if the extracts contain a functional Bora, AURKA, Plk1 pathway. GVBD: germinal vesicle breakdown. The green and red area represents the high and low CyclinB-Cdk1 (MPF) activity. **b** Network representing the pathways controlling Cyclin-Cdk activity and maintaining the "mitotic state" in CSF-arrested Xenopus egg extracts. **c** Flow chart of the experimental approach used to test the functionality of AURKA mutants in maintaining the "mitotic state" in CSF-arrested Xenopus oocytes extracts. Endogenous AURKA and Gwl were sequentially immunodepleted from CSF egg extracts and then complemented with recombinant human AURKA$^{WT}$, catalytically dead AURKA$^{D274N}$, or downregulated mimic AURKA$^{T288V}$ and hyperactive Gwl$^{K72M}$ kinase (K72M) (top panel). CSF extracts non-depleted or sequentially depleted of AURKA (ΔAURKA) and then Greatwall (ΔGwl) were separated by SDS-PAGE and analyzed by Western blot using Xenopus Gwl and AURKA antibodies. Asterisks denote non-specific bands. **d** CSF egg extracts sequentially depleted of AURKA (ΔAURKA) and then Gwl (ΔGwl) were supplemented at time 0 with recombinant AURKA$^{WT}$, or catalytically dead AURKA$^{D274N}$ or AURKA$^{T288V}$ in the presence of hyperactive Gwl$^{K72M}$. A fraction of the extracts was collected at the indicated time-points (0, 20, 40, 60, 90 min) and analyzed by Western blot using specific antibodies to monitor the levels of huGreatwall, Hu/Xe AURKA, Plk1, Cdc25, as well as the phosphorylation of Plx1 on T201 (pPlk1), PP1 phosphatase on T320 (pPP1), and Cdk on Y15 (pTyr). The green and red area represents the high and low CyclinB-Cdk1 (MPF) activity ("mitotic state").

also conserved in two analogous motifs in Bora albeit with greater motif separation (Fig. 5c). Furthermore, the general pattern of hydrophobicity and polarity across the two motifs appear conserved suggesting that Bora might engage AURKA in a similar fashion to Tpx2.

To test this hypothesis, we mutagenized Bora$^{18–120}$ (harboring the benign S27A/T52A double mutation) by substituting key hydrophobic residues with negatively charged residues in M1 (F25D/Y31D) and in M2 (F103D/F104D). Testing of each construct in our in vitro reactions revealed that mutations in M1 or M2 abolished the stimulatory activity of Bora$^{18–120}$ toward AURKA (Fig. 5d, lanes 5–12). Together, these results uncovered three essential determinants in Bora for promoting AURKA phosphorylation of Plk1, namely two Tpx2-like motifs M1 and M2 in addition to a Cyclin-Cdk phospho-regulatory site at Ser112 (denoted M3).

**Bora and Tpx2 compete for AURKA binding and display different abilities to activate AURKA.** The finding that Bora and Tpx2 share similar motifs for binding to AURKA suggested that the two proteins may bind to AURKA in a competitive manner. To test this hypothesis, we established a competitive displacement binding assay using fluorescein-labeled Tpx2$^{1–43}$ (FITC-Tpx2$^{1–43}$) as a probe (see Fig. 6a for schematic). As assessed by fluorescence polarization measurement, FITC-Tpx2$^{1–43}$ bound to AURKA$^{WT}$ and AURKA$^{T288V}$ with $K_d$ values of 0.009 μM and 0.015 μM, respectively (Fig. 6b, d). Supporting the notion of similar binding modes, Tpx2$^{1–43}$, Bora$^{1–224}$, and pBora$^{1–224}$ could all competitively displace FITC-Tpx2$^{1–43}$ from binding to AURKA$^{WT}$ (Fig. 6c) and to AURKA$^{T288V}$ (Fig. 6e). Notably, Tpx2$^{1–43}$ appeared more efficient than pBora$^{1–224}$ and Bora$^{1–224}$ at competitive binding to AURKA$^{WT}$ (IC$_{50}$ = 0.1 μM, 2.7 μM, and ND, respectively), whereas pBora$^{1–224}$ appeared more efficient than Tpx2$^{1–43}$ or Bora$^{1–224}$ at competitive binding to AURKA$^{T288V}$ (IC$_{50}$ = 0.032, 0.11, and 0.73 μM, respectively). We note that while the general trends of binding preference observed by fluorescence polarization (Fig. 6) and SPR (Figs. 2 and 3) are similar, the binding affinities appear under-estimated in the SPR experiment possibly due to Bora immobilization via its N-terminus.

Consistent with expectation, the shorter pBora$^{18–120}$ construct behaved similar to pBora$^{1–224}$ in its ability to displace FITC-Tpx2$^{1–43}$ from AURKA$^{T288V}$, albeit with ≈20-fold less efficiency (Fig. 6f). Introduction of aspartic acid substitutions in the hydrophobic M1 and M2 motifs drastically reduced Tpx2$^{1–43}$ displacement function (Fig. 6f), consistent with our findings that these mutations abrogate the ability of pBora$^{18–120}$ to activate AURKA$^{T288V}$ phosphorylation of Plk1 (Fig. 5d). Together, these

results confirm the prediction that Bora and Tpx2 share similar binding features to AURKA. Furthermore, they demonstrate an optimal binding interaction arises when Bora is phosphorylated and when AURKA is not phosphorylated.

The different apparent binding affinities of Tpx2$^{1–43}$, Bora$^{1–224}$, and pBora$^{1–224}$ toward AURKA$^{WT}$ and AURKA$^{T288V}$ led us to compare the relative regulatory effects of these proteins on AURKA activity using the ADP-Glo assay. Consistent with the competitive binding studies, Tpx2$^{1–43}$ displayed superior activating potential relative to pBora$^{1–224}$ and Bora$^{1–224}$ on AURKA$^{WT}$ (Fig. 6g). Surprisingly, despite its ability to bind AURKA$^{T288V}$, Tpx2$^{1–43}$ (like Bora$^{1–224}$ and unlike pBora$^{1–224}$) displayed no ability to activate AURKA$^{T288V}$ (Fig. 6h). Furthermore, consistent with their ability to bind but not activate AURKA$^{T288V}$, both Tpx2$^{1–43}$ and Bora$^{1–224}$ acted as competitive inhibitors of pBora$^{1–224}$ activation of AURKA$^{T288V}$ (Fig. 6i). Taken at face value, these results demonstrate unequivocally that Bora and Tpx2 compete for binding to AURKA with affinities and activation capabilities dependent on the phosphorylation status of both Bora and AURKA. These results echo previous observations showing that Bora depletion in human cells increases the association of Tpx2 with AURKA on the mitotic spindle while overexpression of an N-terminal fragment of Bora (Bora N-ter$^{[1–379]}$) prevents co-immunoprecipitation of Tpx2 with AURKA[31].

**Mutational analysis of putative phospho-binding sites on AURKA.** Our in vivo and in vitro results indicate that pBora can compensate for the absence of T-loop phosphorylation in AURKA. This raises the question of how structurally this is achieved. One possibility is that the phosphoSer112 moiety of Bora binds the same site on AURKA that normally engages the phosphorylated T-loop residue T288 of AURKA in *cis*. This would involve coordination by Arg255, Arg180, and Arg286 (Fig. 7a and Supplementary Fig. 1). To our knowledge, no precedent for a kinase regulator acting in this manner has been described. An alternate possibility is that the phosphoSer112 motif binds to a different site on AURKA from the phosphorylated T-loop. For example, a Tpx2-AURKA co-structure (PDB: 1OL5) reveals an adjacent and partially overlapping binding site involving His176, Arg180, and Arg286 that coordinates a free sulfate ion (Fig. 7a). A precedent for this site in coordinating phosphate was established for AURKC, which is regulated by binding to the centromere and mitotic regulator INCENP[50]. One of two phospho-sites in INCENP (within a TSS motif) binds to the corresponding sulfate binding site conserved with AURKA (see Supplementary Fig. 7a, b). Of the four residues involved in the two adjacent binding sites, only Arg255 is unique to

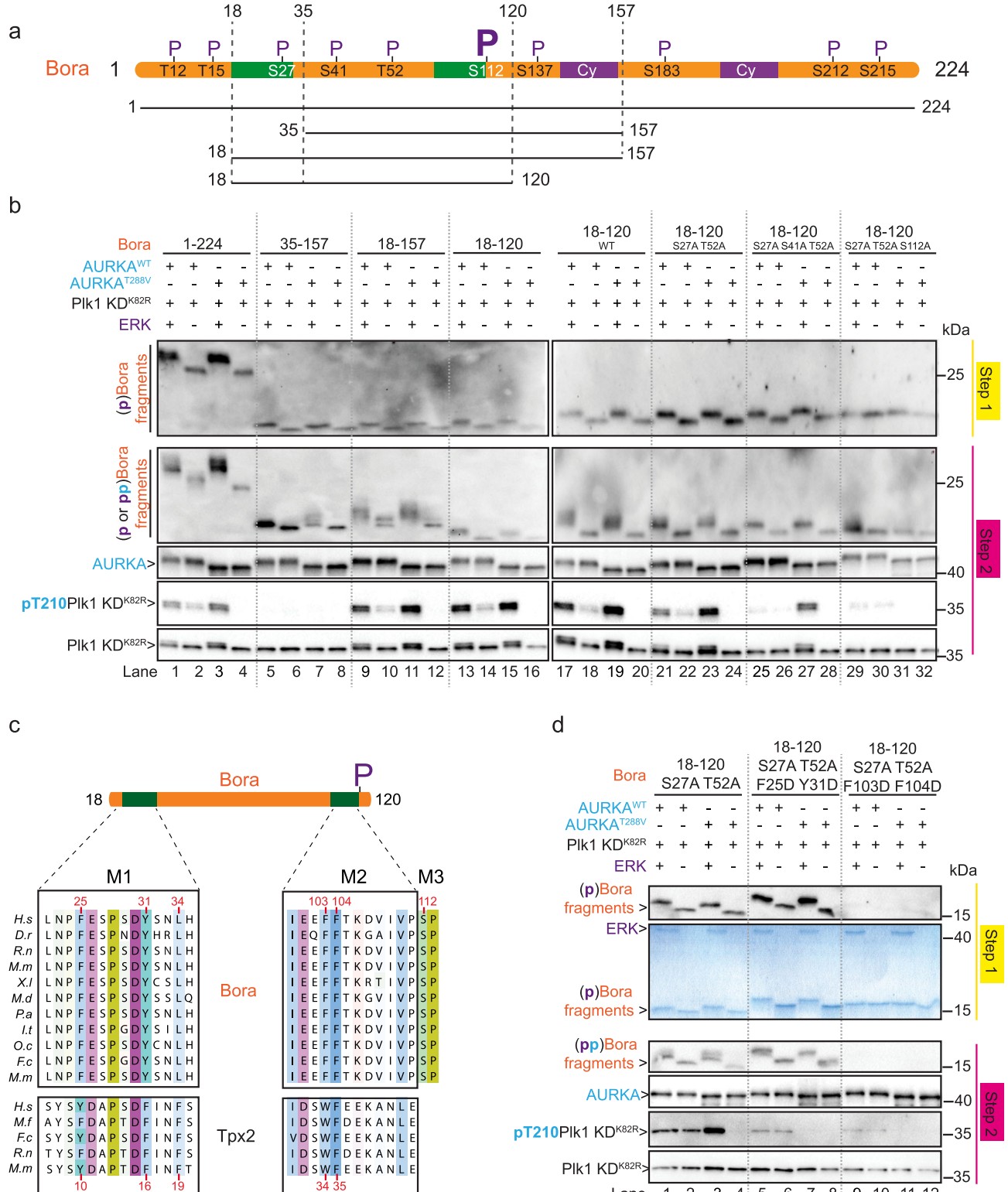

coordinating the phosphorylated T-loop residue T288 whereas only His176 is unique to coordinating the sulfate ion (Fig. 7a and Supplementary Fig. 1).

To probe the involvement of the two aforementioned AURKA sites in binding to a phosphate moiety in Bora, we individually substituted residues His176, Arg180, Arg255, and Arg286 for alanine in AURKA[T288V] and tested for responsiveness to pBora[18–120] in protein kinase assays. All four AURKA mutants

(AURKA[T288V H176A], AURKA[T288V R180A], AURKA[T288V R255A], and AURKA[T288V R286A]), which retained abilities to bind Tpx2[1–43] (ref. [47] and Fig. 7b), displayed reduced responsiveness to pBora[18–120] (Fig. 7c). These observations supported a role for AURKA residues His176, Arg180, Arg255, and Arg286 in coordinating a phosphate moiety in Bora. To examine this issue further, we tested the ability of the four AURKA mutants to bind pBora[1–224]. Taking advantage that all four AURKA mutants bind

**Fig. 5 Bora contains Tpx2-like motifs and a unique phosphosite required for its activation function on AURKA in vitro. a** Schematic of Bora[1–224] highlighting ten S/T-P consensus sites that are phosphorylatable by the proline-directed kinases Cyclin-Cdk or ERK. Residue position of the phospho-sites are indicated and highlighted by the letter P (above). Also highlighted are two Cyclin-binding motifs (Cy, violet) and two Tpx2-like motifs 1 and 2 (green). The boundaries of the different Bora fragments analyzed in **b–d**, are indicated. The most essential phospho-site S112 is highlighted by a large P. **b** Western blot analysis of 2-step kinase reactions carried out with the indicated Bora fragments phosphorylated (+) or not (−) by the ERK kinase (step 1) in the presence of Plk1[K82R] KD substrate and AURKA[WT] or AURKA[T288V] (step 2). Blots were probed with antibodies to Bora, AURKA, and phosphoT210 Plk1 or pan Plk1 as indicated (from top to bottom). **c** Alignments of Bora motifs M1 and M2 with Tpx2 motifs responsible for binding to AURKA. Essential residue numbers in Tpx2 responsible for binding AURKA are highlighted in red below the alignments. The corresponding residue numbers in Bora are highlighted in red above the alignments. Also shown is a phospho-motif M3 in Bora not conserved in Tpx2. **d** Western blot analysis of kinase reactions carried out with the indicated Bora[18–120] mutants (S27A and T52A plus/minus mutations within motifs M1 or M2) phosphorylated (+) or not (−) by the ERK kinase (step 1) in the presence of Plk1[K82R] KD and AURKA[WT] or AURKA[T288V] (step 2). Blots were probed with antibodies to Bora, AURKA, and phosphoT210 Plk1 or pan Plk1 as indicated (from top to bottom). Note that the Bora[18–120 [S27A T52A F103D F104D]] mutant was not recognized by our Bora antibody but Coomassie staining of the step1 reaction (bottom panel of step 1) revealed the presence of the protein at the expected size.

---

with similar affinity to FITC-Tpx2[1–43] (Fig. 7b, with the exception of 3-fold reduced binding for AURKA[T288V R255A]), we measured competitive displacement of FITC-Tpx2[1–43] from each AURKA mutant by pBora[1–224]. Whereas pBora[1–224] competitively displaced Tpx2[1–43] from AURKA[T288V] with an $IC_{50}$ of 35 nM, it was 8-, 20-, 22-, and 15-fold impaired for displacing Tpx2[1–43] from the AURKA[T288V H176A], AURKA[T288V R180A], AURKA[T288V R255A], and AURKA[T288V R286A] mutants ($IC_{50} = 290$, 720, 771, and 516 nM), respectively (Fig. 7d). Taken together, these data provide compelling evidence that His176, Arg180, Arg286, and Arg255 participate in coordinating the phospho-S112 moiety in Bora that is required for the robust activation of AURKA[T288V]. The results are further consistent with a model in which a phospho-S112 moiety in Bora functionally and perhaps structurally substitutes for the phospho-T288 moiety on the T-loop that is normally required to activate AURKA (Fig. 7e).

**Bora determinants are critical for mitotic entry in Xenopus egg extracts.** Having determined that the Tpx2-like motifs (M1 and M2) and the phosphorylated serine Ser112 (M3) are essential in the context of the minimal Bora[18–120] fragment for function in vitro, we next investigated the relevance of these determinants for Bora function in vivo. We first investigated Bora function at mitotic entry in Xenopus egg extracts using a variation of the experimental setup described earlier, whereby interphase egg extracts are forced to enter into mitosis using the Gwl kinase[27,45]. Addition of a recombinant hyperactive Gwl kinase (Gwl[K72M]) to interphase extracts partially inhibits PP2A-B55, stabilizing substrate phosphorylation by basal Cyclin-Cdk activity, which forces entry into mitosis. However, prior immunodepletion of Bora from these extracts prevents mitotic entry. Under these conditions, Plx1 activation by AURKA and the removal of inhibitory phosphorylation on Cdk (specifically on Tyr15) are inhibited. This mitotic entry defect can be fully rescued by addition of wild-type recombinant Bora[27]. Hence, we tested whether the minimal Bora fragments that support Plk1 phosphorylation by AURKA in vitro could promote mitotic entry in interphase extracts depleted of Bora (Fig. 8a). We again made use of ERK to pre-phosphorylate Bora fragments on S/T-P sites, as we did in our in vitro reactions, to circumvent inefficient phosphorylation of some Bora deletion constructs by CyclinA-Cdk1 in Xenopus egg extracts.

Consistent with our previous observations, Bora[1–224] pre-phosphorylated with ERK fully rescued mitotic entry in interphase extract depleted of Bora[27]. As shown in Fig. 8b (lanes 2–5), addition of hyperactive Gwl[K72M] to the depleted lysate (lane 1) resulted in an abrupt increase of Plx1 phosphorylation by 20 min that was followed by entry into mitosis as evidenced by the gradual

loss of inhibitory Tyr15 phosphorylation on CyclinB-Cdk1 and the inhibitory phosphorylation on the PP1 phosphatase. In contrast, Bora[35–157] lacking M1 demonstrated no ability to rescue the mitotic entry defect (despite efficient phosphorylation by ERK, Supplementary Fig. S5) as evidenced by the maintenance of inhibitory Tyr15 phosphorylation on CyclinB-Cdk1 and the absence of PP1 phosphorylation (Fig. 8b, lanes 6–9). Like Bora[1–224], both Bora[18–157] and Bora[18–120] constructs, harboring S27A and T52A mutations at two Cyclin-Cdk phospho-acceptor sites, were fully competent to support mitotic entry of interphase extracts depleted of Bora. In both cases, addition of Gwl[K72M] was accompanied by activation of Plx1 and entry into mitosis as evidenced by the phosphorylation of PP1 and dephosphorylation of Cdk1 on Tyr15 (Fig. 8b, lanes 15–18 and lanes 19–22).

We next tested whether the Tpx2-like motifs M1 and M2 of Bora were similarly required for Bora's function in mitotic entry. As shown in the Fig. 8c, Bora[18–120] harboring the benign S27A and T52A mutations together with mutations in either of the motifs M1 (F25D Y31D) or M2 (F103D F104D) failed to support mitotic entry in Xenopus egg extracts. In both cases, Plx1 was not phosphorylated/activated by addition of active Gwl[K72M] and the extracts did not reach the mitotic state as evidence by the maintenance of inhibitory Tyr15 phosphorylation on Cdk1 and the lack of PP1 phosphorylation (Fig. 8c). Taken together, these results validate our in vitro observations that Bora[18–120] is minimally sufficient for stimulating the phosphorylation of Plk1 by AURKA and for promoting mitotic entry. These observations further indicate that the two Tpx2-like motifs M1 and M2, together with pre-phosphorylation of Cyclin-Cdk regulatory sites, are critical for Bora function in mitotic entry in Xenopus egg extracts.

**S112 residue and Tpx2-like motifs are essential for Bora function during mitotic entry in human cells.** Having determined that the Tpx2-like motifs (M1 and M2) and the phosphorylated serine Ser112 (M3) are essential in the context of the minimal Bora[18–120] fragment for function in vitro and in Xenopus egg extracts, we next investigated whether these determinants are also required for full-length Bora function in mitotic entry during the human cell cycle. To this end, we used CRISPR/Cas9 technology to C-terminally tag endogenous Bora with NeonGreen (NG) and the Auxin-inducible degron (AID) motif (denoted Bora-NG-AID) in DLD1 cells constitutively expressing the F-box protein TIR1 (Fig. 9a). Following treatment of the Bora-NG-AID DLD1 cell line with Auxin (A), the SCF[TIR-1] E3-ligase targets Bora-NG-AID for degradation by the 26S proteasome resulting in Bora knockdown[51] (Supplementary Fig. 8a–c). Using CRISPR/Cas9, we also introduced at the AAVS1 locus either a wild-type or a mutant variant of Bora under the control of a doxycycline-inducible promoter (Fig. 9a). Bora variants included the phospho-

Tavernier at al., Fig 6

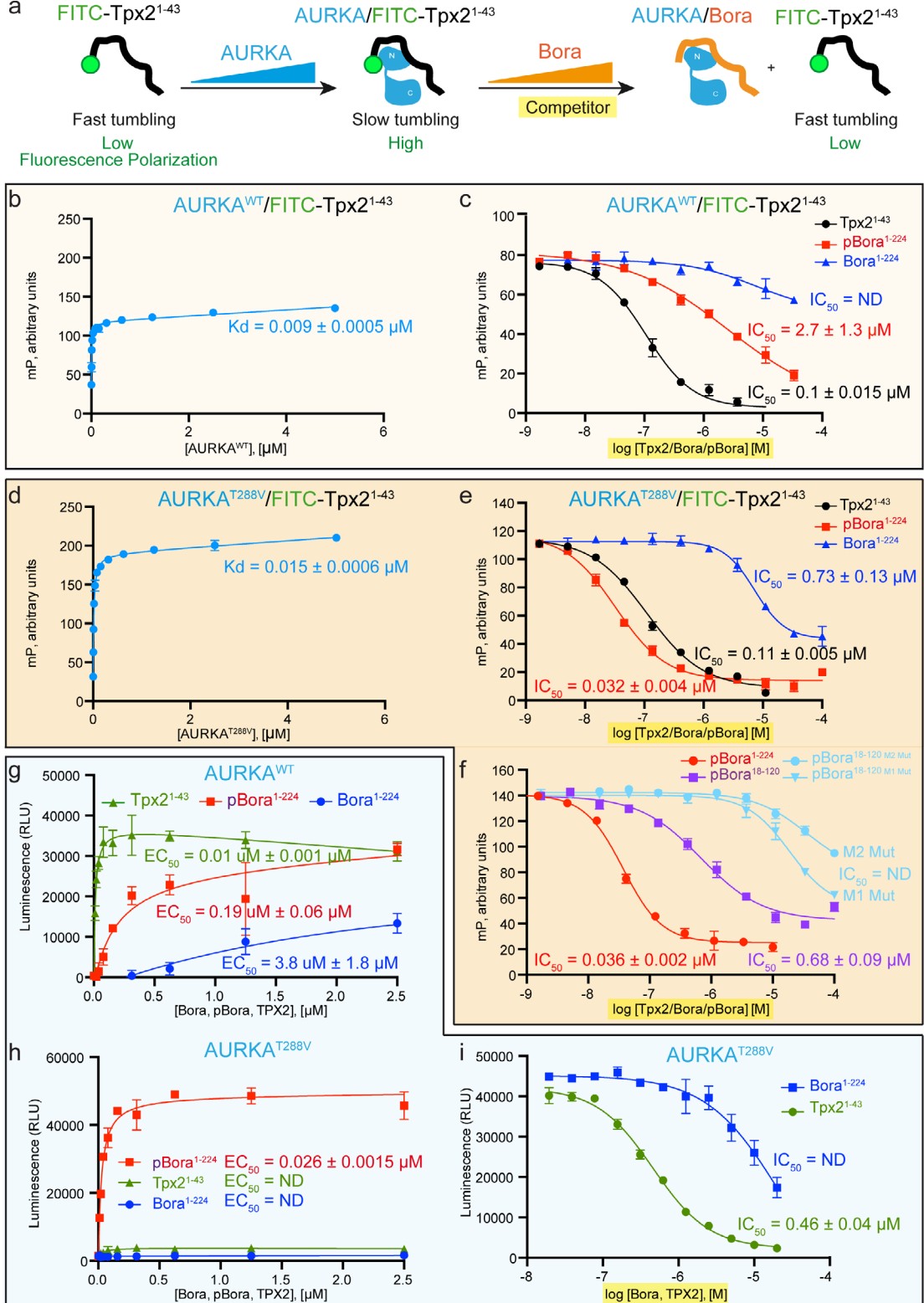

site mutants Bora[S41A], Bora[S112A] (disruption of motif M3), Bora[S137A], Bora[3A [S41A,S112A,S137A]], the deletion mutant Bora[Δ35], and the mutated forms Bora[F25D Y31D] and Bora[F103D F104D] with disruptions to the TPX2-like motifs M1 and M2, respectively. We specifically tested the triple Bora[3A [S41A S112A S137A]] mutant as well as individual Bora phosphosite mutants because

we previously showed that Bora[3A] is defective in mitotic entry after checkpoint recovery from DNA damage[25,26].

Using this experimental setup, we examined mutant Bora function during the human cell cycle by combining endogenous Bora depletion upon Auxin addition with untagged Bora induction under Doxycyclin (D) (Fig. 9a and Supplementary Fig. 8a–c). We

**Fig. 6 Bora and Tpx2$^{1-43}$ compete for AURKA binding. a** Schematics of a competitive displacement binding assay using fluorescein-labeled Tpx2$^{1-43}$ (FITC-Tpx2$^{1-43}$) as a probe. Complex formation between FITC-labeled Tpx2$^{1-43}$ polypeptide and AURKA followed by the disassembly of the FITC-Tpx2$^{1-43}$/AURKA complex by increasing amount of competitor (cold Tpx2$^{1-43}$, Bora, or pBora) was monitored by fluorescence polarization. **b** Binding of fluorescein-labeled Tpx2$^{1-43}$ polypeptide to AURKA$^{WT}$ assessed by monitoring the fluorescence polarization signal in the presence of increasing concentrations of AURKA$^{WT}$. $K_d$ value represents mean value with standard deviations of the mean as error bars ($n = 3$ independent experiment samples). **c** Competitive binding assay where fluorescein-labeled Tpx2$^{1-43}$ polypeptide in complex with AURKA$^{WT}$ is displaced by increasing amount of competitor (cold Tpx2$^{1-43}$, Bora$^{1-224}$, or pBora$^{1-224}$) and monitored by fluorescence polarization signal. Displayed data points and the half maximal inhibitory concentration (IC$_{50}$) value represent the average fluorescence polarization for each reaction conditions with standard deviations of the mean as error bars ($n = 3$ independent experiment samples). ND: not determined. **d** Binding of fluorescein-labeled Tpx2$^{1-43}$ polypeptide to AURKA$^{T288V}$ assessed by monitoring the fluorescence polarization signal in the presence of increasing concentrations of AURKA$^{T288V}$. $K_d$ value represents mean value with standard deviations of the mean as error bars ($n = 3$ independent experiment samples). **e, f** Competitive binding assay where fluorescein-labeled Tpx2$^{1-43}$ polypeptide in complex with AURKA$^{T288V}$ is displaced by increasing amount of cold Tpx2, Bora$^{1-224}$, or pBora$^{1-224}$ competitor (**e**), or by increasing amounts of cold pBora$^{1-224}$, pBora$^{18-120}$, or the indicated pBora$^{18-120}$ mutants in the M1 and M2 motifs. Data presented as in **c**. **g** Activation of AURKA$^{WT}$ ATPase activity by Bora$^{1-224}$, pBora$^{1-224}$, and Tpx2$^{1-43}$ as assessed using the ADP-Glo assay with Kemptide substrate. Displayed data points and EC$_{50}$ values represent the average luminescence for each reaction condition with standard deviations of the mean as error bars ($n = 3$ independent experiment samples). RLU: relative light unit. **h** Activation of AURKA$^{T288V}$ ATPase activity by Bora$^{1-224}$, pBora$^{1-224}$, and Tpx2$^{1-43}$ as assessed using the ADP-Glo assay with Kemptide substrate. Data presented as in **g**. **i** Activity of the AURKA$^{T288V}$/pBora$^{1-224}$ complex in the presence of increasing concentration of competitor Tpx2$^{1-43}$ or Bora$^{1-224}$ assessed using the ADP-Glo assay in the presence of Kemptide substrate. Displayed data points and the half maximal inhibitory concentration (IC$_{50}$) value represent the average fluorescence luminescence for each reaction conditions with standard deviations of the mean as error bars ($n = 3$ independent experiment samples). ND: not determined.

first examined the effect of Bora depletion by Auxin treatment in DLD1 cells expressing Bora-NG-AID. Within 2 h of Auxin treatment, we observed rapid degradation of Bora-NG-AID, which was accompanied by a severe reduction in Plk1 phosphorylation on T210 in unsynchronized cells (Supplementary Fig. 8d). To rigorously evaluate the impact of Bora-NG-AID depletion on cell cycle progression, we synchronized the cells in G1–S phase using a double thymidine block (DTB), and then released the cells by washing out thymidine. We monitored progression through mitosis by FACS and Western blot analysis of cell cycle markers (Fig. 9b and Supplementary Figs. 8e, f and 9). Consistent with previous RNAi experiments[22], Bora-NG-AID depletion severely delayed mitotic entry. While untreated cells accumulated the mitotic marker Histone H3 8 h after release from the DTB, cells treated with Auxin accumulated this marker only after 12 h of release from DTB (Supplementary Fig. 8e, f). FACS analysis confirmed these results and showed that cells depleted of Bora-NG-AID entered mitosis more slowly with a delay of 4 h (Supplementary Fig. 8f).

We next examined the ability of Bora mutants to reverse the observed effects upon Auxin-induced depletion of Bora-NG-AID. We calculated a rescue index (RI), which reflects the capacity of untagged Bora to rescue the delay in mitotic entry resulting from endogenous Bora-NG-AID degradation (Fig. 9c, d). Briefly, for each Bora mutant cell line, we performed a DTB and release regime, and determined from FACS the percentage of cells in the G2 and M phases, at both 10 h and 13 h time points post-release (Fig. 9b). These experiments were performed under three conditions: cells were either untreated for use as reference controls (Ctrl), treated with Auxin (A) to degrade Bora-NG-AID, or treated with Auxin plus doxycycline (A/D) to degrade Bora-NG-AID and simultaneously induce expression of WT Bora or its mutants (Fig. 9b). RI was calculated as a function of cells in G2–M under the three different conditions (RI = [A−A/D] / [A−Ctrl]) (Fig. 9d). As shown in Fig. 9c, the percentage of cells in the G2 and M phases increased in cells depleted of Bora-NG-AID, as compared to untreated control cells, due to mitotic entry delay (Supplementary Fig. 8e, f). If re-expression of a Bora variant was to rescue this phenotype to some degree, the percentage of cells in G2–M should decrease relative to Auxin treatment alone. If the rescue was complete, the percentage of cells at G2–M should equal that observed for untreated control cells (RI = 1). If no rescue was taking place, the percentage of cells

at G2–M should equal that observed for Auxin-treated cells (RI = 0).

As shown in Fig. 9d, a RI value of 0.5 was observed upon induction of wild-type Bora expression at 10 and 13 h after release from the second thymidine block. This indicated that wild-type Bora partially rescued the mitotic delay resulting from Bora-NG-AID depletion possibly due to its lower expression level compared to endogenous Bora (Fig. 9e, compare lanes 1 and 3). Bora$^{S41A}$ and Bora$^{S137A}$ displayed very similar abilities to rescue the cell cycle delay resulting from endogenous Bora-NG-AID depletion (Fig. 9d). In contrast, Bora$^{[S41A\ S112A\ S137A]}$, Bora$^{S112A}$, Bora$^{[F25D\ Y31D]}$, Bora$^{[F103D\ F104D]}$, and Bora$^{\Delta35}$ displayed no ability to rescue the G2–M delay, despite all proteins being efficiently expressed upon addition of doxycycline (Fig. 9e). We corroborated these observations by monitoring T210 phosphorylation on Plk1 10 h after release from DTB (Fig. 9e). The T210 phosphorylation signal was severely reduced upon Auxin addition and this was restored by doxycycline-induced expression of Bora$^{WT}$ (Fig. 9e, compare lanes 1, 2, and 3). Doxycycline-induced expression of Bora$^{S41A}$ and Bora$^{S137}$ similarly restored high levels of T210 phosphorylation (Fig. 9e, lanes 9 and 15) while the other Bora mutants Bora$^{[S41A\ S112A\ S137A]}$, Bora$^{S112A}$, Bora$^{[F25D\ Y31D]}$, Bora$^{[F103D\ F104D]}$, and Bora$^{\Delta35}$ failed to do so (lanes 6, 12, 18, 21, and 24).

To avoid potential complications resulting from thymidine treatment, we monitored mitotic entry by time-lapse video microscopy in asynchronous DLD1 cell populations expressing Bora-NG-AID and doxycycline inducible Bora$^{WT}$ or Bora$^{S112A}$, using fluorescent-tagged RCC1-iFP2 as a reporter of NEBD (Supplementary Fig. 10a, b). More than 1000 cells were analyzed per condition. Bora depletion upon Auxin addition severely delayed mitotic entry for at least 4 h. This phenotype was partially rescued by the induction of Bora$^{WT}$ (+ Doxycycline 5 h before Auxin addition) (Supplementary Fig. 10b). Conversely, the expression of Bora$^{S112A}$ mutant form was totally unable to rescue the kinetics of entry into mitosis (Supplementary Fig. 10c), confirming our results obtained with cells synchronized by a DTB-release regime. Collectively, these results indicate that the Tpx2-like motifs (M1, M2) and S112 phosphorylation (M3) are essential determinants for Bora function in Plk1 activation and mitotic entry during the human cell cycle. These results are also fully consistent with our in vitro data and results obtained in Xenopus egg extracts.

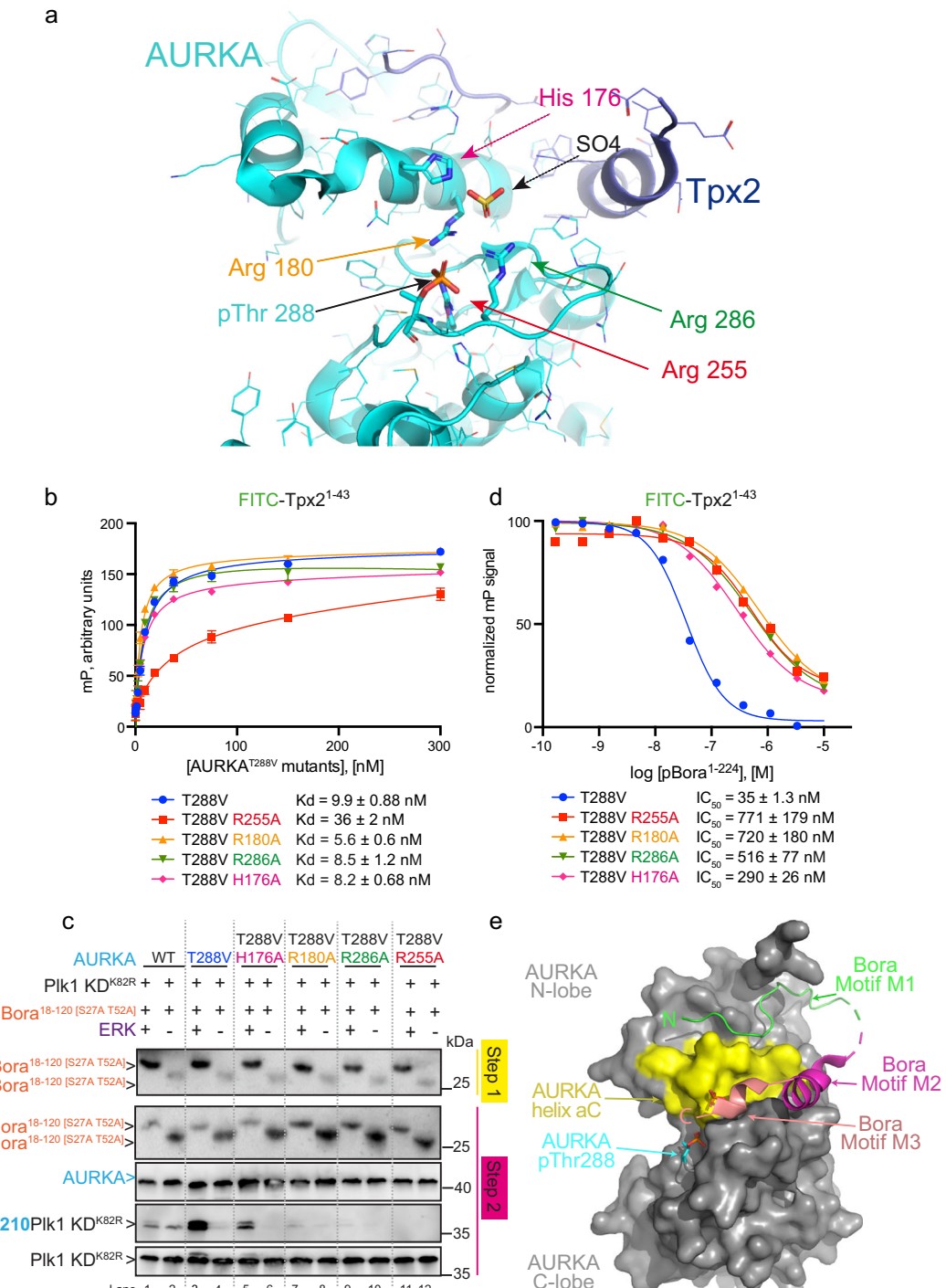

**Fig. 7 Bora binds AURKA through two Tpx2-like motifs that position a phospho-motif in the kinase active site. a** Zoom-in view of the binding interface between Tpx2 and AURKA (PDB: 1OL5). The T-loop phospho-residue Thr288, sulfate ion (yellow), and the phospho-T-loop and sulfate ion coordinating residues of AURKA (H176, R180, R255, and R286) are highlighted in stick representation. **b** Binding of fluorescein-labeled Tpx2[1-43] polypeptide to AURKA mutants assessed by monitoring the fluorescence polarization signal in the presence of increasing concentrations of AURKA mutants. $K_d$ values represent mean value with standard deviations of the mean as error bars ($n = 3$ independent experiment samples). **c** Western blot analysis of 2-step kinase reactions carried out with Bora[18-120 [S27A T52A]] phosphorylated ($+$) or not ($-$) by the ERK kinase (step 1) in the presence of Plk1[K82R] KD (substrate) and AURKA[WT] or mutated versions as indicated (step 2). Blots were probed with antibodies to Bora, AURKA, and phosphoT210 Plk1 or pan Plk1 as indicated (from top to bottom). **d** Competitive displacement of fluorescein-labeled Tpx2[1-43] from the indicated AURKA mutants by increasing concentrations of pBora[1-224] as monitored by fluorescence polarization. Displayed data points and the half maximal inhibitory concentration (IC$_{50}$) value represent the average normalized fluorescence polarization signal for each reaction conditions (starting binding signal $= 100\%$, end titration binding signal $= 0\%$) ($n = 3$ independent experiment samples). **e** Theoretical/illustrative model of pBora binding to the kinase domain of AURKA. AURKA is shown in gray surface with helix αC highlighted in yellow. Bora motifs M1 (blue) and M2 (purple) were modeled by two corresponding motifs from Tpx2 and the phospho-motif M3 (pink) was modeled by the TSS motif from phospho-INCENP.

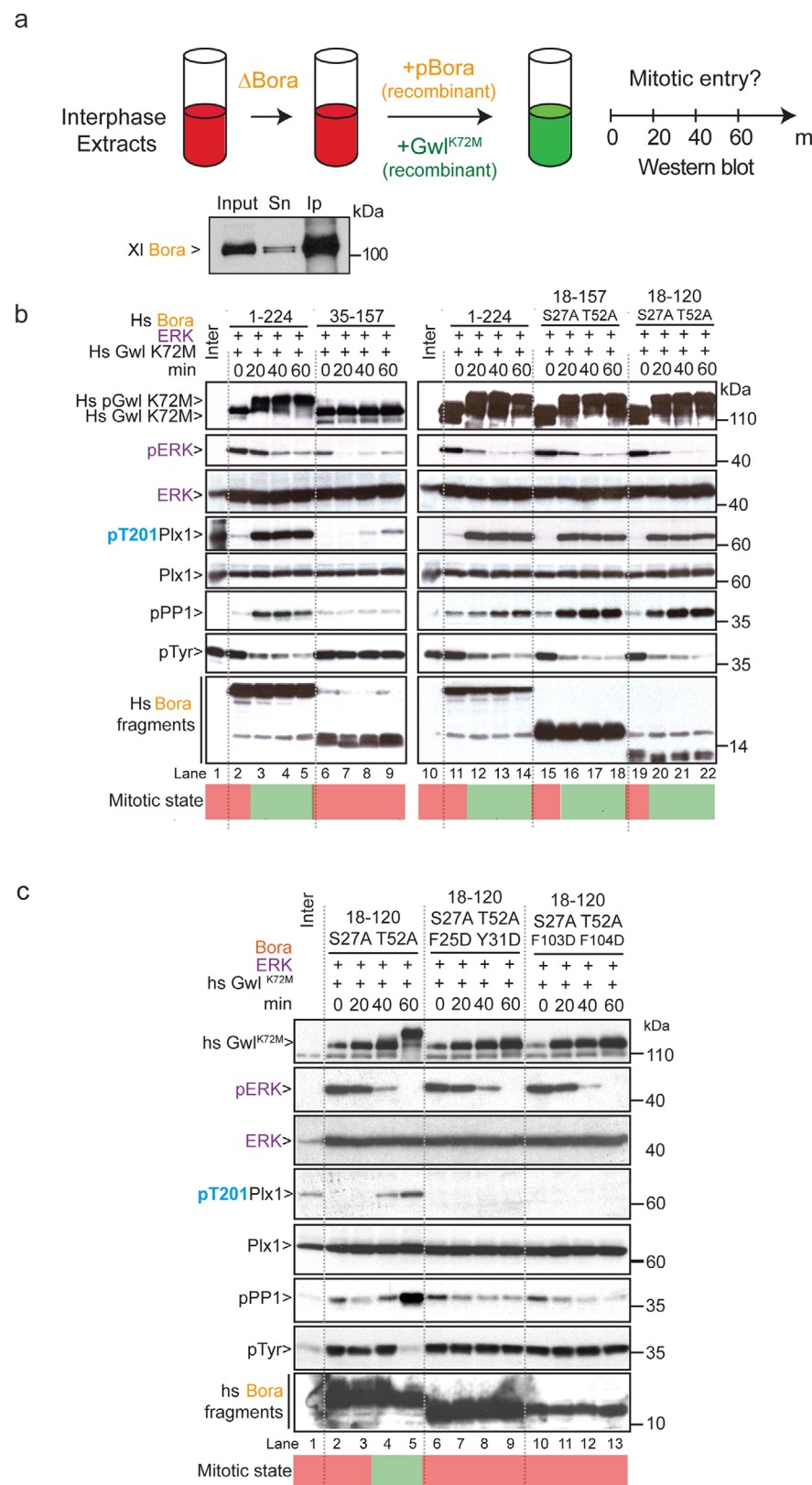

## Discussion

Here, we identify the molecular mechanism by which Bora, phosphorylated minimally on Ser112 by a Cyclin-Cdk activity, acts to promote the AURKA-dependent phosphorylation of Plk1 on the T-loop (residue T210) to trigger timely mitotic entry. We demonstrate that Bora binds directly to AURKA to activate its intrinsic catalytic function. Our data are in line with seminal observations indicating that Bora is a co-factor of AURKA in *Drosophila melanogaster*[23]. Importantly, our mutational studies have uncovered three critical elements in Bora required for AURKA activation in vitro and in vivo. Included are two conserved ~15 residue linear motifs (denoted M1 and M2) separated

**Fig. 8 Tpx2-like motifs and Serine 112 phosphorylation are essential for Bora function in mitotic entry in Xenopus egg extracts. a** Schematic for the structure–function analysis of Bora in mitotic entry in Xenopus egg extracts. Interphase extracts (red) were depleted with Bora antibodies and 30 min later supplemented with Bora fragments (pre-phosphorylated by the ERK kinase) and with the recombinant Gwl$^{K72M}$ hyperactive kinase to force mitotic entry. Samples were collected at different time-points and analyzed by Western blot (0, 20, 40, 60 min) using mitotic markers to determine whether the extracts enter into mitosis (green) or not. The Western blot shows endogenous Bora levels before (Input) and after (Sn: supernatant) immunodepletion (Ip) from Xenopus egg extracts. **b** Interphase extracts were immunodepleted using anti-Bora antibodies and 30 min later supplemented with a recombinant hyperactive Gwl$^{K72M}$ and Bora$^{1–224}$ (lanes 2–5), Bora$^{35–157}$ (lanes 6–9), Bora$^{1–224}$ (lanes 11–14), Bora$^{18–157 [S27A T52A]}$ (lanes 15–18), Bora$^{18–120 [S27A T52A]}$ (lanes 19–22). A fraction of the extracts was collected at the indicated time-points (0, 20, 40, 60 min) and analyzed by Western blot using specific antibodies to monitor the levels of huGwl, ERK, Plk1, huBora and phosphorylation of Plx1 on T201 (pPlk1), PP1 phosphatase on threonine 320 (pPP1), Cdk on Tyr15 (pTyr), and ERK on Thr202 and Tyr204. The green and red areas at bottom represent high and low CyclinB-Cdk1 (MPF) activity periods. **c** Interphase extracts were immunodepleted using anti-Bora antibodies and 30 min later supplemented with recombinant hyperactive Gwl (Gwl$^{K72M}$) and Bora$^{18–120}$ $^{[S27A T52A]}$ (lanes 2–5), Bora$^{18–120 [S27A T52A F25D Y31D]}$ (lanes 6–9), Bora$^{18–120 [S27A T52A F103D F104D]}$ (lanes 10–13). A fraction of the extracts was collected at the indicated time-points (0, 20, 40, 60 min) and analyzed by Western blot using specific antibodies to monitor the levels of Gwl, ERK, Plk1, and Bora as well as phosphorylation of Plx1 on T201 (pPlk1), PP1 phosphatase on threonine 320 (pPP1), and Cdk on Tyr15 (pTyr). The green and red areas at bottom represent high and low CyclinB-Cdk1 (MPF) activity periods.

by an 85 residue linker and an evolutionarily conserved Pro/phospho-Ser112/Pro motif (denoted M3) generated by a proline-directed Cyclin-Cdk kinase. Phosphorylation of motif M3 is likely initiated by CyclinA2-Cdk1 (ref. [27]) and sustained by CyclinB1-Cdk1 (refs. [25,52]).

Based on similarity to Tpx2, we speculate that Bora motif M1 binds in an extended manner parallel to the top surface of helix αC whereas motif M2 adopts a helical conformation and binds parallel to the bottom surface of helix αC (Fig. 7a). As Bora motif M3 is located immediately C-terminal to motif M2, this binding model is attractive because it orients the Ser112 phospho-moiety in motif M3 in close proximity to two possible phosphate coordination sites in AURKA. Included are the binding site for the phosphorylated T-loop of AURKA (T288) and a sulfate ion-binding site that is conserved with AURKB/C and used by AURKB/C to bind a phospho-motif in its corresponding regulator INCENP (Fig. 7c). Mutation of residues in both sites abrogates the responsiveness of AURKA$^{T288V}$ to pBora and diminishes binding of pBora to AURKA suggesting that the S112 phosphate moiety on Bora might engage both binding sites. Final confirmation of the underlying binding mechanism awaits a detailed atomic structure of the AURKA–pBora complex. Importantly, we demonstrate that motifs M1, M2, and M3 are all essential for Bora function in vitro and also in vivo to promote mitotic entry in Xenopus egg extracts and in human cells.

X-ray crystallographic[40,41], enzymatic[53], and spectroscopic[54] studies have established that Tpx2 binding and T-loop phosphorylation activate AURKA catalytic function through distinct but complementary mechanisms (also reviewed in refs. [46,55]). Two key structural elements of the KD that are recurring targets for regulation include the DFG (aspartate, phenylalanine, glycine) motif and helix αC, both of which function to productively coordinate ATP binding and catalytic elements.

Tpx2 binding drives activation primarily by promoting a productive inward conformation of the DFG motif through the anchoring of an inherently flexible helix αC. This is sufficient to promote a 10-fold increase in AURKA catalytic function in vitro. In contrast, T-loop phosphorylation serves primarily to promote a productive conformation of the activation segment, resulting in a more pronounced 100-fold increase in catalytic function[53]. Consistent with the two regulatory arms acting by independent rather than redundant mechanisms, the combination of Tpx2 binding and T-loop phosphorylation results in a synergistic 1000-fold boost in catalytic function[41,53].

Interestingly, Bora contains not only two Tpx2-like motifs (M1 and M2) that mediate binding to the N-lobe of AURKA, but also a highly conserved and essential phosphosite S112 within motif M3. What does this design portend for how Bora mechanistically

regulates AURKA kinase activity? Based on mutational and competition binding experiments, we expect that the two Tpx2-like motifs M1 and M2 will bind and act similar to Tpx2 by anchoring the position of helix αC to enforce a productive inward conformation of the DGF motif. Based on mutational analysis, the phospho-motif of M3 binds to either or both of two likely sites on AURKA. If the phospho-motif of M3 binds to the site normally occupied by the phosphorylated (pT288) T-loop, we expect that M3 would influence AURKA catalytic function (as observed for phosphorylation of the T-loop) by stabilizing a productive conformation of the activation segment. If instead the phospho-motif of M3 binds to the sulfate ion-binding site, M3 could influence AURKA catalytic function either by stabilizing a productive conformation of the activation segment or by anchoring the position of helix αC and in turn by modulating the DFG motif. Either binding mode could in principle account for the observed behavior of Bora on AURKA and both appear feasible based on geometric constraints.

AURKA is instrumental for mitotic entry and progression. Indeed, AURKA not only regulates commitment to mitosis, but also centrosome maturation, mitotic spindle formation and positioning, spindle assembly checkpoint (SAC), central spindle assembly, and cytokinesis[35,56,57]. Our results, along with previous observations, indicate that AURKA regulation in these processes occurs typically at two levels: firstly by allostery through binding to a growing list of trans acting factors and secondly by phosphorylation of the T-loop of AURKA[4]. Our current results reveal a twist in AURKA regulation, whereby pBora activates AURKA both by allosteric binding and by providing a phosphate group in trans that compensates for the lack of AURKA T-loop phosphorylation.

Importantly, we show that this "one stop shopping" mode of AURKA regulation is essential for maintaining the mitotic state of CSF-arrested Xenopus egg extracts (Fig. 4). In this context, the mechanism of action of pBora is ideally suited to sustain AURKA catalytic activity against a prevailing complement of counteracting phosphatases. It also provides a conduit whereby CyclinA2-Cdk activity can regulate the activation of AURKA to trigger the Plk1-CyclinB-Cdk1 cascade that drives mitotic entry. AURKA is degraded at the end of mitosis[58] and thus must be resynthesized during interphase of the next cell cycle. Once translated, AURKA auto-phosphorylates on its own T-loop but this activity is counteracted by the PP1 and PP2A phosphatases in interphase[43,59,60] and by PP6 while bound to Tpx2 during mitosis[61]. As dephosphorylation maintains AURKA in an inactive state, the crucial question arises as to how AURKA overcomes the repressive effect of PPases to drive mitotic commitment. Our observations show that pBora provides a

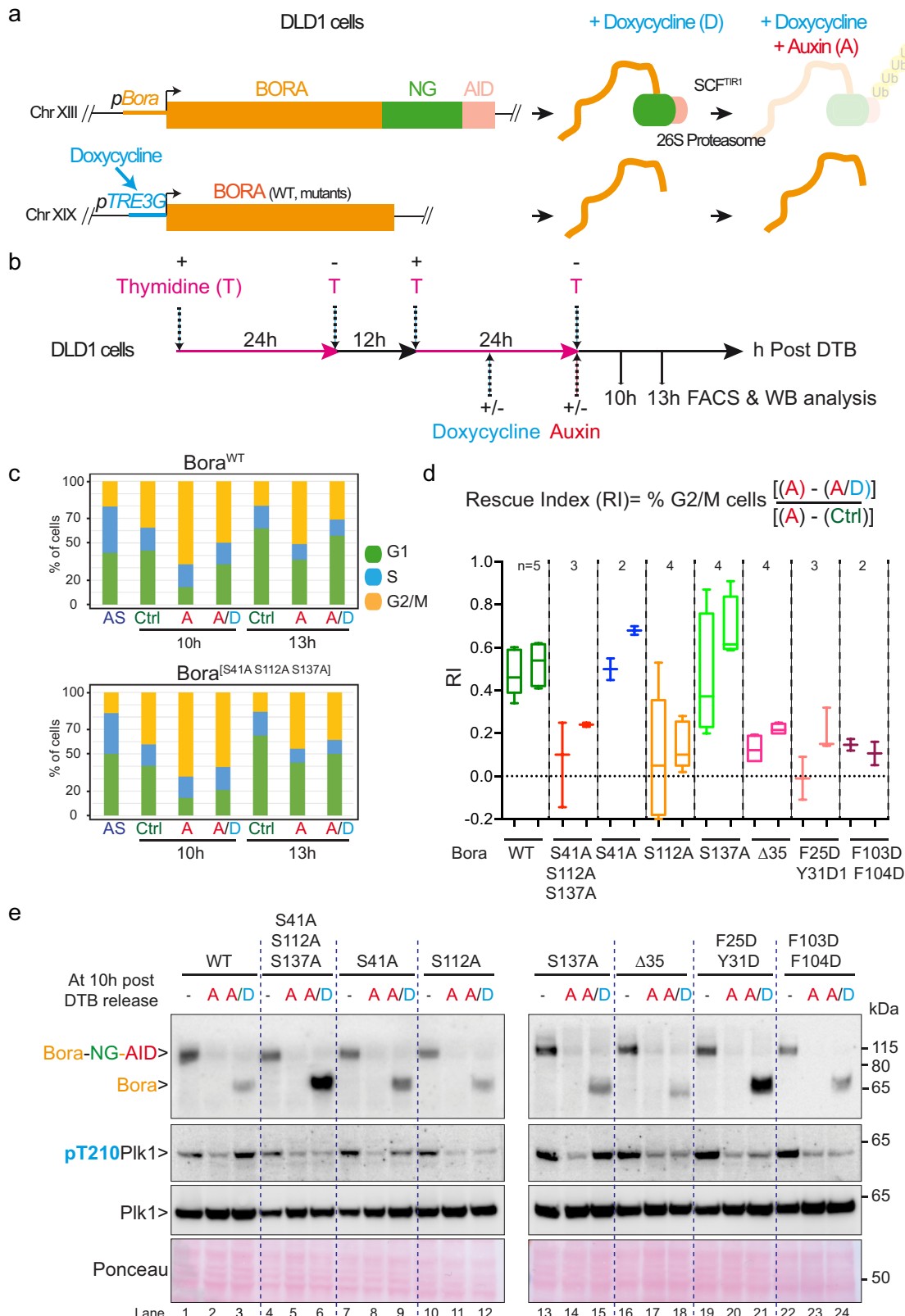

mechanism to activate AURKA activation in spite of counteracting phosphatases. Once activated by pBora, AURKA phosphorylates and activates Plk1, which in turn activates CyclinB-Cdk1 thereby transitioning the cell from a phosphatase-dominant to a mitotic kinase-dominant state. How precisely the Ser112 phospho-epitope of Bora escapes the action of the phosphatases

that counteract AURKA auto-phosphorylation, remains an open and intriguing question.

Our results also show that the phosphomimetic AURKA[T288D] mutant commonly used as a proxy for the activated form of AURKA actually behaves in our in vitro kinase assay more like the constitutive inactive mutant AURKA[T288V]. This once again

**Fig. 9 S112 residue and Tpx2-like motifs are essential for Bora function during the mammalian cell cycle. a** Schematic for the structure–function analysis of Bora during the mammalian cell cycle. Endogenous Bora located on chromosome (Chr) XIII was C-terminally tagged with Neongreen (NG) and the Auxin-Inducible degron (AID) in DLD1 cells constitutively expressing the Tir1 subunit of the SCF$^{TIR1}$ E3-ligase. A wild-type or a mutant copy of untagged Bora driven by the doxycycline promoter (pTRE3G) was inserted on Chr XIX of the same cells using CRISPR/Cas9 genome editing. Doxycycline promotes expression of untagged wild-type or mutant Bora, while Auxin induces SCF$^{TIR1}$-mediated proteasome degradation of Bora-NG-AID resulting in a Bora knockdown. **b** Workflow used to follow the progression of the cells through the cell cycle. Expression of Bora$^{WT}$ or the indicated mutants were induced with doxycycline during the second thymidine block. Bora-NG-AID was then depleted at the time of release from DTB using Auxin. Cells were collected 10 and 13 h after DTB release for Western blot and FACS analyses. **c** Histogram plots showing the cell cycle distributions (percentage of cells in G1, S, and G2/M) for DLD1-Bora-NG-AID cell lines, expressing either Bora$^{WT}$ or Bora$^{[S41A\ S112A\ S137A]}$ and treated as outlined in **b**. AS: Asynchronous cells; Ctrl: non-treated; A: Auxin-treated cells; A/D: Doxycyclin + Auxin-treated cells. **d** Box plots indicate median (middle line) and min max (Wiskers) showing the quantified rescue index (RI) for each DLD1 Bora-NG-AID cell line expressing Bora$^{WT}$ (in dark green) or the indicated mutants: Bora$^{[S41A\ S112A\ S137A]}$ (in red), Bora$^{S41A}$ (in blue), Bora$^{S112A}$ (in orange), Bora$^{S137A}$ (in green), Bora$^{\Delta35}$ (in magenta), Bora$^{[F25D\ Y31D]}$ (in pink), and Bora$^{[F103D\ F104D]}$ (in purple) assessed 10 (first column) and 13 (second column) hours after DTB release; *n* indicates the number of replicates. **e** Western blot analysis of protein lysates from a DLD1-Bora-NG-AID cell line expressing Bora$^{WT}$ or the indicated mutants 10 h after DTB release as depicted in **b**, **c**, and **d**. Bora, pT210-Plk1, and panPlk1 antibodies were used for these analyses (from top to bottom). (−) Asynchronous cells.

highlights the need for caution in the use and interpretation of phospho-mimetic mutations to probe protein kinase functions in biological contexts.

In conclusion, we demonstrate here that Bora is a direct regulator of AURKA that acts by a dual mechanism involving allostery and by providing a phosphate in trans that compensates for the lack of AURKA T-loop phosphorylation. As AURKA and Bora are overexpressed in several cancers and act as oncogenes[62–66], deciphering how AURKA is mechanistically activated by Bora in time and space may pave the way for the development of innovative therapeutic approaches that target only a subset of AURKA's multiple biological functions.

## Methods

**Cell cultures.** DLD1 (ATTCC CCL-221) were grown in GIBCO DMEM (Dulbecco's modified Eagle's medium) supplemented with 10% GIBCO FBS (Fetal Bovine Serum), 2 mL L-glutamine, 100 units/mL penicillin, 100 μg/mL streptomycin (Life Technologies), and 50 μg/mL Normocin (Invivogen). All the cell lines used in the study were regularly tested for contamination.

**_Xenopus laevis_ induction and husbandry.** Regulations for the use of _Xenopus laevis_, as outlined in the Animals Scientific Procedures Act (ASPA) and implemented by the Direction Generale de la Recherche et Innovation, Ministère de l'Enseignement Supérieur de la l'Innovation of France were followed. Frogs were obtained from Centre de Ressources Biologiques Xénopes (CRB) of Rennes, France, and kept in a _Xenopus_ research facility at the CRBM (Facility Center approved by the French Goverment Approval no. B34-172-39). Females were injected with 500 U of Chorulon (Human Chorionic Gonadotrophin) and oocytes layed 18 h later were used for experiments. Adult females were exclusively used to obtain eggs. All procedures were approved by the Direction Generale de la Recherche et Innovation, Ministère de L'Enseignement Supérieur de l'Innovation of France (Approval no. APAFIS#4971-2016041415177715v4).

**Molecular biology and DNA manipulation.** All the DNA constructs were generated by Gateway or with restriction enzymes. All DNA constructs were verified by sequencing. Oligonucleotides used for site-directed mutagenesis were purchased from Eurofins or MerckMillipore and are detailed in the Supplementary Table 3. The list of plasmids used in this study is provided in Supplementary Table 2. AAVS1 T2 CRIPR in pX330 was a gift from Masato Kanemaki (Addgene plasmid # 72833; http://n2t.net/addgene:72833; RRID:Addgene_72833), pX330-U6-Chimeric_BB-CBh-hSpCas9 was a gift from Feng Zhang (Addgene plasmid # 42230; http://n2t.net/addgene:42230; RRID:Addgene_42230). pMK243 (Tet-OsTIR1-PURO) was a gift from Masato Kanemaki (Addgene plasmid # 72835; http://n2t.net/addgene:72835; RRID:Addgene_72835). pET-His6-ERK2-MEK1_R4F_coexpression was a gift from Melanie Cobb (Addgene plasmid # 39212; http://n2t.net/addgene:39212; RRID:Addgene_39212).

**Recombinant protein expression.** Full-length human AURKA$^{WT}$, all AURKA mutants, all human Bora fragments, full-length human CyclinA2-Cdk2, and the fragment 10-360 of human ERK were expressed in _Escherichia coli_ BL21 DE3 pLyS RIL (Thermo Fisher Scientific). Full-length human Plk1$^{K82R}$, Plk1$^{1–370\ K82R}$ delta PBD, Plk1$^{1–330}$ KD$^{WT}$ or $^{K82R}$ were expressed in _E. coli_ Rosetta DE3 (MerckMillipore). Transformed bacteria were incubated in LB media at 37 °C under agitation and the cultures were shifted at 18 °C at OD = 0.8. Protein expression was induced

by addition of IPTG (final concentration 500 μM) and the cultures were incubated overnight at 18 °C under agitation.

**Recombinant protein purification.** AURKA and Plk1 were cloned as a TEV cleavable 6xHis-GST fusion, which leaves two or seven non-native residues (GA and GAMDPEF, respectively) at the N-terminus after TEV digestion. All the subsequent steps were performed at 4 °C. Bacteria were pelleted and resuspended in lysis buffer (25 mM Tris pH 7.5, 300 mM NaCl, 1 mM TCEP, 2 mM PMSF) and lysed in a homogenizer (Avestin). Total extract was clarified by centrifugation (30,000 × *g*, 30 min) and the resulting soluble fraction was passed through a 0.45-μm filter and loaded three times on 5 mL of packed glutathione sepharose 4B medium (Cytiva). After extensive washes, 1 mg of 6xHis-TEV protease was incubated overnight onto the glutathione sepharose medium. Cleaved protein was then eluted by gravity. The salt concentration was lowered to 50 mM NaCl by dilution and the protein was loaded on a 5-mL HiTrap SP HP column (Cytiva) and eluted with a salt gradient on an Akta FPLC system (Cytiva). The pure fractions were collected and concentrated using an ultrafiltration centrifugal protein concentrator (MerckMillipore). The protein was finally resolved on a HiLoad 16/600 Superdex 200 pg sizing column (Cytiva) equilibrated in 25 mM Tris pH 7.5, 150 mM NaCl, 1 mM TCEP. The fractions containing the protein were pooled and concentrated using an ultrafiltration centrifugal protein concentrator. The purified protein was aliquoted, flash frozen in liquid nitrogen, and stored at −80 °C.

All the fragments of human Bora were cloned as a TEV cleavable 6xHis fusions, which leaves non-native residues (GAMDPEF) at the N-terminus after TEV digestion. Bacteria were pelleted and resuspended in lysis buffer (25 mM Tris pH 7.5, 150 mM NaCl, 20 mM imidazole, 1 mM TCEP, 2 mM PMSF) and lysed in a homogenizer at 4 °C. Total extract was clarified by centrifugation (30,000 × *g*, 30 min, 4 °C) and the resulting insoluble pellet was resuspended in 25 mM Tris pH 7.5, 6 M guanidine HCl, 1 mM TCEP at room temperature, sonicated and clarified by centrifugation (30,000 × *g*, 30 min). The supernatant was passed through a 0.45-μm filter and loaded on a 5-mL Ni++ HiTrap Chelating HP (Cytiva) equilibrated in 25 mM Tris pH 7.5, 6 M guanidine HCl, 1 mM TCEP. After extensive washes, 6xHis-Bora was eluted by 5 column volumes of elution buffer (25 mM Tris pH 7.5, 6 M guanidine HCl, 500 mM imidazole, 1 mM TCEP). Fractions containing the protein were pooled and dialyzed against the dialysis buffer (25 mM Tris pH 7.5, 150 mM NaCl, 1 mM TCEP) during 4 h to remove the guanidine HCl. 6xHis-Bora precipitated in absence of guanidine HCl. The content of the dialysis bag was transferred into a 50-mL canonical tube and was centrifuged at 4000 × *g* during 10 min. The supernatant was discarded and the resulting pellet containing Bora was resuspended overnight by agitation in 50 mL of resuspending buffer (25 mM Tris pH 7.5, 150 mM NaCl, 2 mM TCEP). All the subsequent steps were performed at 4 °C. The solubilized 6xHis-Bora was clarified by centrifugation at 4000 × *g* during 10 min. The supernatant was concentrated to 5 mL using an ultrafiltration centrifugal protein concentrator (MerckMillipore) and incubated overnight with 2 mg of 6xHis-TEV protease. Cleaved Bora was then loaded on a 5-mL Ni++ HiTrap Chelating HP (Cytiva) to remove 6xHis-TEV and 6xHis cleaved tag. The flow-through containing Bora was concentrated using an ultrafiltration centrifugal protein concentrator. Bora was finally resolved on a Superdex 200 10/300 GL sizing column (Cytiva) equilibrated in 25 mM Tris pH 7.5, 150 mM NaCl, 1 mM TCEP. The fractions containing Bora were pooled and concentrated using an ultrafiltration centrifugal protein concentrator. The purified protein was aliquoted, flash frozen in liquid nitrogen, and stored at −80 °C.

Full-length human Cdk2 and CyclinA2 were cloned, respectively, as a Precision cleavable GST fusion and TEV cleavable 6xHis fusion, which leave non-native residues (GSPGS on Cdk2 and GA on CyclinA2) at the N-terminus after digestion and were expressed separately. GST-Cdk2 was coexpressed with GST-Cak from _Saccharomyces cerevisiae_ essentially as described[67]. All the subsequent steps were performed at 4 °C. Bacteria expressing GST-Cdk2 or 6xHis-CyclinA2 were

pelleted, mixed, and resuspended in lysis buffer (25 mM HEPES pH 7.5, 250 mM NaCl, 5 mM beta-mercaptoethanol) and lysed in a homogenizer. Total extract was clarified by centrifugation (30,000 × g, 30 min) and the resulting soluble fraction was passed through a 0.45-μm filter and loaded three times on 5 mL of packed glutathione Sepharose 4B medium (Cytiva). After extensive washes, 1 mg of 6xHis-TEV and 1 mg of GST-Precision protease were incubated overnight onto the glutathione Sepharose medium under rotation. Cleaved CyclinA2-Cdk2 were then eluted by gravity. The salt concentration was lowered to 50 mM NaCl by dilution and the proteins were loaded on a 5-mL HiTrap Q HP (Cytiva) and eluted with a salt gradient on an Akta FPLC system (Cytiva). The pure fractions were collected and concentrated using an ultrafiltration centrifugal protein concentrator. CyclinA2-Cdk2 complex was finally resolved on a HiLoad 16/600 Superdex 200 pg sizing column (Cytiva) equilibrated in 25 mM HEPES pH 7.5, 250 mM NaCl, 1 mM TCEP. The fractions containing the CyclinA2-Cdk2 complex were pooled and concentrated using an ultrafiltration centrifugal protein concentrator. The purified protein complex was aliquoted, flash frozen in liquid nitrogen, and stored at −80 °C.

ERK was produced essentially as described[68]. Briefly, bacteria expressing 6xHis-ERK 10-360 were pelleted and resuspended in lysis buffer (25 mM HEPES pH 7.5, 300 mM NaCl, 20 mM Imidazole, 5 mM beta-mercaptoethanol) and lysed in a homogenizer. Total extract was clarified by centrifugation (30,000×g, 30 min) and the resulting soluble fraction was passed through a 0.45-μm filter and loaded on a 5-mL Ni++ HiTrap Chelating HP (Cytiva) and eluted with an imidazole gradient on an Akta FPLC system (Cytiva). The pure fractions were collected and concentrated using an ultrafiltration centrifugal protein concentrator (MerckMillipore). The protein was finally resolved on a HiLoad 16/600 Superdex 200 pg sizing column (Cytiva) equilibrated in 25 mM HEPES pH 7.5, 200 mM NaCl, 2 mM TCEP. The fractions containing the protein were pooled and concentrated using an ultrafiltration centrifugal protein concentrator. The purified protein was aliquoted, flash frozen in liquid nitrogen, and stored at −80 °C.

Concentration of all the purified recombinant proteins was determined using UV spectrophotometry.

**Kinase assay**. The assay was performed in two steps. The step 1 consisted of the phosphorylation of 2 μM of Bora fragments by CyclinA2-Cdk2 or ERK (molar ratio 5:1) in 10 μL of kinase buffer (25 mM HEPES pH 7.5, 150 mM NaCl, 20 mM MgCl₂, 1 mM DTT, 200 μM ATP) incubated for 1 h at 30 °C. The step 2 consisted of the phosphorylation of 200 nM of Plk1$^{K82R}$ by AURKA (molar ratio 1:1) in presence of 400 nM of Bora from step 1 in 30 μL of kinase buffer incubated for 30 min at 30 °C.

**AURKA$^{WT}$ dephosphorylation**. Two milligrams of AURKA$^{WT}$ was mixed with recombinant 6xHIS-Lambda phosphatase (molar ratio 400:1) in 1 mL of phosphatase buffer (50 mM HEPES pH 7.5, 100 mM NaCl, 1 mM MnCl₂, 1 mM TCEP, 0.01% Brij 35) and incubated for 1 h at 30 °C. The mixture was then loaded on a 1-mL Ni++ HiTrap Chelating HP (Cytiva) to remove the 6xHis-Lambda. The flow-through containing dephosphorylated AURKA$^{WT}$ was concentrated using an ultrafiltration centrifugal protein concentrator (MerckMillipore). Dephosphorylated AURKA$^{WT}$ was finally resolved on a Superdex 200 10/300 GL sizing column (Cytiva) equilibrated in 25 mM Tris pH 7.5, 150 mM NaCl, 1 mM TCEP. The fractions containing dephosphorylated AURKA$^{WT}$ were pooled and concentrated using an ultrafiltration centrifugal protein concentrator. The purified protein was aliquoted, flash frozen in liquid nitrogen, and stored at −80 °C. The dephosphorylation efficiency was determined by intact mass-spectrometry.

**AVI-Bora$^{1-224}$ biotinylation**. One milligram of AVI-Bora$^{1-224}$ was mixed with recombinant 6xHIS-BirA (molar ratio 60:1) in 1 mL of biotinylation buffer (50 mM Bicine pH 8.3, 12.5 mM ATP, 12.5 mM MgOAc, 625 μM biotine) and incubated for 1 h at 30 °C. The mixture was then loaded on a 1-mL Ni++ HiTrap Chelating HP (Cytiva) to remove the 6xHis-BirA and free biotin. The flow-through containing biotinylated AVI-Bora$^{1-224}$ was concentrated using an ultrafiltration centrifugal protein concentrator (MerckMillipore). Biotinylated AVI-Bora$^{1-224}$ was finally resolved on a Superdex 200 10/300 GL sizing column (Cytiva) equilibrated in 25 mM Tris pH 7.5, 150 mM NaCl, 1 mM TCEP. The fractions containing biotinylated AVI-Bora$^{1-224}$ were pooled and concentrated using an ultrafiltration centrifugal protein concentrator. The purified protein was aliquoted, flash frozen in liquid nitrogen, and stored at −80 °C. The biotinylation efficiency was determined by intact mass-spectrometry.

**Preparative phosphorylation of Bora$^{1-224}$ by 6xHis-AURKA$^{WT}$**. One milligram of Bora$^{1-224}$ was mixed with recombinant His-AURKA$^{WT}$ (molar ratio 10:1) in 1 mL of kinase buffer (25 mM Tris pH 7.5, 150 mM NaCl, 20 mM MgCl₂, 1 mM TCEP, 2 mM ATP) and incubated for 1 h at 30 °C. The mixture was then loaded on a 1-mL Ni++ HiTrap Chelating HP (Cytiva) to remove the 6xHis-AURKA$^{WT}$. The flow-through containing phosphorylated Bora$^{1-224}$ was concentrated using an ultrafiltration centrifugal protein concentrator (MerckMillipore). Phosphorylated Bora$^{1-224}$ was finally resolved on a Superdex 200 10/300 GL sizing column (Cytiva) equilibrated in 25 mM Tris pH 7.5, 150 mM NaCl, 1 mM TCEP. The fractions containing biotinylated phosphorylated Bora$^{1-224}$ were pooled and concentrated

using an ultrafiltration centrifugal protein concentrator. The purified protein was aliquoted, flash frozen in liquid nitrogen, and stored at −80 °C. The phosphorylation efficiency was controlled by intact mass-spectrometry.

**Preparative phosphorylation of $^{15}$N-Bora$^{1-224}$ and biotinylated AVI-Bora$^{1-224}$ by Cyclin A2-Cdk2**. Five milligrams of $^{15}$N-Bora$^{1-224}$ or biotinylated AVI-Bora$^{1-224}$ were mixed with recombinant CyclinA2-Cdk2 (molar ratio 20:1) in 1 mL of kinase buffer (25 mM Tris pH 7.5, 150 mM NaCl, 20 mM MgCl₂, 1 mM DTT, 2 mM ATP) and incubated for 1 h at 30 °C. The mixture was then loaded on a HiLoad 16/600 Superdex 200 pg sizing column (Cytiva) equilibrated in 25 mM Tris pH 7.5, 150 mM NaCl, 1 mM TCEP. The fractions containing the protein of interest were pooled and concentrated using an ultrafiltration centrifugal protein concentrator. The purified protein was aliquoted, flash frozen in liquid nitrogen, and stored at −80 °C. The phosphorylation efficiency was controlled by intact mass-spectrometry.

**Pull-down assay**. Five-hundred nanograms of biotinylated AVI-Bora$^{1-224}$ phosphorylated or not by CyclinA2-Cdk2 are mixed with AURA$^{WT}$ and/or Plk1 KD$^{WT}$ (molar ratio 1:1) in 100 μL of binding buffer (25 mM Tris pH 7.5, 150 mM NaCl, 20 mM MgCl₂, 1 mM DTT, 0.05% Tween 20) and incubated on ice for 30 min. Then, the mixture was loaded on 5 μL of packed streptavidin sepharose (Cytiva) beads and incubated on ice for 30 min with intermittent gentle shaking. The beads were washed five times with the binding buffer and proteins were eluted by addition of Laemmli buffer on beads. Proteins were then detected by immunoblotting and visualized by treating the blots with ECL (MerckMillipore).

**SPR experiments**. Biotinylated AVI-Bora$^{1-224}$ (7.95 mg/mL; i.e. 296 μM; MW 26800.26) and biotinylated phosphorylated AVI-Bora$^{1-224}$ (12.26 mg/mL; i.e. 457 μM; MW 27440—calculated from average of eight phosphorylations) were diluted to 1.9 μM in running buffer (25 mM HEPES pH 7.5; 150 mM NaCl; 20 mM MgCl₂; 1 mM TCEP; 0.01% Brij 35), and immobilized to streptavidin sensor chip (Series S Sensor Chip SA from Cytiva) using Biacore S200 in running buffer at 25 °C; 7806 RUs of biotinylated AVI-Bora$^{1-224}$ were immobilized to flow cell 2 (flow cell 1 was used as a reference for flow cell 2). Based on AURKA$^{WT}$ $R_{max}$ response, 13% of Bora was active, and based on AURKA$^{T288V}$ $R_{max}$ response, 12% of Bora was active; 6296 RUs of biotinylated AVI-pBora$^{1-224}$ were immobilized to flow cell 4 (flow cell 3 was used as a reference for flow cell 4).

Based on AURKA$^{WT}$ $R_{max}$ response, 25% of the phosphorylated AVI-Bora$^{1-224}$ was active, and based on AURKA$^{T288V}$ $R_{max}$ response, 20% of phosphorylated AVI-Bora$^{1-224}$ was active. Theoretical $R_{max}$ for AURKA$^{WT}$ (MW = 45937.49) interaction with Bora was 13379 RU whereas measured $R_{max}$ for AURKA$^{WT}$ interaction with Bora$^{1-224}$ was 1761 RU; percent active = 1761/13,379 = 13%. Theoretical $R_{max}$ for AURKA$^{T288V}$ (MW = 45935.52) interaction with Bora$^{1-224}$ was 13,379 RU, whereas measured $R_{max}$ for AURKA$^{WT}$ interaction with Bora$^{1-224}$ was 1585 RU; percent active = 1585/13,379 = 12%. Theoretical $R_{max}$ for AURKA WT (MW = 45937.49) interaction with phosphorylated Bora$^{1-224}$ was 10,540 RU, measured $R_{max}$ for AURKA WT interaction with phosphorylated Bora$^{1-224}$ was 2618 RU; percent active = 2618/10,540 = 25%. Theoretical $R_{max}$ for AURKA$^{T288V}$ (MW = 45935.52) interaction with phosphorylated Bora$^{1-224}$ was 10,540 RU, measured $R_{max}$ for AURKA$^{WT}$ interaction with phosphorylated Bora$^{1-224}$ was 2062 RU; percent active = 2062/10,540 = 20%. All protein–protein interactions were tested at 25 °C, flow rate 10 μL/min; injection time 120 s; dissociation time 600 s. Regeneration of the chip surface was performed after each cycle using 30 s injection of 10 mM glycine pH 3 at increased flow rate of 30 μL/min. After regeneration, the sensograms returned to baseline. Raw sensogram data were analyzed and fit using Biacore S200 analysis software (Cytiva).

**ADP-Glo™ assay**. The ATPase activity of recombinant AURKA was evaluated using the ADP-Glo assay Kinase kit (Promega). Bora$^{1-224}$ or pBora$^{1-224}$ was titrated to 10 nM of AURKA$^{WT}$ or AURKA$^{T288V}$ in a buffer containing 40 mM Tris pH 7.5, 150 mM NaCl, 10 mM MgCl₂, 1 mM DTT, 0.1 mg/mL BSA, and 0.01% Brij 35. The final reaction volume was 20 μL. When 100 μM Kemptide (LRRASLG, Cedarlane) was present in the reaction mix, the AURKA concentration was reduced to 2 nM. Reactions were initiated by the addition of 10 μM of ATP and incubated at room temperature for 60 min. Reactions were terminated by transferring 10 μL of the reaction mix to a 384-well white plate (Lumitrec 200, VWR) and adding 10 μL of ADP-Glo™ Reagent (Promega). After a 40-min incubation, 20 μL of kinase detection reagent (Promega) was added and allowed to incubate for an additional 30 min. Luminescence was measured on a Biotek Synergy Neo plate reader (BioTek) using a 1 s integration time. Results were plotted in GraphPad Prism V8.4.2 using a one-site total binding curve fitting analysis.

**Fluorescence polarization binding assay**. FITC-Tpx2$^{1-43}$ and unlabeled Tpx2$^{1-43}$ were synthesized by Proteogenix (Schiltigheim, France). Unlabeled proteins and FITC-Tpx2$^{1-43}$ were mixed in FP buffer (25 mM HEPES pH 7.5, 50 mM NaCl, 10 mM MgCl₂, 0.1 mg/mL BSA (Sigma), 0.01% Brij, 3 mM ADP (Sigma), 2 mM DTT) in a 384-well flat-bottom black plate (Corning 3573) at a final volume of 25 μL. For $K_d$ extraction, FITC-Tpx2$^{1-43}$ probe was kept at 5 nM throughout the experiments and each analyzed protein was added to the concentrations indicated

in each figure. Each data point was derived from a technical duplicate and each binding curve was performed in triplicate. For the competitive displacement assay, FITC-Tpx2[1–43] (5 nM) and AURKA protein concentrations (35 nM) were kept constant throughout experiments, and unlabeled competitor (either Bora or Tpx2) was added to final concentrations indicated in each figure. Fluorescence polarization was measured on a BioTek Synergy Neo plate reader (BioTek) using Gen5 v2.05 software with excitation and emission at 485/528 nm, respectively. Binding graphs and the derived binding constants were generated using GraphPad Prism v8.1.2 and v8.3 (GraphPad).

**Intact protein mass spectrometry**. The liquid chromatography mass spectrometry (LC-MS) system consisted of a 1260 series HPLC connected to 6538 UHD series electrospray ionization quadrupole time-of-flight mass spectrometer (Agilent Technologies, Santa Clara, CA). Approximately 2.5 μL aliquots of protein samples (~0.1 μM) dissolved in a 0.1% formic acid buffer were separated on a 2 mm × 50 mm Sprite C4 column (Higgins Analytical Inc., Mountain View, CA). The mobile phase was composed of A: 0.1% aqueous formic acid and B: acetonitrile (Fisher Scientific Company, Ottawa, ON). The flow rate was 0.25 mL/min and proteins were eluted rapidly using a linear gradient of 5–80% B over 5 min. Mass spectrometry data were collected up to 2500 $m/z$ and raw spectra were deconvoluted using the maximum entropy algorithm of the Agilent MassHunter software version B.06.01. The peak numbering corresponds to the molecular weight of the protein plus the indicated number of phosphate groups (+79.99 Da/phosphate group).

**NMR spectroscopy**. NMR-HSQC spectroscopy spectra (8 scans, 20 min acquisition time) were recorded at 15 °C on a 600-MHz Bruker AVANCE III spectrometers equipped with a 1.7-mm TCI CryoProbe. All NMR samples contained 180 μM of [15]N-Bora[1–224] or [15]N-phospho Bora[1–224] in 25 mM Tris pH 7.5, 150 mM NaCl, 1 mM TCEP and 10% $D_2O$. Data processing was conducted using NMRviewJ and NMRpipe[69], and NMR spectra were analyzed using Analysis[70].

**Preparation of Xenopus extracts**. CSF extracts were obtained from metaphase II-arrested oocytes. Laid MII-arrested oocytes are first dejellied with 2% cysteine solution, extensively rinsed with XB buffer (100 mM KCl, 0.1 mM $MgCl_2$, 50 mM Sucrose, 5 mM NaEGTA, 10 mM HEPES at pH 7.7) and subsequently centrifuged twice for 20 min at 10,000 × $g$ and the cytoplasmic fractions recovered.

Interphase egg extracts were obtained from dejellied unfertilized eggs transferred into MMR solution (25 mM NaCl, 0.5 mM KCl, 0.25 $MgCl_2$, 0.025 mM NaEGTA, 1.25 mM HEPES–NaOH pH 7.7) supplemented with Ca2+ ionophore addition (final concentration 2 μg/mL) and 35 min later recovered and treated as CSF extracts[71]; 1 μL of Interphase or CSF egg extracts was loaded on a polyacrylamide gel and transferred onto N Immobilon-P membranes.

**Immunodepletion Xenopus extracts**. To deplete Xenopus egg extracts, immunoprecipitations were carried out for 20 min at room temperature using 2 μg of affinity-purified antibodies immobilized on 20 μL protein G-Dynabeads (Dynal) and 20 μL of Xenopus egg extracts. When supernatant was used, beads were removed by magnetic racks and supernatants recovered. Two consecutive immunoprecipitations were performed to completely remove endogenous Bora, AURKA, and Gwl proteins. Bora immunoprecipitation was performed using antibodies raised against GST-full length Xenopus Bora whereas Western blot was performed with antibodies against Xenopus-Nter-Bora[27].

**Generation of knock-in cell lines using Cas9/CRISPR method**. To introduce the AID cassette (with a Neon-Green fluorescent tag) at the Bora locus, the guide RNA sequences targeting exon 12 in Bora were as written in the Supplementary Table 3 (Guide #1 and Guide #2).

Each guide was phosphorylated and annealed in a thermocycler then subcloned into pX330-U6-hSpCas9 using BbsI (Addgene #42230) using the primers listed in the Supplementary Table 3 for the Guide #1, and the Guide #2.

The vector pSR75 containing the Bora repair template (7501 bp) was generated by Gibson cloning[72], the repair template contains the two homology arms flanking the Neon-green and the AID cassette followed by a Hygromycin resistance cassette (separated by P2A). The constructs were transfected using Viafect into DLD1 stably expressing OsTIR1. Three days post transfection, cells were trypsinized and plated onto a 10-cm dish and treated with hygromycin (200 μg/mL). Individual clones were isolated, amplified, and screened for homozygosity using PCR and sequencing. Homozygosity was validated using the primers listed in the Supplementary Table 3 (P1 and P2).

For the rescue experiments, the variants of Bora were integrated into the DLD1-Bora-NeonGreen-AID cell line at the AAVS1 locus (Chromosome XIX). Each Bora variants serving as repair templates was sub-cloned into the pMK243 (replacing the OsTir1 cassette). The constructs along with the pX330-U6-HSpCas9 (that contains the gRNA targeting the AAVS1) were transfected using Lipofectamine LTX into the DLD1-Bora-NeonGreen-AID cell line; 48 h after transfection, the cells were split ¼ and treated with Puromycin for selection. Individual clones were isolated, amplified, and screened by PCR and Western blot (upon Doxycyclin treatment).

**Auxin-mediated Bora degradation**. To induce the degradation of Bora-AID, DLD1-Bora-NeonGreen-AID cell line was treated with 500 μM of Auxin (MerckMillipore) from a 280-mM stock solution (50 mg/mL in EtOH). Expression of Bora variants was induced with 0.5 μg/mL Doxycycline (MerckMillipore) for 24 h from a 1 mg/mL stock solution (in DMSO).

**Cell extracts and Western blot analyses**. Whole-cell extracts were prepared from mammalian cells by resuspending cells in 50 mM Tris-HCl at pH 7.5, 1 mM EDTA, 1 mM EGTA, 50 mM sodium fluoride, 0.27 M sucrose, 1% Triton X-100, complete protease inhibitors, and complete phosphatase inhibitor PhosSTOP from Roche. Samples were then snap frozen, thawed on ice, before clarification by centrifugation. Protein concentrations were measured using Pierce BCA Protein assay kit (Thermo Fisher Scientific) and a Spectramax M2 plate reader (Molecular Devices).

To detect protein in cell lysates, the protein samples were separated by SDS-PAGE and transferred onto nitrocellulose membranes. Proteins were detected by Western blot and visualized by treating the blots with ECL (MerckMillipore).

**Immunoprecipitation**. To detect pT210-Plk1 signal in cell extracts, mammalian cells were lysed in 50 mM HEPES pH 7.4, 1 mM $MgCl_2$, 1 mM EGTA, 1% NP40, 1 mM NaF, complete protease inhibitors, and complete phosphatase inhibitor PhosSTOP, incubated on ice for 30 min before clarification by centrifugation. Protein concentrations were measured using Pierce BCA Protein assay kit (Thermo) and a Spectramax M2 plate reader (Molecular Devices). The immunoprecipitation was performed on 1.5 mg of total proteins overnight at 4 °C with 3 μg of anti-Plk1 (Abcam #ab17057). The following day, samples were incubated for 2 h with pre-equilibrated magnetic Protein A/G beads (Ademtech) at 4 °C. Samples were then washed three times with lysis buffer, beads were resuspended in Laemmli buffer and boiled for 10 min. Protein samples were separated by SDS-PAGE and transferred onto nitrocellulose membranes.

**Antibodies**. The antibodies used in this study are the following: anti-human Plk1 (Abcam Cat. #ab17057), anti-human Plk1 (Bethyl Cat. #A300-250A), anti-Bora (Cell Signaling Technologies Cat. #D2B9), anti-Bora (Santa-Cruz Cat. #sc-393741), anti-Phospho-Histone H3 (Ser10) (Cell Signaling Technologies Cat. #9701), anti-Phospho-Plk1 (Thr210) (Cell Signaling, Technologies Cat. #5412), anti-Actin (MP Biomedical Cat. #69100), anti-Tubulin–DM1A (MerckMillipore Cat. #T9026), anti-AURKA (Cell Signaling Technologies Cat. #91590), anti-human Greatwall[73], anti-Xenopus Plx1 (ref. [27]), anti-Phospho-PP1 (Thr320) (Abcam Cat. #ab62334), anti-Phospho-Cdc2 (Tyr15) (Cell Signaling Technologies Cat. #9111), anti-ERK[74], anti-Phospho-ERK (Cell Signaling Technologies Cat. #9106S), anti-Xenopus Bora[27], anti-Cyclin B1 (Cell Signaling Technologies Cat. #D5C10), Peroxidase Goat Anti-Mouse IgG (H+L) (MerckMillipore Cat. #A9917), Peroxidase Goat Anti-Rabbit IgG (H+L) (MerckMillipore Cat. #A0545), TrueBlot ULTRA (Rockland Cat. #18-8817-30). Antibodies were diluted at 1/1000.

**FACS analysis and rescue index**. After harvest, the cells were fixed in ice-cold 70% ethanol, washed and stained with 50 μg/mL propidium iodide (PI) and 50 μg/mL RNase A. FACS analyses were done using a Beckman coulter cyan ADP flow cytometer; 10,000 events were recorded in a gate within PI-Area versus PI-width dot plot, cell cycle phases were determined via flowjo using the Watson pragmatic algorithm.

**Microscopy**. DLD1 cells were seeded 2 days prior to the experiment (80,000 cells per well) on Krystal 24-well 175 m glass-bottom microplates (Wildcat Laboratory Solutions) precoated with bovine fibronectin (Promocell) at 1 mg/cm[2] and imaged in DMEM without phenol red and supplemented with 5% FCS. Imaging was performed using an inverted microscope (Nikon Eclipse TI-E) controlled by Metamorph software and equipped with perfect focus system, fast emission filter wheel (lambda 10-3, Sutter), electron multiplying charge coupled device camera (iXon 3 888 Ultra Andor), Plan Apochromat 20×/NA 0.75 lens, and light-emitting diode (LED)-based illumination system (spectra X-light engine, Lumencor). Filters used for IFP2 were ET620/60x, T660lpxr beamsplitter, and ET700/75m from Chroma. Quantifications were performed using ImageJ software.

**Statistics and reproducibility**. All experiments presented in this manuscript have been repeated at least three times with the exception of SPR and NMR analysis. $K_d$ and $EC_{50}/IC_{50}$ SD values represent deviation between values obtained from three independent experiments.

**Reporting summary**. Further information on research design is available in the Nature Research Reporting Summary linked to this article.

## Data availability
The authors declare that the data supporting the findings of this study are available within the paper and its Supplementary Information files. Source data are provided with this paper.

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

## Acknowledgements

We thank P. Moussounda and R. Servouze (Institut Jacques Monod) for help with media preparation, M. Cobb, E. Nigg, A. Santamaria, M. Kanemaki, F. Zhang, T. Mittag for sharing plasmids and D. Durocher for the recombinant Histone H3 protein. We acknowledge the ImagoSeine core facility of the Institut Jacques Monod, member of IbiSA and the France-BioImaging (ANR-10-INBS-04) infrastructure. NMR spectrometers were funded by the Canada Foundation for Innovation and the NMR Core Facility was supported by the Princess Margaret Cancer Foundation. We acknowledge Matthew Forbes from the AIMS (Advanced Instrumentation for Molecular Structure) Mass Spectrometry Laboratory of University of Toronto (Department of Chemistry) for his help regarding intact mass-spectrometry. Research was supported by the Fondation ARC (PJA 20181207931 to L.T.), the Agence Nationale de la Recherche (AMBRE Project ANR-17- CE13-0011 to P.L., L.T., and G.O.), the Labex EpigenMed (ANR-10-LABX-12-01 to C.A., L.T.), Ligue Nationale Contre le Cancer (programme équipe labelisée to P.L.), the Canadian Cancer Society (CCSRI-Impact 704116 to S.F.), the Canadian Institutes of Health Research (FDN 143277 to S.F.; FRN 414829 to Mai.P.), and the Terry Fox Research Institute (to S.F.). T.N. was supported by postdoctoral fellowships from the "Fondation pour la Recherche Médicale" (SPE20150331777) and from the CIHR (MFE 152464). T.Y. was supported by a post-doctoral fellowship from the LabEx "Who Am I?" (#ANR-11-LABX-0071) and the Université de Paris (IdEx #ANR-18-IDEX-0001) funded by the French Government through its "Investments for the Future" program. J.N. was supported by the Fondation ARC pour la Recherche sur le Cancer and is supported by funding "Dynamic Research" from Agence Nationale de la Recherche (ANR-18-IDEX-0001), Université de Paris excellency initiative (IdEx). V.A.G. is supported by CONACYT (grant CVU 364106). R.S. and D.M. are supported by the National Institute for Child Health and Human Development Intramural project Z01 HD008954.

## Author contributions

Conceptualization, T.N., P.L., and S.F. Methodology, R.S. and D.M. Investigation, T.N., T.Y., V.S., Mai.P., O.S., Mad.P., VH.L., L.N., GS.G., J.N., P.M., VA.G., G.O., C.A., L.T., P.L., and S.F. Writing—original draft, T.N., P.L., and S.F. Writing—review and editing, T.N., P.L., T.Y., Mai.P., R.S., L.N., G.O., L.T., and S.F. Visualization, T.N., T.Y., Mai.P., P.L., and S.F. Supervision, P.L. and S.F. Funding acquisition, T.N., G.O., L.T., P.L., and S.F.

## Competing interests

The authors declare no competing interests.
