## [Peer Review File · Nature Communications]

REVIEWER COMMENTS

Reviewer #1 (Remarks to the Author):

The authors describe that Bora binds specifically to AurA and that this is the mechanism by which CDKs induce phosphorylation of PLK1 at its activation loop along the cell cycle. The results presented in the paper are important. Overall I like the manuscript and the experiments and results presented are of high quality. For example, it becomes very clear from the work by the authors that AurA binds with high affinity to Bora, that the interaction is mediated by two sequences of Bora M1 and M2 located about 100 aa apart, and that the interaction activates AurA in vitro. Also, in experiments using the kinase domain of PLK1, the authors show that that the Bora interaction with AurA is needed for the CDK-dependent phosphorylation of PLK1. The authors further describe that a third motif, a phosphor-motif M3 on Bora acts on AurA possibly by mimicking the activation loop phosphate.

The final model proposed by the authors partly contrasts with a previous model by Fang and colleagues, concluding that "Bora interacts with Plk1 and controls the accessibility of its activation loop for phosphorylation and activation by Aurora A".

Although I like the paper very much, and I find it important, I also find that the claims of the work are not yet fully supported by the data presented and, to be conclusive, further experiments must be performed.

A main concern about the work is related to the conclusion that Bora does not bind to PLK1. The model therefore, disregards a role of the binding of Bora to PLK1 and the possible change in conformation of PLK1 to support the phosphorylation by AurA. Also, there is a need to further support the proposed mechanism by with M1, M2 and M3 participate in the activation of AurA.

1- The authors should conclusively show that Bora does not bind to PLK1, in particular to full length PLK1 and to the PBD of PLK1.

2- In figure 2 A and 2B in NMR experiments the authors show that Bora interacts with AurA but not comparably with the CD of PLK1. The authors should show using the same assay if PLK1 full length interacts with Bora.

(The authors show that Avi-tagged Bora did not pull down PLK1. (suppl fig 2). Based on this, the authors conclude a lack of interaction of Bora with PLK1 full length. However, the NMR and pull down experiments look into two different properties of interactions. The pull down identifies binding that must have low relative OFF-rates (to remain bound during the washes) while the NMR binding in solution is more sensible to interactions. The pull-down assay on its own is not good evidence of a lack of interaction. Possibly even choosing different buffer conditions during the washing could decrease the specific binding proteins, depending on their mode of interaction.)

3- The conclusion "the kinase domain of PLK1 is sufficient for phospho-Bora potentiated phosphorylation by AurA" is correct, but along the conclusions of the work the authors seem to extrapolate outside the actual statement. Currently there is still a doubt if the PBD of PLK1 does not play a role in the Bora induced phosphorylation of the activation loop of PLK1. The fact that a PLK1 construct lacking the PBD can be phosphorylated by Aurora in a CDK-dependent manner does not rule out a role of the PBD.

It is correct that the full length PLK1 is not required to observe the Bora-induced phosphorylation of the kinase domain. However, it does not mean that it is dispensable in the model. It may well be that Bora also binds to PLK to open up the structure, which, upon binding, stabilizes a different conformation -the active conformation- that is the conformation of the isolated cat domain of PLK1. So the studies on PLK1 CD could well bypass the function of Bora on PLK1 FL to "open" the structure. It is an apparent small difference but can change dramatically the conclusions and the validity of previous

work on the model. Of course, this item is related to a previous concern, the possible interaction of Bora with PLK1 FL. I should note that the interaction of Bora with PLK1 has been described in different papers, including one in which the authors set-up a HTS screening system on the interaction between PLK1 with a polypeptide from Bora.

4- Related to the above concern is how the authors draw the PLK1 cartoon of full length and catalytic domain cartoons. The cartoons representing the bean structures of protein kinases are very explicative. However, they can also lead to wrong conclusions, if the simplification of the cartoon avoids depicting important features. In the cartoon of PLK1 FL the Polo Box Domain appears binding at the back of the cat domain. OK. Now, we know that this interaction renders inactive the cat domain; so the cat domain could be drawn in an inactive conformation when the PBD is attached to the cat domain. On the other hand, the isolated cat domain of PLK1 is fully active; so in the cartoon, the isolated cat domain could better be drawn in an active conformation (a conformation different from the cat domain on the -inactive- full length PLK1). This is not a minor point: the difference between FL and forms lacking the PBD can be explained by a different conformation of the CD.

5- The authors suggest that two M1 and M2 identified regions of Bora could interact with AurA at the TPX2 binding site. Given that the structural information of TPX2 is available, the obvious synthesis of Bora polypeptide joining both regions must be done a- to show that it activates AurA to phosphorylate the activation loop of full length PLK1 vs kinase domain of PLK1; b- to confirm that the polypeptide displaces the interaction with TPX2 and c- to show that the polypeptide activates AurA in vitro to phosphorylate an unrelated peptide substrate; c- The role of the phosphor-motif M3 on the activity should also be validated with a proper synthetic polypeptide joining M1, M2 and M3.

6- Previous work considered that Bora would bind to PLK1 and "open" the structure, relieving the interaction of PBD with the cat domain. Since the authors prove in vitro the proposed new model using the isolated cat domain (which is the conformation of the catalytic domain when the PBD interaction is released), this referee still wonders if the "opening" of full length PLK1 could also contribute to the phosphorylation of AurA of the activation loop and therefore to the overall CDK-dependent phosphorylation of PLK1 by AurA. Raab et al. (2018) showed that inhibitors of PLK1, like Behringer Ingelheim (BI)-cpd, binding at the active site, actually "open" the structure of full length PLK1. To exclude a role of "opening" of PLK1 in the model, the authors should check the effect of BI-cpd on the in vitro phosphorylation of full length PLK1 by AurA.

Other points:

7- The text that states EC50 of activation is shown in Fig 2D and E is wrong. Fig 2D and 2 show the binding of Bora and phosphoBora to AurA by SPR.

8- It is not easy to understand what is shown in Fig 3B. What is the difference between the top panel and the second panel where both show Bora and P-Bora. Please, label the panels in the figure to help readers follow the results.

9- The ADP-glow assay does not measure kinase activity. It does not measure the phosphorylation reaction. It measures ATPase activity. It is not the same. Sometimes it can be interpreted as an appropriate read out of activity. But not in this particular context. The authors can only make the stated conclusions if they measure the kinase activity (phosphorylation of a peptide substrate) in parallel.

10- Fig 3D, in comparison to Fig 3c shows very clearly that the activation is observed at much lower concentrations. However, it fails to show the actual concentrations at which the effect is observed. The authors could help to visualize this in some way, for example in an inset of Fig 3D.

11- Along the presentation of the results the authors make a point about the unexpected identified

activity of AurA that is mutated at the activation loop. Is this really unexpected? AurA has been historically produced in bacteria. Is the bacterially produced protein phosphorylated? Is it active in vitro?

12- I have to disagree with the conclusions from Figure 1. The results do not show that the PB1 domain does not mediate the effect: it may well be that the PLK1 has to be in conformation A. The A conformation can be induced by Bora (i.e. by binding to the PBD on full length PLK1) or by the lack of the PB1 domain in the construct comprising only the cat domain.

Reviewer #2 (Remarks to the Author):

The manuscript by Tavernier and colleagues working from the Sicheri lab proposes an interesting new mechanism that might be used to 'allosterically' activate Aurora A directly through the co-factor Bora, which possesses newly described 'TPX2-like' activating motifs. The paper demonstrates that this might be independent of Aurora A T-loop phosphorylation.

It is known that Bora phosphorylated on Cdk sites is essential for PLK1 activation by AURKA at mitotic entry in eukaryote models. However, and what remains confusing in the field, is whether Bora works directly through PLK1 or Aurora A (or both). The molecular mechanism by which Bora contributes to T-loop phosphorylation of Plk1 (which is under the control of Aurora A since T210 lies in an Aurora A motif) is therefore the subject of this paper. The manuscript brings together NMR structural analysis, quantitative biochemistry and immunoblotting, in vivo *Xenopus* egg extract analysis (convincing) and human cell analysis (including CRISPR tagging, Auxin-induced degradation and mitotic index analysis between WT and S112A BORA) to study these phenomena.

Overall, I agree with the title and conclusions of the paper, which rules out a 'simple' model in which phospho-Bora binds to the PLK1 PBD to induce a conformational changes that simply exposes T210 residue to AURKA. Instead, a much more satisfactory outcome emerges, with much mechanistic insight, and I believe that the study demonstrates nicely that phospho-Bora acts by direct Aurora A binding to generate an Aurora A active conformation in the absence of T-loop phosphorylation, which can then support PLK1 phosphorylation at T210. The paper is very high quality overall, and should be published essentially as is, since it will influence new thinking in the field, and is convincing throughout. I have a few questions and comments below that might be useful to address.

Although it might be guessed based on comments below, I am happy to confirm that P. Eyers is the referee, so you will understand my historical, and indeed, current interest in this work (PMID: 32636306 indeed, we recently found that TPX2 controls the redox sensitivity of Aurora A, which may be worth evaluating for BORA also!).

Major points:

1. I think it would be useful early on to explain that T295 (xAurora A) is functionally equivalent to T288 (human Aurora A), also because the Littlepage reference (p.9) is an xAurora A paper. This is because (as I understand it), human-derived Aurora A sequences are used in the *Xenopus* extract experiments? For clarity, it also makes sense (but for the opposite reason) do the same explanation for xPlk1, where T201 is the relevant T-loop phosphorylation site (and confusingly, is similar at first glance to T210 in human PLK1, which is the site used with antibody analysis in other Figures).

2. In particular, the data in Fig 3 and 4 and 7 are convincing, and the finding that T288V promoted Plx1 T-loop phosphorylation and high Cyclin-Cdk activity in *Xenopus* extracts with comparable kinetics to WT Aurora A is a particularly important finding, in my opinion. Related to point 3 below, were T288D or T288E kinetics compared to WT/T288V Aurora A, since these proteins have also been shown

to be 'active' in vitro when bound to TPX2, and more so than T288A (but using *Xenopus* Aurora A, Evers and Maller, 2004)? It has long been a mystery why T288D/E is such a poor phosphomimetic, so there is a really nice opportunity to test if acidic amino acids do something different to Ala/Val here in an ex vivo context. Moreover, I think it worth pointing out in the Discussion that you might predict that the 'phosphomimetic' Aurora A proteins that appear sometimes in the literature might actually be providing good evidence for T-loop independent functions (but TPX2 or BORA-dependent?)

3. Also p.9. Experiments similar to Figure 3 were attempted in 2004, but using xAurora A and xTPX2 to probe the dependence of T-loop for activity-induced effects of TPX2 (Evers and Maller, 2004). I believe that it is worth mentioning these in this context, most critically that T295A xAurora A still has low, but detectable, activity when compared to D274N Aurora A when measured for its ability to phosphorylate TPX2 substrate, (but not Histone H3 alone) in vitro. T295V was also analyzed by MS in a separate piece of work, and shown to be catalytically inactive and unphosphorylated in the absence of TPX2 (PMID: 12885952, MS analysis relevant to Fig S3A shown here). Interestingly, in both cases, TPX2 phosphotransferase activity is much less with T295A/V than with T295D or T295E Aurora A in these assays (Evers and Maller, 2004). These old observations certainly do raise the possibility that TPX2 can also productively switch the active conformation of Aurora A, so that TPX2 can now be phosphorylated when acting as a substrate in a holoenzyme complex independent of T288 equivalent (also subsequently Zorba eLife). This finding is also relevant in this paper, but now in the specific comparative context of BORA/pBORA as an activator. The experimental question is, therefore, whether TPX2 was compared side-by-side with pBORA/BORA, to see if it can indeed activate Aurora A T288V/D/E in trans as measured by PLK1 pT210 (perhaps also using TPX2 as the dual activator/substrate and 32P/band shift. Of equal interest,

- A) Can TPX2 compete with pBORA/BORA to activate Aurora A and
- B) Can pBORA/BORA-bound Aurora A T288V also phosphorylate TACC3 on Ser 558?

I do understand that this second experiment would be complicated by the fact that TACC3 is also an Aurora A activator, rather like TPX2 (the work of Bayliss, which is not cited here, but might be).

Minor points:

1. In the introduction, key references where TPX2 was originally purified from *Xenopus* as an Aurora A activator (PMIDs: 12577065, 12699628 and 16332542) are omitted, and also in the context of protein phosphatases (PMID: 21187329).
2. p.8 'pulled-down', not 'pull-downed'
3. In my version of the PDF, Figure 4D, Figure 7 B, C and Fig S7E all have strange colour artefacts in the Western Blots. This may just be software related, but worth checking.

Reviewer #3 (Remarks to the Author):

Review of "Bora phosphorylation substitutes in trans for Aurora A kinase T-loop phosphorylation to activate Plk1 and to promote mitotic entry" by Tavernier et al.

This is a very well written manuscript that thoroughly describes important molecular level details of phospho-Bora's activation of Aurora A, which subsequently activates Plk1. Key regulatory sites on Bora are defined and the most striking mechanistic finding is that a specific phosphosite in Bora (pS112) compensates for activation loop phosphorylation in Aurora A. Overall the conclusions are well supported by the high-quality data and the findings clarify numerous previously reported results for this system. There are some points that need to be clarified:

- 1) In the discussion, the authors present many good points about the interplay between allosteric

activation of Aurora A (via motifs shared by Bora and Tpx-2) and the more novel mechanism of pS112 within Bora replacing the need for Aurora A activation loop phosphorylation. That said, the physiological significance of compensating for lack of activation loop phosphorylation within Aurora A remains unclear. Can the authors point to specific signaling scenarios where it is known that Aurora A T288 is not phosphorylated? When/where is a lack of Aurora A T-loop phosphorylation encountered? It is understood that these questions might not yet have answers and so the discussion can in this case be improved to more clearly describe the state of knowledge and in turn the importance of what the authors call the "essential phosphosite S112". Finally, the abstract mentions overexpression of mitotic kinases in cancers and yet the link between the Bora focused mechanistic findings and this statement is underdeveloped. Is there evidence of S112 mutation linked to disease?

2) There is an odd sentence on page 9: "...phosphorylation of Bora by Cyclin-Cdk appears to significantly enhance affinity of AURKA to a small degree...". Which is it, a significant or small enhancement? SPR binding data seems to show a bigger difference between Aurora A binding to Bora1-224 versus phosphor-Bora1-224 than the authors indicate (Fig. 2D & E and corresponding Supplemental data). The 52 micromolar affinity derived from SPR for binding of Aurora A to unphosphorylated Bora is probably an overestimate of the binding affinity; the binding curve (Fig. 2D) does not saturate suggesting that the interaction of Aurora A with unphosphorylated Bora is even weaker than reported—and therefore the increase in affinity promoted by Bora phosphorylation may be larger than the 52 versus 18 uM affinity values suggest.

3) Can the authors explain why the binding of AURKA(T288V) to unphosphorylated Bora (Fig. 3E) seems more robust than the corresponding WT AURKA binding to unphosphorylated Bora (Fig. 2D)? Both in terms of calculated affinity and actual response units? After suggesting in point (1) that the difference between wt AURKA binding to phosphorylated and unphosphorylated Bora is perhaps greater than the K_d values indicate, for the AURKA T228V mutant it appears there is little difference between the affinity for phosphorylated and unphosphorylated Bora. Is it expected that the T228V mutation would have this effect?

4) The data shown in Fig. S5 raise questions related to Erk phosphorylation. Figs. S5E, F, G and I all seem to indicate more sites of phosphorylation than expected. Can the authors clarify the mass spectrometry data? In particular the fact that the mass spec data suggests that the important mutant that retains S112 (Fig. S5 panel F) is phosphorylated on multiple sites.

5) Possibly related to point 4 it is unclear why Bora 18-120(S27A, S41A, T52A) does not lead to phosphorylated Plk1 in the presence of WT AURKA (Fig. 5B lane 25).

6) On pages 15-16 the authors describe their efforts to localize the putative phospho-binding site on AURKA. Two binding sites are suggested that seem to share two of the three sidechain defining each. It is therefore not clear why the authors describe the second site as 'remote'. As well, the experimental data only probe 3 of the 4 residues listed (Fig. 6B). Overall this section seems less well developed than other parts of the manuscript.

Reviewer #4 (Remarks to the Author):

Major points:

1. A key finding of the paper is that Bora as an activator of Aurora-A independent of substrate. This conclusion is based on the results of a series of ADP-Glo assays (Figure 3, Figure S3). ADP-Glo measures the concentration of ADP, formed by the hydrolysis of ATP by the kinase. While it is clear that Bora significantly increases the production of ADP by Aurora-A, this also happens whether or not an additional peptide substrate is included (Fig 3C vs Fig S3C). I would therefore draw an alternative

conclusion: that the increased production of ADP in the presence of Bora is due to Bora acting as a substrate for the kinase, thereby enabling a kinase reaction to take place. The authors should present additional experiments to distinguish between their interpretation and mine. One way to do this is an assay that measures phosphorylation of a peptide (or protein substrate) in the presence of increasing amounts of Bora - could be through incorporation of ^{32}P into a substrate, or an immunoblot against a known Aurora-A substrate (e.g. HH3, TACC3).

2. Another key finding of the paper is that phosphorylation of S112 is minimally required for Aurora-A activation (see page 14) - but this conclusion depends on experiments that compare a double mutant (S27A, T52A) with a triple mutant (S27A, T52A, S112A). To draw the conclusion that S112 is minimally required, the experiment (FIG. 5B) should be repeated using Bora with the single S112 mutation.

3. Moreover, several subsequent experiments are done using mutations based on a background of the S27A, T52A mutations (e.g. p.15). This complicates the interpretation of experiments only slightly, and so I don't think any other experiments need to be repeated with single mutants. However, it is somewhat confusing, and justification for using this mutant should be clearly stated (is there a technical reason? Or was decided to remove the potential for complications due to phosphorylation at these positions by other kinases?)

4. Mutational analysis of Aurora-A to identify the basic residue(s) that respond to Bora phosphorylation is neat and the results are clear (p.16). There is, however, a possibility that one or more of these mutations affects Aurora-A activity directly (e.g. by destabilising the protein), and it would be reassuring to have some evidence that the mutated Aurora-A proteins are stable & active. This should ideally be done experimentally (e.g. thermal stability, proteins retain binding to Bora). You could also (or instead) refer to data in the McIntyre (2017) paper in your reference list, in which the R286 and R180 mutations affects the binding of TPX2 to Aurora-A phosphorylated on T288, but retain full binding to the unphosphorylated protein.

5. The authors present a model of Bora and TPX2 binding to the same sites on Aurora-A. Testing whether TPX2 and Bora compete for binding Aurora-A (through immunoprecipitation or "pulldowns" using recombinant proteins) would be a straightforward experiment that would strengthen the conclusions of the paper considerably, with minimal effort.

Minor points

1. The purpose of showing the ponceau staining on figure 8e isn't clear to me. I couldn't see any obvious differences between the samples, so is this simply to show equivalent loading of samples on the blot? Please explain!

I am happy to sign the review.
Prof. Richard Bayliss, University of Leeds, UK.

DETAILED RESPONSE TO REVIEWER COMMENTS (author comments in blue)

Reviewer #1 (Remarks to the Author):

The authors describe that Bora binds specifically to AurA and that this is the mechanism by which CDKs induce phosphorylation of PLK1 at its activation loop along the cell cycle. The results presented in the paper are important. Overall I like the manuscript and the experiments and results presented are of high quality. For example, it becomes very clear from the work by the authors that AurA binds with high affinity to Bora, that the interaction is mediated by two sequences of Bora M1 and M2 located about 100 aa apart, and that the interaction activates AurA in vitro. Also, in experiments using the kinase domain of PLK1, the authors show that that the Bora interaction with AurA is needed for the CDK-dependent phosphorylation of PLK1. The authors further describe that a third motif, a phosphor-motif M3 on Bora acts on AurA possibly by mimicking the activation loop phosphate.

The final model proposed by the authors partly contrasts with a previous model by Fang and colleagues, concluding that “Bora interacts with Plk1 and controls the accessibility of its activation loop for phosphorylation and activation by Aurora A”.

Although I like the paper very much, and I find it important, I also find that the claims of the work are not yet fully supported by the data presented and, to be conclusive, further experiments must be performed.

A main concern about the work is related to the conclusion that Bora does not bind to PLK1. The model therefore, disregards a role of the binding of Bora to PLK1 and the possible change in conformation of PLK1 to support the phosphorylation by AurA.

We apologize for our lack of clarity on this issue. It is well proven that Bora binds to Plk1. The interaction between Plk1 and Bora has been reported by several different labs including our group. Bora is found in complex with Plk1 in *C. elegans* embryos (Noatynska et al., 2010; Tavernier et al., 2015), *Xenopus* egg extracts (T. Lorca) and in human cells (Chan et al., 2008; Seki et al., 2008b; Macurek et al., 2008; Thomas et al., 2016). Thus, with respect to full-length proteins assayed in cells or in vivo, Bora binds to Plk1.

With regards to the nature of how the two proteins interact, full length Bora interacts with the PBD domain of Plk1 via an essential phosphorylated Polo-docking site (S252) generated by Cyclin-Cdk activity (Chan et al., 2008). This interaction is also essential for the Plk1-dependent proteasomal degradation of Bora mediated by the E3 ligase SCF/CRL1^{BT^{rcp}} (Chan et al., 2008; Seki et al., 2008a).

The binding of phosphorylated Polo-docking site peptides to Plk1 was also shown to drive Plk1 from a closed auto inhibited conformation to an open activated conformation (Johnson et al., 2008; Xu et al., 2013; Zhu et al., 2016). Thus, it is fully possible that the binding of the pS252 site of pBora to the PBD of full length Plk1 would affect the accessibility of the Plk1 T-loop to phosphorylation by AURKA.

However, as shown in the Figure 1C of our submitted manuscript, Bora¹⁻²²⁴ lacking the Polo-docking motif at S252 (and thus defective for binding to the PDB domain of Plk1) efficiently

promotes the phosphorylation of full-length Plk1 on its T-loop (while in its closed conformation) and also phosphorylation of the isolated kinase domain of Plk1 (in its open conformation) by AURKA. This result highlights an additional, novel and unappreciated mechanism beyond that involving a direct interaction of pBora with the PBD of Plk1. This is the focus of our current work.

Likewise, in *Xenopus* egg extracts, Bora¹⁻²²⁴ (lacking the S252 polo docking site) can rescue Bora-depletion to activate full-length Plk1 (Vigneron et al., 2018). Taken together, these results suggest that the transition of Plk1 from a closed to an open structure in response to PBD binding to a phospho-epitope in Bora is not the sole mechanism by which phospho-Bora acts. Rather, we have discovered a novel mechanism at play in which phospho-Bora binds and allosterically activates the kinase domain of AURKA – an effect that is most pronounced when the T-loop of AURKA lacks phosphorylation. We have revised the manuscript text to make these points clear.

Also, there is a need to further support the proposed mechanism by with M1, M2 and M3 participate in the activation of AurA.

We agree with the reviewer that this is an interesting area for further investigation. In response to the reviewer and in an attempt to generate a minimal pBora fragment that would be conducive to co-crystallization with AURKA (an effort we have failed in so far), we chemically synthesized a minimal polypeptide linking M1 and M2-M3 motifs via a five residue glycine [5G] linker. As described in response to point #5 below, this polypeptide is not sufficient to activate the ATPase activity of AURKA^{T288V} or its phosphorylation of Plk1 in vitro. Furthermore, it is deficient for robust binding to AURKA as assessed by a competitive displacement assay.

We suspect that the length of the artificial linker between the M1 and the M2/M3 motifs is geometrically suboptimal or that there are other important determinants between the M1 and M2/3 motifs that are missing from the construct. Clearly, further work is required to define a minimal polypeptide of pBora that faithfully recapitulates the regulatory functions in question but we believe it is beyond the scope of the present study.

However, we believe that the competition binding experiments requested by all four reviewers (New Figure 6), demonstrates clearly that Bora and Tpx2 compete for binding to AURKA, which strengthens our conclusions that both proteins use related binding motifs to engage the same surface of AURKA.

Furthermore, in a new experiment shown in Figure 6F, we demonstrate that mutations in motifs M1 and M2 of pBora¹⁻²²⁴ greatly perturb binding to AURKA. Together with new mutational studies showing the essentiality of phosphorylation on Ser112 in motif M3 for activation of AURKA (Supplementary Figure 5C), these data provide strong evidence for the involvement of motifs M1 and M2 (which are highly similar to functional elements in Tpx2) and the phospho motif of M3 in Bora's ability to bind and activate AURKA.

1- The authors should conclusively show that Bora does not bind to PLK1, in particular to full length PLK1 and to the PBD of PLK1.

We apologize as we were not sufficiently clear with our descriptions of prior knowledge in our original submission. As noted above, full-length Bora is well appreciated to bind full length Plk1.

We have revised the manuscript text to make this point clear. These clarifying textual changes also address points 2b, 3a, 3b, 6 and 12 below.

2a- In figure 2 A and 2B in NMR experiments the authors show that Bora interacts with AurA but not comparably with the CD of PLK1. The authors should show using the same assay if PLK1 full length interacts with Bora.

We apologize as we were not sufficiently clear with our descriptions of prior knowledge in our original submission. As noted above, full-length Bora is well appreciated to bind full length Plk1. We do not contest this point, and as such we have revised the manuscript text to make this point more clear.

2b-The authors show that Avi-tagged Bora did not pull down PLK1. (suppl fig 2). Based on this, the authors conclude a lack of interaction of Bora with PLK1 full length. However, the NMR and pull down experiments look into two different properties of interactions. The pull down identifies binding that must have low relative OFF-rates (to remain bound during the washes) while the NMR binding in solution is more sensible to interactions. The pull-down assay on its own is not good evidence of a lack of interaction. Possibly even choosing different buffer conditions during the washing could decrease the specific binding proteins, depending on their mode of interaction.

We apologize for the misunderstanding. We do not want to claim that full-length Bora does not bind to full length Plk1. We note however that in Supplementary Figure 2, we used the isolated kinase domain of Plk1. We have revised the manuscript text to make this point more clear.

3a- The conclusion "the kinase domain of PLK1 is sufficient for phospho-Bora potentiated phosphorylation by AurA" is correct, but along the conclusions of the work the authors seem to extrapolate outside the actual statement. Currently there is still a doubt if the PBD of PLK1 does not play a role in the Bora induced phosphorylation of the activation loop of PLK1. The fact that a PLK1 construct lacking the PBD can be phosphorylated by Aurora in a CDK-dependent manner does not rule out a role of the PBD.

We entirely agree with the reviewer and we apologize if we were not sufficiently clear. We did not want to extrapolate outside the actual statement.

3b-It is correct that the full length PLK1 is not required to observe the Bora-induced phosphorylation of the kinase domain. However, it does not mean that it is dispensable in the model. It may well be that Bora also binds to PLK to open up the structure, which, upon binding, stabilizes a different conformation -the active conformation- that is the conformation of the isolated cat domain of PLK1. So the studies on PLK1 CD could well bypass the function of Bora on PLK1 FL to "open" the structure. It is an apparent small difference but can change dramatically the conclusions and the validity of previous work on the model. Of course, this item is related to a previous concern, the possible interaction of Bora with PLK1 FL.

We do not contest this point and thus we have revised the manuscript text accordingly.

3c- I should note that the interaction of Bora with PLK1 has been described in different papers, including one in which the authors set-up a HTS screening system on the interaction between PLK1 with a polypeptide from Bora.

The reviewer is correct. As noted above, the interaction between Bora and PLK1 has been described previously. In the manuscript noted by the reviewer (we assume he is referring to Lee et al BioChip 2013) the authors did not mention which Bora peptide they used in their assay but we presume it corresponds to a peptide spanning the S252 Polo-docking site. In the second paper, Bora binding to Plk1 is shown to occur via interaction between the PBD and the S252 Polo-docking site of Bora (Chan et al., 2008).

4- Related to the above concern is how the authors draw the PLK1 cartoon of full length and catalytic domain cartoons. The cartoons representing the bean structures of protein kinases are very explicative. However, they can also lead to wrong conclusions, if the simplification of the cartoon avoids depicting important features. In the cartoon of PLK1 FL the Polo Box Domain appears binding at the back of the cat domain. OK. Now, we know that this interaction renders inactive the cat domain; so the cat domain could be drawn in an inactive conformation when the PBD is attached to the cat domain. On the other hand, the isolated cat domain of PLK1 is fully active; so in the cartoon, the isolated cat domain could better be drawn in an active conformation (a conformation different from the cat domain on the -inactive- full length PLK1). This is not a minor point: the difference between FL and forms lacking the PBD can be explained by a different conformation of the CD.

The reviewer is correct, Plk1 is auto-inhibited in its resting state and we made note of this in the introduction of our manuscript. T-loop phosphorylation and binding of the PBD to phosphopeptides relieves this autoinhibition in part (Archambault et al., 2015; Johnson et al., 2008; Xu et al., 2013; Zhu et al., 2016). This being the case, we have modified the drawing in Figure 1B to show that T-loop conformation and phosphorylation is linked to and disrupts the PBD/KD interaction (Kachaner et al., 2017).

5- The authors suggest that two M1 and M2 identified regions of Bora could interact with AurA at the TPX2 binding site. Given that the structural information of TPX2 is available, the obvious synthesis of Bora polypeptide joining both regions must be done

a- to show that it activates AurA to phosphorylate the activation loop of full length PLK1 vs kinase domain of PLK1;

b- to confirm that the polypeptide displaces the interaction with TPX2 and

c- to show that the polypeptide activates AurA in vitro to phosphorylate an unrelated peptide substrate;

The role of the phosphor-motif M3 on the activity should also be validated with a proper synthetic polypeptide joining M1, M2 and M3.

We agree with the reviewer that this is an interesting area for further investigation.

In response to the reviewer's comment and in an attempt to generate a minimal pBora fragment that would be conducive to co-crystallization with AURKA, we chemically synthesized a minimal polypeptide linking M1 and M2-M3 motifs via a five-residues [5G] glycine linker (denoted pBora^{M1 M2 M3}) (panel A). Using a newly developed competitive displacement assay with FITC-Tpx2¹⁻⁴³ as a binding probe, we show that unlike pBora¹⁻²²⁴, pBora^{M1 M2 M3} was a poor competitive binder (worse than non-phosphorylated Bora 1-224) (panel B). Consistent with its poor ability to bind

AURKA^{T288V}, pBora^{M1 M2 M3} did not activate AURKA kinase activity as assessed by the ADP-Glo ATPase assay (panel C) or as assessed by phosphorylation of Plk1 in vitro (panel D).

A: Design of a minimal pBora fragment containing the M1, M2, and M3 motifs (pBora^{M1-M2-M3}). pBora^{M1 M2 M3} is composed of the M1 motif (residues 22-35) fused to the M2 and M3 motifs (residues 100 to 113) with a 5 residue polyglycine linker (5G). B: Competition displacement assay for the binding of FITC-Tpx2¹⁻⁴³ to AURKA^{T288V} by Bora¹⁻²²⁴, pBora¹⁻²²⁴, or pBora^{M1 M2 M3}. C: Activation of AURKA^{T288V} kinase activity by phospho-Bora¹⁻²²⁴ and pBora^{M1-M2-M3} as assessed using the ADP-Glo assay with Kemptide as substrate. Displayed data points and EC₅₀ values represent the average luminescence for each reaction condition ± s.d. RLU: relative light unit.

D: Western blot analysis of 2 step kinase reactions carried out with pBora¹⁻²²⁴ (1X = 0.4 μM) or pBora^{M1-M2-M3} (1X = 0.4 μM) in the presence of Plk1^{K82R} KD substrate and AURKA^{T288V} (step 2). Blots were probed with antibodies to Bora, AURKA, and phosphoT210 Plk1 or pan Plk1 as indicated (from top to bottom).

We suspect that the length of the artificial linker between the M1 and the M2/M3 motifs is geometrically suboptimal or that there are other important determinants between the M1 and M2/3 motifs that are missing from the construct. Clearly, further work is required to define a minimal polypeptide of pBora (smaller than the current minimal construct pBora¹⁸⁻¹²⁰) that faithfully recapitulates the regulatory functions in question but we believe it is beyond the scope of the present study.

However, we believe that the competition binding experiments requested by all four reviewers (New Figure 6 – also described above), demonstrates clearly that Bora and Tpx2 compete for binding to AURKA, which strengthens our claim that both proteins use related binding motifs to engage the same surface of AURKA.

Furthermore, in a new experiment shown in Figure 6F, we demonstrate that mutations in motifs M1 and M2 of pBora¹⁻²²⁴ greatly perturb binding to AURKA. Together with new mutational studies showing the essentiality of phosphorylation on Ser112 in motif M3 for the activation of AURKA (Supplementary Figure 5C), these data provide strong evidence for the requirements of motifs M1 and M2 (which are highly similar to functional elements in Tpx2) and phospho motif M3 for Bora's ability to bind and activate AURKA in its dephosphorylated state.

6- Previous work considered that Bora would bind to PLK1 and “open” the structure, relieving the interaction of PBD with the cat domain. Since the authors prove in vitro the proposed new model using the isolated cat domain (which is the conformation of the catalytic domain when the PBD interaction is released), this referee still wonders if the “opening” of full length PLK1 could also contribute to the phosphorylation of AurA of the activation loop and therefore to the overall CDK-dependent phosphorylation of PLK1 by AurA. Raab et al. (2018) showed that inhibitors of PLK1, like Behringer Ingelheim (BI)-cpd, binding at the active site, actually “open” the structure of full length PLK1. To exclude a role of “opening” of PLK1 in the model, the authors should check the effect of BI-cpd on the in vitro phosphorylation of full length PLK1 by AurA.

We do not dispute a potential role for direct binding of Bora to PLK1 in promoting the phosphorylation of Plk1 by AURKA. We have modified the test to make this point clear. As we do not exclude a role for Bora in the opening up of Plk1 in our model, we have not attempted the suggested experiment.

However, as discussed above, we note that phospho-Bora¹⁻²²⁴, which is defective for binding to the PBD of Plk1 due to the absence of the critical S252 Polo-docking motif, is still able to activate AURKA phosphorylation of full-length Plk1 on its T-loop. This result indicates that even in the absence of binding to the PBD of Plk1 (and thus in opening up of PLK1's structure) that phospho-Bora can promote AURKA phosphorylation of Plk1.

Other points:

sous la co-tutelle de

INSTITUT JACQUES MONOD
Unité mixte de recherche du CNRS
(UMR7592)
et de l'université Paris Diderot

Bâtiment Buffon - 15 rue Hélène
Brion
75205 Paris cedex 13 - France

Tél. : +33 (0)1 57 27 80 89
Fax : +33(0)1 57 27 80 87
lionel.pintard@ijm.fr

7- The text that states EC₅₀ of activation is shown in Fig 2D and E is wrong. Fig 2D and 2 show the binding of Bora and phosphoBora to AurA by SPR.

We apologize for the lack of clarity. We presented data on AURKA activation by Bora and pBora in Figure 3C and then referred to the results of the SPR-binding experiments (Figures 2D and 2E) without making references to the specific figures. We have modified the text to read as follows:

“The EC₅₀ value for potentiation was lower for phospho-Bora relative to Bora (0.19 +/- 0.1 μ M vs 1.2 +/- 0.7 μ M respectively) (Figure 3C), consistent with the tighter binding affinity discerned by SPR (Figure 2D, 2E)”.

8- It is not easy to understand what is shown in Fig 3B. What is the difference between the top panel and the second panel where both show Bora and P-Bora. Please, label the panels in the figure to help readers follow the results.

For the in vitro kinase experiments presented in Figures 1C, 3B, 5B, 5D, 7C, S2C, S3B, S4, S5, there are two distinct steps to the reactions. In the first step, we pre-phosphorylate Bora with a proline-directed kinase (either Cyclin A2-Cdk2 or ERK). In the second step, we add the product of our phosphorylation reactions in step 1 (ie phospho-Bora) to AURKA and Plk1. During the second step, Bora becomes further phosphorylated by AURKA (as evidenced by its mobility shift). The top panel corresponds to the Western blot analysis of the first step while the second panel corresponds to the Western blot analysis of the second step. We have now included a color code to better highlight step1 (yellow) and step2 (magenta) and to better guide readers through the figures.

9- The ADP-glow assay does not measure kinase activity. It does not measure the phosphorylation reaction. It measures ATPase activity. It is not the same. Sometimes it can be interpreted as an appropriate read out of activity. But not in this particular context. The authors can only make the stated conclusions if they measure the kinase activity (phosphorylation of a peptide substrate) in parallel.

The reviewer is correct in that the ADP-glow assay does not directly measure the transfer of phosphate to a substrate but instead measures the generation of ADP from ATP. The ATP consumed by the active site of AURKA can be due to the transfer of phosphate from ATP to a hydroxyl bearing side chain in a substrate (i.e. kinase activity) or to water (ie hydrolysis) or to a combination of the two.

We note that when Kemptide peptide substrate is present in the ADP-Glo reactions, the ATPase activity of AURKA is 1.5 to 2-fold higher than when it is absent (Figure 3C). This suggests that a major proportion of the ATPase signal measured reflects the kinase activity of AURKA rather than its hydrolysis activity. Furthermore, as observed in our analysis with Plk1 as a substrate (and as noted by Reviewer 4 in point 1), Bora itself is a substrate of AURKA (Figure 1 see mobility shift in Bora between lanes 1 versus 4 and lanes 2 versus 5), and thus the ATPase activity observed in ADP-Glo reactions in the absence of Kemptide may also reflect some level kinase activity rather than just ATPase activity.

Notwithstanding, we have edited the manuscript text to clarify the point in question. Specifically, we now state ATPase activity rather than kinase activity in reference to the ADP-Glo assay. In addition, to further our claim that pBora activates the kinase activity of AURKA towards substrates, we have performed experiments using a third substrate (in addition to Kemptide and Plk1), namely Histone H3 with Ser10 as the monitored phospho-acceptor site and see consistent results (New Figure S4 panel A).

10- Fig 3D, in comparison to Fig 3c shows very clearly that the activation is observed at much lower concentrations. However, it fails to show the actual concentrations at which the effect is observed. The authors could help to visualize this in some way, for example in an inset of Fig 3D.

Figure 3 is dense with multiple panels. Due to space constrains we can't include an inset for each graph. However, we do list the EC_{50} values on each graph to quantitatively clarify the issue in question. In addition, we now provide all the quantitative data related to the binding assays (SPR and Fluorescence Polarization) and to the ADP-Glo ATPase assay in the new Table 1.

11- Along the presentation of the results the authors make a point about the unexpected identified activity of AurA that is mutated at the activation loop. Is this really unexpected? AurA has been historically produced in bacteria. Is the bacterially produced protein phosphorylated? Is it active in vitro?

The finding that pBora specifically activates AURKA^{T288V} is completely unexpected.

The majority of protein kinases minimally require phosphorylation on a site within a flexible regulatory element in their kinase domains termed the activation segment. This event transitions the activation segment from a non-productive conformation in which the active site is malformed to a productive conformation in which the active site is competent for binding peptide substrate and for catalysis. These kinases are recognizable from protein kinases that do not require activation segment phosphorylation for their kinase activity by the presence of an arginine active site residue within their 'HRD' motifs. This arginine interacts with the phospho-motif in the activation segment, providing a direct conduit by which activation segment phosphorylation affects catalytic function.

AURKA produced in E coli is active because, like many but not all proteins kinases, it can autophosphorylate its own activation segment (at sufficiently high enzyme concentration) (Bayliss et al., 2003). Upon dephosphorylation of AURKA, the enzyme returns to a low state of catalytic activity, which can be reactivated by incubation of the enzyme with ATP at sufficiently high enzyme concentration for a sufficient period of time.

Using intact mass spectrometry, we showed that wild type AURKA produced in E. coli is autophosphorylated on many sites (reflecting an active form) whereas the AURKA T288V mutant like the AURKA D274N mutant (lacking the essential catalytic base) produced in E. coli are completely devoid of phosphorylation (Supplementary Figure S3).

We also showed that in the absence of intermolecular regulatory factors (TPX2 and pBora) that the AURKA^{T288V} mutant is devoid of kinase activity in contrast to AURKA^{WT}.

12- I have to disagree with the conclusions from Figure 1. The results do not show that the PB1 domain does not mediate the effect: it may well be that the PLK1 has to be in conformation A. The A conformation can be induced by Bora (i.e. by binding to the PBD on full length PLK1) or by the lack of the PB1 domain in the construct comprising only the cat domain.

As noted above, we do not rule out a role for pBora in binding to the Plk1 PBD, and thus we have modified our conclusions from Figure 1 accordingly.

Reviewer #2 (Remarks to the Author):

The manuscript by Tavernier and colleagues working from the Sicheri lab proposes an interesting new mechanism that might be used to 'allosterically' activate Aurora A directly through the co-factor Bora, which possesses newly described 'TPX2-like' activating motifs. The paper demonstrates that this might be independent of Aurora A T-loop phosphorylation.

It is known that Bora phosphorylated on Cdk sites is essential for PLK1 activation by AURKA at mitotic entry in eukaryote models. However, and what remains confusing in the field, is whether Bora works directly through PLK1 or Aurora A (or both). The molecular mechanism by which Bora contributes to T-loop phosphorylation of Plk1 (which is under the control of Aurora A since T210 lies in an Aurora A motif) is therefore the subject of this paper. The manuscript brings together NMR structural analysis, quantitative biochemistry and immunoblotting, in vivo *Xenopus* egg extract analysis (convincing) and human cell analysis (including CRISPR tagging, Auxin-induced degradation and mitotic index analysis between WT and S112A BORA) to study these phenomena.

Overall, I agree with the title and conclusions of the paper, which rules out a 'simple' model in which phospho-Bora binds to the PLK1 PBD to induce a conformational changes that simply exposes T210 residue to AURKA. Instead, a much more satisfactory outcome emerges, with much mechanistic insight, and I believe that the study demonstrates nicely that phospho-Bora acts by direct Aurora A binding to generate an Aurora A active conformation in the absence of T-loop phosphorylation, which can then support PLK1 phosphorylation at T210. The paper is very high quality overall, and should be published essentially as is, since it will influence new thinking in the field, and is convincing throughout. I have a few questions and comments below that might be useful to address.

Although it might be guessed based on comments below, I am happy to confirm that P. Eyers is the referee, so you will understand my historical, and indeed, current interest in this work (PMID: 32636306 indeed, we recently found that TPX2 controls the redox sensitivity of Aurora A, which may be worth evaluating for BORA also!).

We thank Professor P. Eyers for his insightful comments and for pointing this connection, which we deem worthwhile investigating in the near future.

Major points:

1. I think it would be useful early on to explain that T295 (xAurora A) is functionally equivalent to T288 (human Aurora A), also because the Littlepage reference (p.9) is an xAurora A paper. This is because (as I understand it), human-derived Aurora A sequences are used in the *Xenopus* extract experiments? For clarity, it also makes sense (but for the opposite reason) do the same explanation for xPlk1, where T201 is the relevant T-loop phosphorylation site (and confusingly, is similar at first glance to T210 in human PLK1, which is the site used with antibody analysis in other Figures).

As suggested, we have revised the text to clarify that T295 in *Xenopus* AURKA is functionally equivalent to T288 in human AURKA.

2. In particular, the data in Fig 3 and 4 and 7 are convincing, and the finding that T288V promoted Plx1 T-loop phosphorylation and high Cyclin-Cdk activity in *Xenopus* extracts with comparable kinetics to WT Aurora A is a particularly important finding, in my opinion. Related to point 3 below, were T288D or T288E kinetics compared to WT/T288V Aurora A, since these proteins have also been shown to be 'active' in vitro when bound to TPX2, and more so than T288A (but using *Xenopus* Aurora A, Eyers and Maller, 2004)?

It has long been a mystery why T288D/E is such a poor phosphomimetic, so there is a really nice opportunity to test if acidic amino acids do something different to Ala/Val here in an ex vivo context.

The reviewer raises an interesting question that has relevance to many protein kinases. As suggested, we have tested whether the phospho-mimic mutant AURKA^{T288D} behaves more like the down-regulated state mimic AURKA^{T288V} or the activated (auto-phosphorylated) form of the enzyme AURKA^{WT} in our in vitro kinase assay.

As shown in new Supplementary Figure 4B, AURKA^{T288D} behaves most similar to AURKA^{T288V} as both are unable to phosphorylate Plk1 T210 in the presence of Bora¹⁻²²⁴ but are strongly able to phosphorylate Plk1 T210 in the presence of phosphoBora¹⁻²²⁴. For comparison, AURKA^{WT} could be activated to phosphorylate Plk1 T210 by either Bora¹⁻²²⁴ or phosphoBora¹⁻²²⁴. However, the potentiation effect of phosphoBora¹⁻²²⁴ on AURKA^{WT} was not as strong as observed on AURKA^{T288V} or AURKA^{T288D}. Thus, the phosphomimetic mutant AURKA^{T288D} is not a faithful and/or useful surrogate for the T-loop phosphorylated 'active' form of AURKA – at least not in the context of its activation by Bora/phosphoBora.

Moreover, I think it worth pointing out in the Discussion that you might predict that the 'phosphomimetic' Aurora A proteins that appear sometimes in the literature might actually be providing good evidence for T-loop independent functions (but TPX2 or BORA-dependent?)

This is a very good point and indeed our new results along with published data suggest that AURKA has functions that are not dependent on the phosphorylation of its T-loop. We have included the following sentence in the discussion to highlight this point:

“Our results also show that the phosphomimetic AURKA^{T288D} mutant commonly used as a proxy for the activate form of AURKA actually behaves in our in vitro kinase assay more like the constitutive inactive mutant AURKA^{T288V}. This once again highlights the need for caution in the use and interpretation of phospho-mimetic mutations to probe protein kinase functions in biological contexts”.

3. Also p.9. Experiments similar to Figure 3 were attempted in 2004, but using xAurora A and xTPX2 to probe the dependence of T-loop for activity-induced effects of TPX2 (Eyers and Maller, 2004). I believe that it is worth mentioning these in this context, most critically that T295A xAurora A still has low, but detectable, activity when compared to D274N Aurora A when measured for its ability to phosphorylate TPX2 substrate, (but not Histone H3 alone) in vitro. T295V was also analyzed by MS in a separate piece of work, and shown to be catalytically inactive and unphosphorylated in the absence of TPX2 (PMID: 12885952, MS analysis relevant to Fig S3A shown here). Interestingly, in both cases, TPX2 phosphotransferase activity is much less with

T295A/V than with T295D or T295E Aurora A in these assays (Eyers and Maller, 2004). These old observations certainly do raise the possibility that TPX2 can also productively switch the active conformation of Aurora A, so that TPX2 can now be phosphorylated when acting as a substrate in a holoenzyme complex independent of T288 equivalent (also subsequently Zorba eLife). This finding is also relevant in this paper, but now in the specific comparative context of BORA/pBORA as an activator. The experimental question is, therefore, whether TPX2 was compared side-by-side with pBORA/BORA, to see if it can indeed activate Aurora A T288V/D/E in trans as measured by PLK1 pT210 (perhaps also using TPX2 as the dual activator/substrate and 32P/band shift. Of equal interest,

As suggested, we have compared Bora, pBora and Tpx2 side by side in their ability to activate AURKA^{T288V} using the ADP-Glo ATPase assay in the presence of Kemptide as a substrate (new Figure 6G, H). Whereas pBora strongly activated AURKA^{T288V}, Tpx2¹⁻⁴³ (like non-phosphorylated Bora) had no activating effect on AURKA^{T288V} in this assay.

A) Can TPX2 compete with pBORA/BORA to activate Aurora A and

This excellent question was raised by all four reviewers. In response, we first developed a fluorescence polarization-based assay to detect binding of AURKA to FITC-labeled Tpx2¹⁻⁴³. In new Figure 6B and Figure 6D we show that AURKA^{WT} and AURKA^{T288V} bind to FITC-Tpx2¹⁻⁴³ with a Kd = 0.009 μ M and a Kd = 0.015 μ M, respectively.

We then show that unlabeled Tpx2¹⁻⁴³, pBora and Bora to varying degrees can all competitively displace FITC-Tpx2¹⁻⁴³ from binding to AURKA^{WT} and AURKA^{T288V} (Figure 6C, E, F).

Interestingly, Tpx2¹⁻⁴³ is a better competitor than phosphoBora at displacing FITC-Tpx2 from AURKA^{WT} indicating that Tpx2 is a better binder than pBora to AURKA^{WT}.

In contrast pBora is better competitor than Tpx2 at displacing FITC-Tpx2 from AURKA^{T288V} indicating that pBora is a better binder than Tpx2 to AURKA^{T288V}.

These quantitative biophysical observations confirm that Tpx2 and pBora compete for binding to the same surfaces of AURKA with variable affinities dependent on the phosphorylation status of both AURKA and Bora.

We have extended these binding studies to compare the effects of Tpx2, Bora¹⁻²²⁴ and pBora¹⁻²²⁴ on AURKA kinase activity using the ADP-Glo assay (Figure 6G, H). Interestingly, although Tpx2¹⁻⁴³ can bind to AURKA^{T288V} (albeit with diminished affinity relative to AURKA^{WT}), it displayed little ability to activate AURKA^{T288V} kinase activity.

Exploiting this behaviour, and the fact that Tpx2¹⁻⁴³ and pBora¹⁻²²⁴ compete for binding to the same surfaces on AURKA, we then showed that titration of Tpx2¹⁻⁴³ (and also Bora¹⁻²²⁴) can competitively abrogate the ability of pBora¹⁻²²⁴ to activate AURKA^{T288V} kinase activity (Figure 6I).

Together, our biophysical binding and enzymology studies unequivocally demonstrate that Bora, phosphoBora and Tpx2 compete for binding to the same surfaces of AURKA. These observations echo previous observations showing that Bora depletion in human cells increases the association

of Tpx2 with AURKA on the mitotic spindle while overexpression of N-terminal region of Bora [Bora¹⁻³⁷⁹] prevents co-immunoprecipitation of Tpx2 with AURKA (Chan et al., 2008).

B) Can pBORA/BORA-bound Aurora A T288V also phosphorylate TACC3 on Ser 558? I do understand that this second experiment would be complicated by the fact that TACC3 is also an Aurora A activator, rather like TPX2 (the work of Bayliss, which is not cited here, but might be).

We thank the reviewer for pointing out this omission. We have added the missing citation to the revised manuscript.

Instead of using TACC3 as substrate, which as noted by the reviewer is also an AURKA activator and hence would complicate the interpretation of results, we performed new experiments using Histone H3 as a substrate with phosphorylation on Ser10 as a readout. In new Supplementary Figure 4B we show that pBora¹⁻²²⁴ can activate AURKA^{T288V} phosphorylation of Histone H3 on S10. Taken together, our data indicate that pBora activates non-phosphorylated AURKA kinase activity against at least three different substrates.

Minor points:

1. In the introduction, key references where TPX2 was originally purified from *Xenopus* as an Aurora A activator (PMIDs: 12577065, 12699628 and 16332542) are omitted, and also in the context of protein phosphatases (PMID: 21187329).

We thank the reviewer for pointing out these omissions. We have added the citations in question to our revised manuscript.

2. p.8 'pulled-down', not 'pull-downed'

We have corrected this error.

3. In my version of the PDF, Figure 4D, Figure 7 B, C and Fig S7E all have strange colour artefacts in the Western Blots. This may just be software related, but worth checking.

In the original submission, we included a color code to help guide the readers through the figures in question. Specifically, the Western blots indicative of interphase were artificially (using Adobe illustrator) coloured red and the Western blots indicative of mitosis were coloured green. Instead of coloring the blots themselves (which appears to have caused confusion), we have now included a color code legend at the bottom of each figure panel.

Reviewer #3 (Remarks to the Author):

Review of “Bora phosphorylation substitutes in trans for Aurora A kinase T-loop phosphorylation to activate Plk1 and to promote mitotic entry” by Tavernier et al.

This is a very well written manuscript that thoroughly describes important molecular level details of phospho-Bora’s activation of Aurora A, which subsequently activates Plk1. Key regulatory sites on Bora are defined and the most striking mechanistic finding is that a specific phosphosite in Bora (pS112) compensates for activation loop phosphorylation in Aurora A. Overall the conclusions are well supported by the high-quality data and the findings clarify numerous previously reported results for this system. There are some points that need to be clarified:

1) In the discussion, the authors present many good points about the interplay between allosteric activation of Aurora A (via motifs shared by Bora and Tpx-2) and the more novel mechanism of pS112 within Bora replacing the need for Aurora A activation loop phosphorylation. That said, the physiological significance of compensating for lack of activation loop phosphorylation within Aurora A remains unclear.

- a) Can the authors point to specific signaling scenarios where it is known that Aurora A T288 is not phosphorylated?
- b) When/where is a lack of Aurora A T-loop phosphorylation encountered?

It is understood that these questions might not yet have answers and so the discussion can in this case be improved to more clearly describe the state of knowledge and in turn the importance of what the authors call the “essential phosphosite S112”.

We thank the reviewer for raising a very relevant and interesting point.

We have now included a new paragraph in the discussion addressing the potential significance of our discoveries. In brief, it is well established that during expression in the G2 phase of the cell cycle, AURKA autophosphorylation at the T-loop (and hence its partial activation) is counteracted by protein phosphatases. This raises the critical question of how AURKA is activated at the G2-M transition.

Our data suggests that by providing a phosphate moiety on S112 in trans to substitute for a phosphorylated T-loop in cis, phospho-Bora can activate AURKA in spite of counteracting phosphatases. Activation of AURKA by phospho-Bora then initiates a cascade of events that switch the cell from a regime dominated by protein phosphatases to a regime dominated by protein kinases. Consistent with this model, the single site substitution of S112 to alanine in Bora is sufficient to delay mitotic entry. The most obvious next question to decipher, which is beyond the scope of the present study, is how and why phospho-Bora S112 unlike the phosphorylated T-loop of AURKA is resistant to the action of phosphatases.

Finally, the abstract mentions overexpression of mitotic kinases in cancers and yet the link between the Bora focused mechanistic findings and this statement is underdeveloped. Is there evidence of S112 mutation linked to disease?

To our knowledge, there is no evidence of S112 mutation in Bora linked to disease. However, recent reports indicate that Bora is overexpressed in several cancers and that it acts as an oncogene (Zhang et al., 2017; Parrilla et al., 2020). We now make references to these papers in the discussion section of the revised manuscript.

2) There is an odd sentence on page 9: "...phosphorylation of Bora by Cyclin-Cdk appears to significantly enhance affinity of AURKA to a small degree...". Which is it, a significant or small enhancement?

We removed the word "significantly" (we meant to emphasize small but real).

SPR binding data seems to show a bigger difference between Aurora A binding to Bora¹⁻²²⁴ versus phospho-Bora¹⁻²²⁴ than the authors indicate (Fig. 2D & E and corresponding Supplemental data). The 52 micromolar affinity derived from SPR for binding of Aurora A to unphosphorylated Bora is probably an overestimate of the binding affinity; the binding curve (Fig. 2D) does not saturate suggesting that the interaction of Aurora A with unphosphorylated Bora is even weaker than reported—and therefore the increase in affinity promoted by Bora phosphorylation may be larger than the 52 versus 18 μ M affinity values suggest.

This is a very good point. The likelihood that the effect is larger is supported by new data presented in Figure 6B-F, where we compare the affinity of Bora¹⁻²²⁴ and Tpx2¹⁻⁴³ to AURKA^{WT} and AURKA^{T288V} using a fluorescence polarization competitive binding assay. This new data in particular shows that non-phosphorylated Bora¹⁻²²⁴ binds AURKA^{T288V} 23-fold more weakly than phosphoBora¹⁻²²⁴.

While the trends in binding between SPR and FP experiments appear the same, the overall binding affinities appear under-estimated in the SPR experiment relative to the FP experiment (and also based on inhibitor potency as assessed in the ADP-Glo ATPase assay). We speculate that this might be due to reduced accessibility of Bora for binding to AURKA due to its immobilization to the SPR chip. Indeed, the affinities we see in the FP assay are more consistent with the activating effects of Bora on AURKA we see in the ADP-Glo ATPase assay (Figure 6G, H, I).

3) Can the authors explain why the binding of AURKA(T288V) to unphosphorylated Bora (Fig. 3E) seems more robust than the corresponding WT AURKA binding to unphosphorylated Bora (Fig. 2D)? Both in terms of calculated affinity and actual response units?

After suggesting in point (1) that the difference between wt AURKA binding to phosphorylated and unphosphorylated Bora is perhaps greater than the K_d values indicate, for the AURKA T288V mutant it appears there is little difference between the affinity for phosphorylated and unphosphorylated Bora. Is it expected that the T288V mutation would have this effect?

As noted in the prior point, perhaps due to protein immobilization limitations, the SPR experiments in question with Bora and phospho-Bora immobilized to the SPR chip may not be giving a fully nuanced picture of binding in solution. They do however provide qualitative evidence to suggest that phospho-Bora is a better binder to AURKA than non-phospho-Bora.

To investigate this issue more thoroughly, we have now assessed the binding properties of Bora for AURKA in solution by exploiting our discovery that Bora and Tpx2 compete for binding to AURKA.

In brief, we first developed a fluorescence polarization-based assay to detect binding of AURKA to FITC-labeled Tpx2¹⁻⁴³. In new Figure 6B and Figure 6D we show that AURKA^{WT} and AURKA^{T288V} bind to FITC-Tpx2¹⁻⁴³ with a $K_d = 0.009 \mu\text{M}$ and a $K_d = 0.015 \mu\text{M}$, respectively.

We then show that unlabeled Tpx2¹⁻⁴³, pBora¹⁻²²⁴ and Bora¹⁻²²⁴ to varying degrees can competitively displace FITC-Tpx2¹⁻⁴³ from binding to AURKA^{WT} and AURKA^{T288V} (Figure 6C, E, F)

Interestingly, Tpx2¹⁻⁴³ is a better competitor than pBora¹⁻²²⁴ at displacing FITC-Tpx2¹⁻⁴³ from AURKA^{WT} indicating that Tpx2¹⁻⁴³ is a better binder than pBora to AURKA^{WT}. In contrast pBora is better competitor than Tpx2 in displacing FITC-Tpx2¹⁻⁴³ from AURKA^{T288V} indicating that pBora¹⁻²²⁴ is a better binder than Tpx2¹⁻⁴³ to AURKA^{T288V}.

These quantitative biophysical observations not only confirm that Tpx2 and phosphoBora compete for binding to the same surfaces of AURKA, they also provide insight into the relative binding affinities of Bora for AURKA in their different phosphorylated states in solution. Furthermore, these more quantitative binding results are much more consistent with the relative activating effects of Bora on AURKA we see by enzymology using the ADP-Glo ATPase assay (Figure 6G, H, I).

Lastly, the new results clearly indicate that phosphorylation of Bora greatly enhances affinity for AURKA, with the enhancement greatly biased to AURKA^{T288V} than to AURKA^{WT}.

4) The data shown in Fig. S5 raise questions related to Erk phosphorylation. Figs. S5E, F, G and I all seem to indicate more sites of phosphorylation than expected. Can the authors clarify the mass spectrometry data? In particular the fact that the mass spec data suggests that the important mutant that retains S112 (Fig. S5 panel F) is phosphorylated on multiple sites.

It is very difficult to titrate the kinase and time in our reactions to produce preparative quantities of protein with precise phosphorylation states for biochemical experiments. However, to exclude the possibility that a site other than S112 is phosphorylated and contributes to Bora function, we have repeated the experiment in question using Bora¹⁸⁻¹²⁰ with only S112 substituted by Alanine. Our data presented in the new Supplementary Figure 5C shows that S112 phosphorylation is absolutely essential for the ability of pBora to activate AURKA^{T288V}.

5) Possibly related to point 4 it is unclear why Bora 18-120(S27A, S41A, T52A) does not lead to phosphorylated Plk1 in the presence of WT AURKA (Fig. 5B lane 25).

To corroborate the result in question, we have now tested Bora¹⁸⁻¹²⁰ bearing the single S41A mutation in the same experiment. Similar to what we observed for Bora¹⁸⁻¹²⁰ bearing the triple mutations S27A, S41A, and T52A, the single site S41A mutant is not able to activate AURKA^{WT} but retains a robust ability to activate AURKA^{T288V}. Consistent with these in vitro observations, the single mutation S41A in the context of full-length Bora has no effect on mitotic entry in human cells (Figure 9) and in *Xenopus* extracts (Vigneron et al., 2018). We reason that S41 likely plays a

phosphorylation independent role on the binding of Bora and pBora to AURKA, but since Bora binds more weakly to AURKA and AURKA than pBora, that the effect of the mutation on non-phosphorylated Bora is more pronounced. We now make note of a potential role for S41 in the results section as follows:

Since mutation of Ser41 adversely affected the ability of both Bora and pBora to activate AURKA, we reason that this residue may play an important structural and non-phosphorylation dependent role in supporting the Bora-AURKA interaction.

6) On pages 15-16 the authors describe their efforts to localize the putative phospho-binding site on AURKA. Two binding sites are suggested that seem to share two of the three sidechain defining each. It is therefore not clear why the authors describe the second site as 'remote'. As well, the experimental data only probe 3 of the 4 residues listed (Fig. 6B). Overall this section seems less well developed than other parts of the manuscript.

We have now included the R255A mutant of AURKA^{T288V} in our analysis of potential phospho-binding sites and show that unlike AURKA^{T288V}, the AURKA^[T288V R255A] mutant is not stimulated by pBora.

In addition, using a new fluorescence polarization based competitive binding assay, we also show that all four AURKA phospho-site mutants tested (namely AURKA^[T288V H176A], AURKA^[T288V R180A], AURKA^[T288V R255A], and AURKA^[T288V R286A]) retain all or most of their ability to bind Tpx2¹⁻⁴³. This result indicates that the mutations in question did not affect AURKA stability or protein fold.

Finally, by exploiting this same fluorescence polarization assay in a competitive displacement format, we now provide direct evidence that all four AURKA residues in question (namely H176, R180, R255 and R286) contribute to phosphorylation dependent binding to Bora. These new data are presented in new Figure 7.

We have also revised the figure to better illustrate how the four mutated residues of AURKA (H176, R180, R286 and R255) localize on the AURKA structure and how they contact the phosphate moiety of the T-loop or the sulfate ion (Figure 7A).

Reviewer #4 (Remarks to the Author):

Major points:

1. A key finding of the paper is that Bora as an activator of Aurora-A independent of substrate. This conclusion is based on the results of a series of ADP-Glo assays (Figure 3, Figure S3). ADP-Glo measures the concentration of ADP, formed by the hydrolysis of ATP by the kinase. While it is clear that Bora significantly increases the production of ADP by Aurora-A, this also happens whether or not an additional peptide substrate is included (Fig 3C vs Fig S3C).

We agree that the activating effect of Bora on AURKA in the ADP-Glo assay is apparent even in the absence of peptide substrate. However, the effect is more pronounced when peptide substrate is present (1.5 to 2 fold). This is not uncommon for protein kinases, which can transfer phosphate to water in the absence of peptide substrate. It is also possible that the ADP-Glo signal in the kinase reaction reflects phosphorylation of Bora. Indeed, you can see Bora being phosphorylated by AURKA in our PLK1 phosphorylation assay (Figure 1 – compare lanes 1 and 4 and lanes 1 and 5).

I would therefore draw an alternative conclusion: that the increased production of ADP in the presence of Bora is due to Bora acting as a substrate for the kinase, thereby enabling a kinase reaction to take place. The authors should present additional experiments to distinguish between their interpretation and mine. One way to do this is an assay that measures phosphorylation of a peptide (or protein substrate) in the presence of increasing amounts of Bora - could be through incorporation of ^{32}P into a substrate, or an immunoblot against a known Aurora-A substrate (e.g. HH3, TACC3).

As suggested by the reviewer, we have repeated the *in vitro* kinase assays using Histone H3 as a substrate (monitoring Ser10 phosphorylation by immunoblot). Similar to our observation with Plk1 T210 as a substrate, our new data in Supplementary Figure 4A shows that phospho-Bora stimulates AURKA^{T288V} activity towards Histone H3 Ser10.

2. Another key finding of the paper is that phosphorylation of S112 is minimally required for Aurora-A activation (see page 14) - but this conclusion depends on experiments that compare a double mutant (S27A, T52A) with a triple mutant (S27A, T52A, S112A). To draw the conclusion that S112 is minimally required, the experiment (Flg. 5B) should be repeated using Bora with the single S112 mutation.

As requested, we have repeated the *in vitro* kinase assay using Bora¹⁸⁻¹²⁰ with only S112 substituted by alanine. We have also included the analysis of Bora¹⁸⁻¹²⁰ with only S41 substituted by alanine. As shown in new Supplementary Figure S5B, Bora¹⁸⁻¹²⁰ S112A, but not Bora¹⁸⁻¹²⁰ S41A, is defective for the ability to promote AURKA^{T288V} phosphorylation of Plk1 on T210. These results indicate that phosphorylation of Bora¹⁸⁻¹²⁰ on the Cyclin-Cdk site S112, is required to potentiate the catalytic activity of AURKA^{T288V} towards Plk1.

These results are fully consistent with our *in vivo* data indicating that the sole mutation of S112 to alanine in the context of full-length Bora is sufficient to abrogate Bora's biological function in mitotic entry in human cells (Figure 9).

3. Moreover, several subsequent experiments are done using mutations based on a background of the S27A, T52A mutations (e.g. p.15). This complicates the interpretation of experiments only slightly, and so I don't think any other experiments need to be repeated with single mutants. However, it is somewhat confusing, and justification for using this mutant should be clearly stated (is there a technical reason? Or was decided to remove the potential for complications due to phosphorylation at these positions by other kinases?)

Our goal was to reduce the complexity of the system – both for proving the importance of the S112 site in activating AURKA and for attempts to crystallize Bora with AURKA (which we have not had luck in achieving to date). Bora¹⁸⁻¹²⁰ possesses four S/T-P sites that can be efficiently phosphorylated by Cyclin-Cdk kinases or by ERK kinases. To demonstrate that phosphorylation on S112 is sufficient for Bora's function in promoting AURKA^{T288V} activation, we mutated the other three S/T-P sites to Alanine. Our new data clearly demonstrate that Bora S112 phosphorylation is both necessary and sufficient for the activation of AURKA^{T288V} kinase activity (Supplementary Figure 5B).

4. Mutational analysis of Aurora-A to identify the basic residue(s) that respond to Bora phosphorylation is neat and the results are clear (p.16). There is, however, a possibility that one or more of these mutations affects Aurora-A activity directly (e.g. by destabilising the protein), and it would be reassuring to have some evidence that the mutated Aurora-A proteins are stable & active. This should ideally be done experimentally (e.g. thermal stability, proteins retain binding to Bora).

You could also (or instead) refer to data in the McIntyre (2017) paper in your reference list, in which the R286 and R180 mutations affects the binding of TPX2 to Aurora-A phosphorylated on T288, but retain full binding to the unphosphorylated protein.

In the original version of the manuscript, we only tested three of the four residues in AURKA^{T288V} that we predicted might coordinate the phospho-moiety on S112 of Bora (we had trouble cloning the fourth mutant). We have now successfully generated and tested the fourth mutant AURKA^{T288V}_{R255A} and show that it can not be activated by phospho-Bora.

Thus, individual mutation of the four AURKA^{T288V} residues, H176, R180, R255 and R286 abrogates the ability of pBora to potentiate AURKA^{T288V} function towards Plk1 T210.

Since these four mutations might have perturbed AURKA^{T288V} protein function, we used a fluorescence polarization binding assay, to show that all four AURKA^{T288V} mutants retain the ability to bind FITC-labeled Tpx2¹⁻⁴³, indicating that the mutations in question did not greatly perturb the stability or fold of AURKA^{T288V}. Finally, by exploiting this same fluorescence polarization assay, we now provide direct evidence that all four AURKA residues contribute to binding to phospho-Bora. This data is presented in new Figure 7.

These new binding results provides further support to a model whereby phospho-Bora substitutes for phosphorylation of the AURKA T-loop by providing a phosphate moiety (specifically on S112) in trans. The Bora S112 phospho-site is likely coordinated on AURKA by the constellation of H176, R180, R255 and R286 positively charged residues in close vicinity to the catalytic cleft.

We note that our choice of the word 'remote' was suboptimal (as the author mentioned). We have replaced it with the word 'adjacent'.

5. The authors present a model of Bora and TPX2 binding to the same sites on Aurora-A. Testing whether TPX2 and Bora compete for binding Aurora-A (through immunoprecipitation or "pull-downs" using recombinant proteins) would be a straightforward experiment that would strengthen the conclusions of the paper considerably, with minimal effort.

This excellent question was raised by all four reviewers. In response, we first developed a fluorescence polarization-based assay to detect binding of AURKA to FITC-labeled Tpx2¹⁻⁴³. In new Figure 6B and Figure 6D we show that AURKA^{WT} and AURKA^{T288V} bind to FITC-Tpx2¹⁻⁴³ with $K_d = 0.009 \mu\text{M}$ and $K_d = 0.015 \mu\text{M}$, respectively.

We then show that unlabeled Tpx2¹⁻⁴³, pBora and Bora to varying degrees can all competitively displace FITC-Tpx2¹⁻⁴³ from binding to AURKA^{WT} and AURKA^{T288V} (Figure 6C, E, F).

Interestingly, Tpx2¹⁻⁴³ is a better competitor than phosphoBora at displacing FITC-Tpx2 from AURKA^{WT} indicating that Tpx2 is a better binder than pBora to AURKA^{WT}.

In contrast pBora is a better competitor than Tpx2 at displacing FITC-Tpx2 from AURKA^{T288V} indicating that pBora is a better binder than Tpx2 to AURKA^{T288V}.

These quantitative biophysical observations confirm that Tpx2 and pBora compete for binding to the same surfaces of AURKA with variable affinities dependent on the phosphorylation status of both AURKA and Bora.

We have extended these binding studies to compare the effects of Tpx2, Bora¹⁻²²⁴ and pBora¹⁻²²⁴ on AURKA kinase activity using the ADP-Glo assay (Figure 6G, H). Interestingly, although Tpx2¹⁻⁴³ can bind to AURKA^{T288V} (albeit with diminished affinity relative to AURKA^{WT}), it displayed little ability to activate its kinase activity.

Exploiting this behaviour, and the fact that Tpx2¹⁻⁴³ and pBora¹⁻²²⁴ compete for binding to the same surfaces on AURKA, we then showed that titration of Tpx2¹⁻⁴³ (and also Bora¹⁻²²⁴) can competitively abrogate the ability of pBora¹⁻²²⁴ to activate AURKA^{T288V} kinase activity (Figure 6I).

Together, our biophysical binding and enzymology studies unequivocally demonstrate that Bora, phosphoBora and Tpx2 compete for binding to the same surfaces of AURKA. These observations echo previous observations showing that Bora depletion in human cells increases the association of Tpx2 with AURKA on the mitotic spindle while overexpression of N-terminal region of Bora [Bora¹⁻³⁷⁹] prevents co-immunoprecipitation of Tpx2 with AURKA (Chan et al., 2008).

Minor points

1. The purpose of showing the ponceau staining on figure 8e isn't clear to me. I couldn't see any obvious differences between the samples, so is this simply to show equivalent loading of samples on the blot? Please explain!

The purpose of showing ponceau staining is indeed to show equivalent loading.

sous la co-tutelle de

PARIS
DIDEROT
UNIVERSITÉ

INSTITUT JACQUES MONOD
Unité mixte de recherche du CNRS
(UMR7592)
et de l'université Paris Diderot

Bâtiment Buffon - 15 rue Hélène
Brion
75205 Paris cedex 13 - France

Tél. : +33 (0)1 57 27 80 89
Fax : +33(0)1 57 27 80 87
lionel.pintard@jm.fr

I am happy to sign the review.

Prof. Richard Bayliss, University of Leeds, UK.

Thank you very much Professor R. Bayliss for insightful comments on our manuscript.

References

- Archambault, V., Lépine, G., and Kachaner, D. (2015). Understanding the Polo Kinase machine. *Oncogene* 34, 4799-4807.
- Bayliss, R., Sardon, T., Vernos, I., and Conti, E. (2003). Structural basis of Aurora-A activation by TPX2 at the mitotic spindle. *Mol Cell* 12, 851-862.
- Chan, E. H., Santamaria, A., Sillje, H. H., and Nigg, E. A. (2008). Plk1 regulates mitotic Aurora A function through betaTrCP-dependent degradation of hBora. *Chromosoma* 117, 457-469.
- Johnson, T. M., Antrobus, R., and Johnson, L. N. (2008). Plk1 activation by Ste20-like kinase (Slk) phosphorylation and polo-box phosphopeptide binding assayed with the substrate translationally controlled tumor protein (TCTP). *Biochemistry* 47, 3688-3696.
- Kachaner, D., Garrido, D., Mehsen, H., Normandin, K., Lavoie, H., and Archambault, V. (2017). Coupling of Polo kinase activation to nuclear localization by a bifunctional NLS is required during mitotic entry. *Nat Commun* 8, 1701.
- Macurek, L., Lindqvist, A., Lim, D., Lampson, M. A., Klompaker, R., Freire, R., Clouin, C., Taylor, S. S., Yaffe, M. B., and Medema, R. H. (2008). Polo-like kinase-1 is activated by aurora A to promote checkpoint recovery. *Nature* 455, 119-123.
- Noatynska, A., Panbianco, C., and Gotta, M. (2010). SPAT-1/Bora acts with Polo-like kinase 1 to regulate PAR polarity and cell cycle progression. *Development* 137, 3315-3325.
- Parrilla, A., Barber, M., Majem, B., Castellví, J., Morote, J., Sánchez, J. L., Pérez-Benavente, A., Segura, M. F., Gil-Moreno, A., and Santamaria, A. (2020). Aurora Borealis (Bora), Which Promotes Plk1 Activation by Aurora A, Has an Oncogenic Role in Ovarian Cancer. *Cancers (Basel)* 12,
- Seki, A., Coppinger, J. A., Du, H., Jang, C. Y., Yates, J. R., and Fang, G. (2008a). Plk1- and beta-TrCP-dependent degradation of Bora controls mitotic progression. *J Cell Biol* 181, 65-78.
- Seki, A., Coppinger, J. A., Jang, C. Y., Yates, J. R., and Fang, G. (2008b). Bora and the kinase Aurora a cooperatively activate the kinase Plk1 and control mitotic entry. *Science* 320, 1655-1658.
- Tavernier, N., Noatynska, A., Panbianco, C., Martino, L., Van Hove, L., Schwager, F., Léger, T., Gotta, M., and Pintard, L. (2015). Cdk1 phosphorylates SPAT-1/Bora to trigger PLK-1 activation and drive mitotic entry in *C. elegans* embryos. *J Cell Biol* 208, 661-669.

Thomas, Y., Cirillo, L., Panbianco, C., Martino, L., Tavernier, N., Schwager, F., Van Hove, L., Joly, N., Santamaria, A., Pintard, L., and Gotta, M. (2016). Cdk1 Phosphorylates SPAT-1/Bora to Promote Plk1 Activation in *C. elegans* and Human Cells. *Cell Rep* 15, 510-518.

Vigneron, S., Sundermann, L., Labbé, J. C., Pintard, L., Radulescu, O., Castro, A., and Lorca, T. (2018). Cyclin A-cdk1-Dependent Phosphorylation of Bora Is the Triggering Factor Promoting Mitotic Entry. *Dev Cell* 45, 637-650.e7.

Xu, J., Shen, C., Wang, T., and Quan, J. (2013). Structural basis for the inhibition of Polo-like kinase 1. *Nat Struct Mol Biol* 20, 1047-1053.

Zhang, Q. X., Gao, R., Xiang, J., Yuan, Z. Y., Qian, Y. M., Yan, M., Wang, Z. F., Liu, Q., Zhao, H. D., and Liu, C. H. (2017). Cell cycle protein Bora serves as a novel poor prognostic factor in multiple adenocarcinomas. *Oncotarget* 8, 43838-43852.

Zhu, K., Shan, Z., Zhang, L., and Wen, W. (2016). Phospho-Pon Binding-Mediated Fine-Tuning of Plk1 Activity. *Structure* 24, 1110-1119.

REVIEWERS' COMMENTS

Reviewer #1 (Remarks to the Author):

The work establishes that a region of Bora comprising two subsites binds to the catalytic domain of Aurora A, at a region overlapping with the TPX2 binding site and activates Aurora A. In addition a third site, includes a Ser (112) which becomes phosphorylated by CDK2, also participates in the activation, even of a mutant of Aurora A that has mutated the activation loop T288 site to V.

The work further establishes that the phosphorylated Ser interacts with the phosphate binding site where the activation loop phosphate interacts. The study therefore partially explains the molecular mechanism by which Bora enhances the phosphorylation of PLK1 by Aurora A.

In the revised version the authors extend the experimental work significantly and the resulting version is certainly more solid in diverse aspects. On the other hand, the authors do not extend on the role of Bora interaction with the PBD of PLK1 as part of the mechanism of Aurora A phosphorylation of PLK1. The current manuscript delimits its model to the action of Bora on Aurora A and does not attempt to complete the model with the action that another region of Bora produces upon the interaction with the PBD of PLK1.

According to previous work it appears rather likely that the binding of Bora to PLK1 would open the PLK1 structure, to mimic the conformation of the catalytic domain used in this study. The two actions of Bora, on Aurora A and PLK1 are likely related, but the authors did not attempt to explore nor discussed in the paper the possible link between their new finding with the previously known interaction of Bora with PLK1. The authors argue that they limit the study on Aurora A, and not on the whole model, and this is acceptable due to the focus of the work on the Aurora side. However, I see that some parts of the manuscript may still overstate the claim of their experiments. In particular, the subtitle: "A minimal fragment of Bora encompassing residues 18-120 supports AURKA dependent phosphorylation of Plk1 in vitro" could be misleading, as the work presented in figure 5 and Supplementary Fig 5 employ the PLK1 catalytic domain (kinase domain, KD). Before publication, the authors should correct the text of the manuscript, to limit their claims to the catalytic domain of PLK1. In some new addition to the manuscript the authors found that while phosphor-Bora 124 activated Aurora A towards PLK1 and Kemptide, the same phosphor-Bora-124 did not show activation towards another substrate, Histone H3, while this was efficiently activated by the non-phosphorylated form. While this result may shade doubts on the global role of the findings described, I also find that this is interesting. The result would support the idea that Histone H3 may provide its own priming region to favor the active conformation of Aurora A, perhaps interacting with the activation loop phosphate binding site. Small compounds that bind at the main TPX2 binding sites have been developed and activate Aurora A in vitro. Due to the substrate selectivity found at the phosphate-binding site, small compounds interacting with the TPX2-Bora allosteric site could differentially affect the phosphorylation of substrates of Aurora.

Reviewer #2 (Remarks to the Author):

My comments and suggestions, including appropriate references, have been addressed in full. The rebuttal to the 3 other authors also appears comprehensive. I think this is a significant piece of work that should be published.

Reviewer #3 (Remarks to the Author):

The authors have satisfactorily addressed my concerns in the revised manuscript.

Reviewer #4 (Remarks to the Author):

The authors have addressed all of my previous points, and I am satisfied with the revised manuscript. The inclusion of the TPX2/Bora competition data strengthens the conclusions. This is a very interesting study.

Prof. Richard Bayliss

DETAILED RESPONSE TO REVIEWER COMMENTS (author comments in blue)

Reviewer #1 (Remarks to the Author):

The work establishes that a region of Bora comprising two subsites binds to the catalytic domain of Aurora A, at a region overlapping with the TPX2 binding site and activates Aurora A. In addition a third site, includes a Ser (112) which becomes phosphorylated by CDK2, also participates in the activation, even of a mutant of Aurora A that has mutated the activation loop T288 site to V. The work further establishes that the phosphorylated Ser interacts with the phosphate binding site where the activation loop phosphate interacts. The study therefore partially explains the molecular mechanism by which Bora enhances the phosphorylation of PLK1 by Aurora A.

In the revised version the authors extend the experimental work significantly and the resulting version is certainly more solid in diverse aspects. On the other hand, the authors do not extend on the role of Bora interaction with the PBD of PLK1 as part of the mechanism of Aurora A phosphorylation of PLK1. The current manuscript delimits its model to the action of Bora on Aurora A and does not attempt to complete the model with the action that another region of Bora produces upon the interaction with the PBD of PLK1.

According to previous work it appears rather likely that the binding of Bora to PLK1 would open the PLK1 structure, to mimic the conformation of the catalytic domain used in this study. The two actions of Bora, on Aurora A and PLK1 are likely related, but the authors did not attempt to explore nor discussed in the paper the possible link between their new finding with the previously known interaction of Bora with PLK1.

We agree with the reviewer that It is indeed a possibility that the binding of Bora to full length PLK1 would open the PLK1 structure and thus enhance the ability of AURKA (activated by Bora) to phosphorylate the activation segment of PLK1. However, this specific function has not been demonstrated previously in the literature nor in our studies, and thus it remains speculative. We have not discounted this model in our manuscript and describe and cite previous work in the introduction that proposes the possibility of such a mechanism (as shown below). Thus, we do not feel it necessary to further emphasize the point with additional text.

*"Cyclin-Cdk phosphorylates Bora at multiple sites including notably a polo-docking site (S252), which primes Bora binding to the Plk1 PBD {Chan et al., 2008; Seki et al., 2008}. This binding event was proposed to increase the accessibility of T210, thereby supporting T-loop phosphorylation by AURKA {Seki et al., 2008; Hyun et al., 2014}. However, a Bora¹⁻²²⁴ fragment, lacking the polo-docking site S252, can still promote AURKA-dependent phosphorylation of Plk1 on the T-loop {Thomas et al., 2016} and support mitotic entry in *Xenopus* egg extracts {Vigneron et al., 2018}, suggesting that Bora harbors additional functions beyond opening of the Plk1 structure".*

The authors argue that they limit the study on Aurora A, and not on the whole model, and this is acceptable due to the focus of the work on the Aurora side. However, I see that some parts of the manuscript may still overstate the claim of their experiments. In particular, the subtitle: "A minimal fragment of Bora encompassing residues 18-120 supports AURKA dependent phosphorylation of Plk1 in vitro" could be misleading, as the work presented in figure 5 and Supplementary Fig 5

employ the PLK1 catalytic domain (kinase domain, KD). Before publication, the authors should correct the text of the manuscript, to limit their claims to the catalytic domain of PLK1.

We have edited the manuscript text as follows to emphasize that the experiments in question were carried out with the minimal kinase domain of PLK1:

Page 13 Results subsection title:

*“A minimal fragment of Bora encompassing residues 18-120 supports AURKA-dependent phosphorylation of Plk1 **kinase domain** in vitro”*

In some new addition to the manuscript the authors found that while phosphor-Bora 124 activated Aurora A towards PLK1 and Kemptide, the same phosphor-Bora-124 did not show activation towards another substrate, Histone H3, while this was efficiently activated by the non-phosphorylated form. While this result may shade doubts on the global role of the findings described, I also find that this is interesting. The result would support the idea that Histone H3 may provide its own priming region to favor the active conformation of Aurora A, perhaps interacting with the activation loop phosphate binding site. Small compounds that bind at the main TPX2 binding sites have been developed and activate Aurora A in vitro. Due to the substrate selectivity found at the phosphate-binding site, small compounds interacting with the TPX2-Bora allosteric site could differentially affect the phosphorylation of substrates of Aurora.

Reviewer #2 (Remarks to the Author):

My comments and suggestions, including appropriate references, have been addressed in full. The rebuttal to the 3 other authors also appears comprehensive. I think this is a significant piece of work that should be published.

Reviewer #3 (Remarks to the Author):

The authors have satisfactorily addressed my concerns in the revised manuscript.

Reviewer #4 (Remarks to the Author):

The authors have addressed all of my previous points, and I am satisfied with the revised manuscript. The inclusion of the TPX2/Bora competition data strengthens the conclusions. This is a very interesting study.

Prof. Richard Bayliss